# Dilution impacts on smoke aging: Evidence in BBOP data

Anna L. Hodshire[1], Emily Ramnarine[1], Ali Akherati[2], Matthew L. Alvarado[3], Delphine K. Farmer[4], Shantanu H. Jathar[2], Sonia M. Kreidenweis[1], Chantelle R. Lonsdale[3], Timothy B. Onasch[5], Stephen R. Springston[6], Jian Wang[6,a], Yang Wang[7,b], Lawrence I. Kleinman[6], Arthur J. Sedlacek III[6], Jeffrey R. Pierce[1]

[1] Department of Atmospheric Science, Colorado State University, Fort Collins, CO 80523, United States
[2] Department of Mechanical Engineering, Colorado State University, Fort Collins, CO 80523, United States
[3] Atmospheric and Environmental Research, Inc., Lexington, MA 02421, United States
[4] Department of Chemistry, Colorado State University, Fort Collins, CO 80523, United States
[5] Aerodyne Research Inc., Billerica, MA 01821, United States
[6] Environmental and Climate Sciences Department, Brookhaven National Laboratory, Upton, NY 11973, United States
[7] Center for Aerosol Science and Engineering, Washington University, St. Louis, MO 63130, United States
[a] Now at Center for Aerosol Science and Engineering, Washington University, St. Louis, MO 63130, United States
[b] Now at Department of Civil, Architectural and Environmental Engineering, Missouri University of Science and Technology, Rolla, Missouri 65409, United States

*Correspondence to*: Anna L. Hodshire (Anna.Hodshire@colostate.edu)

**Abstract.** Biomass burning emits vapors and aerosols into the atmosphere that can rapidly evolve as smoke plumes travel downwind and dilute, affecting climate- and health-relevant properties of the smoke. To date, theory has been unable to explain observed variability in smoke evolution. Here, we use observational data from the BBOP field campaign and show that initial smoke organic aerosol mass concentrations can help predict changes in smoke aerosol aging markers, number concentration, and number-mean diameter between 40-262 nm. Because initial field measurements of plumes are generally >10 minutes downwind, smaller plumes will have already undergone substantial dilution relative to larger plumes and have lower concentrations of smoke species at these observations closest to the fire.  The extent to which dilution has occurred prior to the first observation is not a directly measurable quantity. We show that initial observed plume concentrations can serve as a rough indicator of the extent of dilution prior to the first measurement, which impacts photochemistry, aerosol evaporation, and coagulation. Cores of plumes have higher concentrations than edges. By segregating the observed plumes into cores and edges, we find evidence that particle aging, evaporation, and coagulation occurred before the first measurement. We further find that on the plume edges, the organic aerosol is more oxygenated while a marker for primary

biomass burning aerosol emissions has decreased in relative abundance than in the plume cores. Finally, we attempt to
decouple the roles of the initial concentrations and physical age since emission by performing multivariate linear regression
of various aerosol properties (composition, size) on these two factors.

**1 Introduction**

Smoke from biomass burning is a major source of atmospheric primary aerosol and vapors (Akagi et al., 2011;

Gilman et al., 2015; Hatch et al., 2015, 2017; Jen et al., 2019; Koss et al., 2018; Reid et al., 2005; Yokelson et al., 2009),
influencing air quality, local radiation budgets, cloud properties, and climate (Carrico et al., 2008; O'Dell et al., 2019; Petters
et al., 2009; Ramnarine et al., 2019; Shrivastava et al., 2017), as well as the health of impacted communities (Ford et al.,
2018; Gan et al., 2017; Reid et al., 2016). Dilution of a smoke plume occurs as the plume travels downwind, mixing with
regional 'background' air, reducing the concentrations of  smoke aerosols and vapors and potentially driving changes in the
physical and chemical properties of the emissions  (Adachi et al., 2019; Akagi et al., 2012; Bian et al., 2017; Cubison et al.,
2011; Hecobian et al., 2011; Hodshire et al., 2019a, 2019b; Jolleys et al., 2012, 2015; Konovalov et al., 2019; May et al.,
2015; Noyes et al., 2020; Sakamoto et al., 2015, Palm et al., 2020).  Fires span an immense range in size, from small
agricultural burns, which may be only a few $m^2$ in total area and last a few hours, to massive wildfires, which may burn
10,000s of $km^2$ over the course of weeks (Andela et al., 2019). This range in size leads to variability in initial plume size and
extent of dilution by the time of the first measurement. Plumes can dilute unevenly, with edges of the plume mixing in with
surrounding air more rapidly than the core of the plume. Hence overall, these large, thick plumes dilute more slowly than
small, thin plumes for similar atmospheric conditions, as the cores of larger plumes are at a greater physical distance to the
background air, shielding them from dilution for longer (Akagi et al., 2012; Bian et al., 2017; Cubison et al., 2011; Hecobian
et al., 2011; Hodshire et al., 2019a, 2019b; Jolleys et al., 2012, 2015; Konovalov et al., 2019; May et al., 2015; Sakamoto et
al., 2015, Lee et al., 2020, Garofalo et al., 2019). Variability in dilution leads to variability in the evolution of smoke
emissions as instantaneous plume aerosol concentrations will control shortwave radiative fluxes (and thus photolysis rates
and oxidant concentrations), gas-particle partitioning, and particle coagulation rates (Akagi et al., 2012; Bian et al., 2017;
Cubison et al., 2011; Hecobian et al., 2011; Hodshire et al., 2019a, 2019b; Jolleys et al., 2012, 2015; Konovalov et al., 2019;
May et al., 2015; Sakamoto et al., 2015, Garofalo et al., 2019, Ramnarine et al., 2019; Sakamoto et al., 2016). Thus,
capturing variability in plume aerosol concentrations and dilution between fires and within fires can aid in understanding
how species change within the first few hours of emission for a range of plume sizes.

The evolution of total particulate matter (PM) or organic aerosol (OA) mass from smoke has been the focus of

many studies, as PM influences both human health and climate. Secondary organic aerosol (SOA) production occurs through
oxidation of gas-phase volatile organic compounds (VOCs) that can form lower-volatility products that partition to the
condensed phase (Jimenez et al., 2009; Kroll and Seinfeld, 2008). SOA formation may also arise from heterogeneous and
multi-phase reactions in both the organic and aqueous phases (Jimenez et al., 2009; Volkamer et al., 2009). In turn, oxidant

concentrations depend on shortwave fluxes (Tang et al., 1998; Tie, 2003; Yang et al., 2009) and the composition of the plume (Yokelson et al. 2009; Akagi et al. 2012; Hobbs et al. 2003; Alvarado et al. 2015). Smoke particles contain semivolatile organic compounds (SVOCs) (Eatough et al., 2003; May et al., 2013), which may evaporate off of particles as the plume becomes more dilute (Huffman et al. 2009; May et al. 2013; Garofalo et al. 2019; Grieshop et al. 2009), leading to losses in total aerosol mass. Field observations of smoke PM and OA mass normalized for dilution (e.g. through a long-lived tracer such as CO) report that for near-field (<24 hours) physical aging, net PM or OA mass can increase (Cachier et al., 1995; Formenti et al., 2003; Liu et al., 2016; Nance et al., 1993; Reid et al., 1998; Vakkari et al., 2014, 2018; Yokelson et al., 2009), decrease (Akagi et al., 2012; Hobbs et al., 2003; Jolleys et al., 2012, 2015; May et al., 2015), or remain nearly constant (Brito et al., 2014; Capes et al., 2008; Collier et al., 2016; Cubison et al., 2011; Forrister et al., 2015; Garofalo et al., 2019; Hecobian et al., 2011; Liu et al., 2016; May et al., 2015; Morgan et al., 2019; Sakamoto et al., 2015; Sedlacek et al., 2018; Zhou et al., 2017). It is theorized that both losses and gains in OA mass are likely happening concurrently in most plumes through condensation and evaporation (May et al. 2015; Hodshire et al. 2019; Hodshire et al. 2019; Bian et al. 2017; Palm et al. 2020), with the balance between the two determining whether net increases or decreases or no change in mass occurs during near-field aging. However, there is currently no reliable predictor of how smoke aerosol mass concentration (normalized for dilution) may change for a given fire.

Evolution of total aerosol number, size, and composition is critical for improving quantitative understanding of how biomass burn smoke plumes impact climate. These impacts include smoke aerosols' abilities to both act as cloud condensation nuclei (CCN) and to scatter/absorb solar radiation (Albrecht, 1989; Petters and Kreidenweis, 2007; Seinfeld and Pandis, 2006; Twomey, 1974; Wang et al., 2008). Particles can increase or decrease in size as well as undergo compositional changes through condensation or evaporation of more volatile compounds. In contrast, coagulation always decreases total number concentrations and increases average particle diameter. Plumes with higher aerosol number concentrations will undergo more coagulation than those with lower concentrations (Sakamoto et al., 2016).

Fires in the western United States region are predicted to increase in size, intensity, and frequency (Dennison et al., 2014; Ford et al., 2018; Spracklen et al., 2009; Yue et al., 2013). In response, several large field campaigns have taken place in the last 7 years examining wildfires in this region (Kleinman et al., 2020; Garofalo et al. 2019; Palm et al., 2020). Here, we present smoke plume observations from the Biomass Burning Observation Project (BBOP) campaign of aerosol properties from five research flights sampling wildfires downwind in seven pseudo-Lagrangian sets of transects to investigate the evolution of OA mass and oxidation state, aerosol number, and aerosol number mean diameter. A range of initial (at the time of the first plume pass in the aircraft) plume OA mass concentrations were captured within these flights and fast (1 second) measurements of aerosols and key vapors were taken. The time resolution of the data was fast enough to segregate each transect into edge, core, or intermediate regions of the plume and examine aerosol properties within the context of both the location within the plume (edge, core, or intermediate) and the initial OA mass loading of the given location. The differences in aerosol loading serve as a proxy for differences in initial fire and plume sizes, mass fluxes, and subsequent amount of dilution. The extent to which dilution has occurred prior to the first observation is not a measurable

quantity, and fire sizes and mass fluxes were not estimated as a part of the BBOP campaign. We create mathematical fits for
predicting OA oxidation markers and mean particle diameter given initial plume OA mass concentration and physical age
(time) of the smoke. These fits may be used to evaluate other smoke datasets and assist in building parameterizations for
regional and global climate models to better-predict smoke aerosol climate and health impacts.

## 2 Methods

The BBOP field campaign occurred in 2013 and included a deployment of the United States Department of Energy
Gulfstream 1 (G-1) research aircraft in the Pacific Northwest region of the United States (Kleinman and Sedlacek, 2016;
Sedlacek et al., 2018) from June 15 to September 13. We analyze five cloud-free BBOP research flights that had seven total
sets of across-plume transects that followed the smoke plume downwind in a Lagrangian manner (see Figs. S1-S6 for
examples; Table S1) from approximately 15 minutes after emission to 2-4 hours downwind (Kleinman and Sedlacek, 2016).
The G-1 sampling setup is described in (Kleinman and Sedlacek, 2016; Sedlacek et al., 2018; Kleinman et al., 2020).
Number size distributions were obtained with a Fast-integrating Mobility Spectrometer (FIMS), providing particle
size distributions nominally from approximately 20-350 nm (Kulkarni and Wang, 2006; Olfert and Wang, 2009); data was
available between 20-262 nm for the flights used in this study. A Soot Photometer Aerosol Mass Spectrometer (SP-AMS)
provided organic and inorganic (sulfate, chlorine, nitrate, ammonium) aerosol mass concentration of PM1 (sub-micron
aerosol) (Canagaratna et al. 2007), select fractional components (the fraction of the AMS OA spectra at a given mass-to-
charge ratio) (Onasch et al., 2012), and elemental analysis (O/C and H/C) (Aiken et al., 2008; Canagaratna et al., 2015).
Extended details on the SP-AMS are provided in Text S1 in the supplementary information, and a brief overview is given
here. The SP-AMS had its highest sensitivity between 70-500 nm, dropping to 50% of peak sensitivity by 1000 nm (Liu et
al. 2007). It was characterized to have a collection efficiency of 0.5 when the instrument's laser was off and 0.76 when the
instrument's laser was on during the BBOP campaign, and these corrections have been applied to the data. There is evidence
from other studies that the CE of the tungsten vaporizer (laser off mode) (Lim et al., 2019) and the laser vaporizer (laser on
mode, run nominally at 600° C) (Willis et al., 2014) to change as a function of chemical composition, rBC coating thickness,
size, and sphericity in laboratory studies (Middlebrook et al., 2012; Willis et al., 2014; Corbin et al., 2015; Massoli et al.,
2015; Collier et al., 2018) and in aircraft observations (Kleinman et al. 2007). Results pertinent to changes in CE due to
aging (including physical aging as well as chemical changes including oxidation, coating thickness, and sphericity) in smoke
plumes are scarce (see discussion in Kleinman et al., 2020). We assume these CEs for the laser on and off modes are
constant in space and time, which is a limitation of this study. We use the calculated $f_{60}$ and $f_{44}$ fractions (the unit mass
resolution mass concentration ratios of m/z 60 and 44 normalized by the total OA mass concentration) and O/C and H/C
elemental ratios of OA as tracers of smoke and oxidative aging. Elevated $f_{60}$ values are indicative of "levoglucosan-like"
species (levoglucosan and other molecules that similarly fragment in the AMS) (Aiken et al., 2009; Cubison et al., 2011; Lee
et al., 2010) that are known tracers of smoke primary organic aerosol (POA) (Cubison et al., 2011). $f_{44}$, the OA fractional
component observed by the SP-AMS as the high-resolution ion fragment $CO_2+$ as well as some acid groups, is a proxy for
SOA arising from oxidative aging (Alfarra et al., 2004; Cappa and Jimenez, 2010; Jimenez et al., 2009; Volkamer et al.,
2006). Fractional components $f_{60}$ and $f_{44}$ have been shown to decrease and increase with photochemical aging, respectively,
likely due to both evaporation and/or oxidation of semivolatile species that contribute to m/z 60 in the SP-AMS and addition
of oxidized species that contribute to m/z 44 in the SP-AMS (Alfarra et al., 2004; Huffman et al., 2009). O/C tends to
increase with oxidative aging (Decarlo et al., 2008) whereas H/C ranges from increasing to decreasing with oxidative aging,
depending on the types of reactions occurring (Heald et al., 2009). Changes in O/C and H/C (as well as changes in total OA
mass, number, f44, and f60) are also influenced by mixing of different air masses and co-oxidation of different VOC
precursors (Chen et al. 2015). Tracking H/C with aging may provide clues upon the types of reactions that may be occurring;
however, variable oxidation timescales can make inferences of this type difficult (Chen et al. 2015). A Single-Particle Soot
Photometer (SP2; Droplet Measurement Technologies) was used to measure refractory black carbon (BC) between 80-500
nm (Schwarz et al. 2010) through laser-induced incandescence (Moteki and Kondo, 2010; Schwarz et al., 2006). An Off-
Axis Integrated-Cavity Output Spectroscopy instrument (Los Gatos, Model 907) measured CO concentrations.  An SPN1
radiometer (Badosa et al., 2014; Long et al., 2010) measured total shortwave irradiance. Kleinman et al. (2020) provides
extensive details for the BBOP instruments used in this work. The supporting information also includes more details on the
instruments used.

To determine the contribution to the concentration of species X  from smoke emissions (ΔX), the background

concentration of X is subtracted off of the measured in-plume species concentrations. To correct for dilution, we normalize
ΔX by background-corrected CO (ΔCO), which is inert on timescales of near-field aging (Yokelson et al., 2009). Increases
or decreases of ΔX/ΔCO along the Lagrangian flight path indicate whether the total amount of X in the plume has increased
or decreased (implying production or removal) since time of emission. The background concentration of X is determined as
a regional average of the observed out-of-plume concentrations of X. To avoid using smoke-impacted measurements we
apply a threshold of only using measurements of X that occur in regions that correspond to the lowest 10% of CO data. We
determine the lowest 10% of CO concentrations for each flight during time periods with a similar altitude, latitude, and
longitude as the smoke plume. We perform sensitivity calculations on our assumptions of background regions and discuss
them in Section 3.

Mass concentrations of O, H, and C are calculated using the O/C and H/C and OA data from the SP-AMS

(assuming all of the OA mass is from O, C, and H, and we acknowledge that omitting lower-abundance atoms, such as S and
N, contributes to some errors in this assumption), allowing us to calculate the background-corrected OA atomic ratios,
ΔO/ΔC, and ΔH/ΔC, following equation 1 (where X = O or H):
$\quad \frac{\Delta X}{\Delta C} = \frac{(X_{in\,plume} - X_{out\,of\,plume})}{(C_{in\,plume} - C_{out\,of\,plume})}$        Eq. 1
We note that any non-linear changes in chemistry and composition between the plume and background will not perfectly
isolate the elemental factors in smoke. We also background-correct fractional $f_{60}$ and $f_{44}$ (using the mass concentrations of
m/z 60, m/z 44, and OA inside and outside of the plume), but we do not normalize by CO due to these values already being
normalized by OA, following equation 2 (where $f = f_{60}$ or $f_{44}$):
$$\Delta f = \frac{(f_{in}*OA_{in}) - (f_{out}*OA_{out})}{\Delta OA} \qquad \text{Eq. 2}$$
We only consider data to be in-plume if the absolute CO >= 150 ppbv. This threshold appears to be capturing clear plume
features as seen in the number concentration while excluding background air (Figs. S7-S11). We note that we use different
definitions of in-plume and background (i.e. the lowest 10% of absolute CO measurements) in order to provide a buffer
between the plume and background to ensure to the best of our abilities that we are capturing non-smoke impacted air for the
background and smoke-impacted air for in-plume cases. The regions of the lowest 10% of CO measurements always fall
under 150 ppbv (Figs. S7-S11). Similarly, we exclude the lowest 5% of CO data in the in-plume measurements in our
analyses to provide a further buffer between smoke-impacted and background air.  We perform sensitivity analyses of our
results to our assumptions about background and in-plume values in Section 3. Figures S2-S6 indicate the locations of the
lowest 10% of CO for each flight.
From the FIMS, we examine the background-corrected, normalized number concentrations of particles with
mobility diameters between 40-262 nm, $\Delta N/\Delta CO$. This size range allows us to exclude potential influence of fresh
nucleation upon the total number concentrations. Occasionally, the background-corrected, normalized number concentration
in the FIMS size range between 20-40 nm increases by 1-2 orders of magnitude relative to typical plume conditions,
indicating possible nucleation events, primarily at the edges or in between smoke plumes (Figs. S7-S11). Smoke plumes
contain particles with diameters larger than 262 nm (Janhäll et al., 2009): thus, we cannot provide total number
concentrations, but we can infer how $\Delta N/\Delta CO$ within our observed size range evolves. We also obtain an estimate of how
the number mean diameter between 40-262 nm, $\overline{D_p}$, changes with aging through:

$$\overline{D_p} = \frac{\Sigma N_i * D_{p,i}}{\Sigma N_i} \qquad \text{Eq. 3}$$

where $N_i$ and $D_{p, i}$ are the number concentration and geometric mean diameter within each FIMS size bin, respectively.
All of the data are provided at 1 Hz and all but the SP-AMS fractional component data are available on the DOE
ARM web archive (https://www.arm.gov/research/campaigns/aaf2013bbop). As the plane traveled at approximately 100 m s$^-$
$^1$ on average, the approximate spatial resolution of the data is every 100 m across the plume. The plumes spanned from
approximately 5-50 km wide (Figs. S2-6). The instruments used here had a variety of time lags (all <10 seconds) relative to a
TSI 3563 nephelometer used as reference. The FIMS also showed additional smearing in flushing smoky air with cleaner air
when exiting the plume with maximum observed flushing timescales around 30 seconds, but generally less (Fig. S12). To
test if these lags impact our results, we perform an additional analysis where we only consider the first half of each in-plume
transect, when concentrations are generally rising with time (Figure S12-S13), and our main conclusions are unaffected. We
do not test the impacts of other time lags and do not attempt to further correct the data for any time lags. Kleinman et al.
(2020) provides further information on instrument time delays during BBOP.
We use MODIS Terra and Aqua fire and thermal anomalies detection data to determine fire locations (Giglio et al.,
2006, 2008). We estimate the fire center to be the approximate center of all clustered MODIS detection points for a given
sampled fire (Figs. S1-S6). The true fire location at the time of sampling is likely different than the MODIS estimates,
depending on the speed of the fire front. To estimate the physical age of the plume, we use the estimated fire center as well
as the total FIMS number concentration to determine an approximate centerline of the plume as the smoke travels downwind
(an example is provided in Fig. S1). The centerline is subjectively chosen to approximately capture the most-concentrated
portion of each plume pass (as estimated using total aerosol number concentrations). We use the mean wind speed and this
estimated centerline to calculate an estimated physical age for each transect, and this physical age is assumed to be constant
across the transect, as plume crossings took between 50-500 seconds; however, transects that were not perfectly tangential to
the mean wind would have sampled different plume ages on the opposite sides of the plume. We did not propagate
uncertainty in fire location, wind speed, or centerline through to the physical age, which is a limitation of this study.
**3 Results and discussion**
As a case example, we examine the aging profiles of smoke from the Colockum fire during the first set of pseudo-
Lagrangian transects for flight 730b (Table S1).  Figure 1 provides $\Delta OA/\Delta CO$, $\Delta BC/\Delta CO$, $\Delta f_{60}$, $\Delta f_{44}$, $\Delta H/\Delta C$, $\Delta O/\Delta C$,
$\Delta N/\Delta CO$, and $\overline{D_p}$ as a function of the estimated physical age; Figs. S14-S18 provides this information for the other pseudo-
Lagrangian transect flight sets studied. (Here, BC represents the refractory BC from the SP2; Sect. 2.) We have divided each
transect into four regions: between the 5-15 (edge), 15-50 (intermediate, outer), 50-90 (intermediate, inner), and 90-100
(core) percentile of $\Delta CO$ within each transect. (As discussed above, we exclude the lowest 5% in order to provide a buffer
between the plume edge and background air.) Note that in Figure 1 (and Figures S14-S18), the points represent the mean
values for each transect/percentile and do not include error bars for uncertainty in the mean or measurement uncertainty as
characterization of systematic variance (within plume percentiles) with age is beyond the scope of this study . Figures S2-S6
show the locations of these CO percentile bins for each transect of individual flights. Figure 1 shows the edge and core data,
both averaged per transect, andFigs. S14-18 provides all four percentile bins for each flight. These percentile bins correspond
with the thinnest (lowest CO mixing ratio) to thickest (highest CO mixing ratio) portions of the plume, respectively. If a fire
has uniform emissions ratios across all regions and dilutes evenly downwind, these percentile bins would correspond to the
edges, intermediate outer and intermediate inner regions, and the core of the diluting plume. We use this terminology in this
study but note that uneven emissions, mixing, and/or dilution lead to the percentile bins not physically corresponding to our
defined regions in some cases. We note that some plumes show more than one maxima in CO concentrations within a given
plume crossing, which implies that there may be more than one fire or fire front, and that these plumes from separate fires or
fronts are not mixing perfectly. Multiple maxima could also imply vertical variations in the location of the core of the

plumes that the flights did not capture. As well, in at least one of the fires (in flights '730a' and '730b'), the fuels vary between different sides of the fire, as discussed in Kleinman et al., (2020). However, the lowest two $\Delta CO$ bins tend towards the physical edges of the plume, and the highest two tend more towards the physical center of the plume (Figs. S2-S6). We do not know where the plane is vertically in the plume, which is a limitation as vertical location will also impact the amount of solar flux able to penetrate through the plume.

Figure 1 shows that for this specific plume, $\Delta OA/\Delta CO$ and $\Delta BC/\Delta CO$ systematically vary little with age for both the 5-15 and 90-100 percentile of $\Delta CO$ (p-values>0.5), yet both show non-systematic variability between transects. A true Lagrangian flight with the aircraft sampling the same portion of the plume and no measurement artifacts (e.g. coincidence errors at high concentrations) would have a constant $\Delta BC/\Delta CO$ for each transect set. This flight and other flights studied here have variations in $\Delta BC/\Delta CO$ (Fig. 1; Figs. S14-S18), which may be indicative of deviations from a Lagrangian flight path with temporal variations in emission and/or measurement uncertainties. The remaining variables plotted also show some noise and few clear trends, but it is apparent that the transect-mean values 5-15 and 90-100 percentiles do show a separation for some of the individual metrics, in particular $\Delta f_{44}$ and $\Delta O/\Delta C$. In order to determine the existence or lack of trends for these metrics, we spend the remainder of this study examining each metric from all of the pseudo-Lagrangian flights together.

### 3.1 Organic aerosol aging: $\Delta OA/\Delta CO$, $\Delta f_{60}$, $\Delta f_{44}$, $\Delta H/\Delta C$, and $\Delta O/\Delta C$

Figure 2a-e shows available $\Delta OA/\Delta CO$, $\Delta f_{60}$, $\Delta f_{44}$ $\Delta H/\Delta C$, and $\Delta O/\Delta C$ edge and core data versus physical age for each transect for each flight of this study. We color each line by the mean $\Delta OA$ within a $\Delta CO$ percentile bin from the transect closest to the fire, $\Delta OA_{initial}$, in order to examine whether each variable ($\Delta OA/\Delta CO$, $\Delta f_{60}$, $\Delta f_{44}$, $\Delta H/\Delta C$, and $\Delta O/\Delta C$) vary with $\Delta OA_{initial}$. (Some transects do not have data available for specific instruments.) As with Fig. 1, the points in Fig. 2 represent the mean values for each transect and percentile, and we do not include error bars as we do not attempt to characterize systematic variance (within plume percentiles) with age in this study. We note that $\Delta OA_{initial}$ does not actually represent the true initial emitted OA from each fire, but instead serves as a proxy for the general fire size, intensity, and emission rate (as larger fires and fires with faster rates of fuel consumption per area will have larger mass fluxes than smaller fires or fires with less fuel consumption per area, all else equal). Thus, $\Delta OA_{initial}$ and other "initial" metrics referred to in this study are not to be taken as emission values and direct comparison to studies with direct emissions values is not appropriate, as dilution and chemistry may occur before the initial flight transect, which we discuss further below. We show the 5-15 (edge) and 90-100 (core) $\Delta CO$ percentile bins in Fig. 2; Fig. S19 shows the same information for all four $\Delta CO$ percentiles. We use the simple 'edge' and 'core' terminology throughout the following discussion but note that the 5-15 and 90-100 $\Delta CO$ percentile bins do not necessarily correspond to the physical (spatial) edges and cores of each plume. They instead correspond to the most CO-dense and least CO-dense portions of the plume. We also note that although some of the physical ages appear to start at approximately 0 hours (e.g. over the fire), this is from a limitation of our physical age estimation

method (Sect. 2), as no flights captured data before approximately 15 minutes after emission (Kleinman et al., 2016). Flights with two sets of pseudo-Lagrangian transects ('726a' and '730b') have two separate lines in Fig. 2, one for each set. As well, two transects for flight '809a' nearly overlap (Fig. S5), with the transect that is further from the fire occurring first in the flight path, leading to an apparent slight decrease in physical age for the sequential transect (see, e.g., the white dashed line in Fig. 2a).

Also included in Fig. 2 are the Spearman rank-order correlation tests (hereafter Spearman tests), which are tests for monotonicity. The Spearman tests show correlation coefficients for each flight set (Table S1) with the initial $\Delta OA$ of a flight set ($\Delta OA_{initial}$) against $\Delta OA/\Delta CO$, $\Delta f60$, $\Delta f44$, $\Delta H/\Delta C$, and $\Delta O/\Delta C$ as the smoke aerosol ages downwind. We also include Spearman tests for the calculated physical age of the smoke for each flight set against these same variables. The R values are labeled $R_{\Delta OA,initial}$ and $R_{age}$, respectively, in Fig. 2. We calculate these correlation coefficients separately for Figure 2 to determine the strength and direction of association for each variable from $\Delta OA_{initial}$ or age alone (and whether the data are correlated vs. anticorrelated with these predictors). To complement these independent correlation coefficients, we also perform multivariate linear regressions (Eqns. 4 and 5 and Figure 3, discussed later) to explicitly decouple the influence of the two predictors. For the correlations with $\Delta OA_{initial}$, all transects in a given pseudo-Lagrangian set of transects have the same $\Delta OA_{initial}$ value; for flights with two pseudo-Lagrangian sets of transects, each set has its own $\Delta OA_{initial}$ value. Correlating to $\Delta OA_{initial}$ provides an estimate of how the plume aerosol concentrations at the time of the initial transect impact plume aging (aging both before and after this initial transect). We define the following categories of correlation for the absolute value of R: 0.0-0.19 is 'very weak', 0.2-0.39 is 'weak', 0.4-0.59 is 'moderate', 0.6-0.79 is 'strong', and 0.8-1.0 is 'very strong' (Evans 1996) .

As individual flights show scatter in the metrics of Fig. 2 (Figs. 1, Figs. S14-S18), we also include $R_{\Delta OA,initial}$ and $R_{age}$ for each metric of Fig. 2 sequentially removing one flight from the statistical analysis. These results are summarized in Table S2. In general, removing single flights does not change our conclusions, particularly when correlations are moderate or stronger. Scatter in $\Delta OA_{initial}$ leads to weaker $R_{age}$ values than would be obtained if we normalized changes with aging to the first (normalized) value. However, as plume-density-dependent aging prior to the first transect is one of the potentially interesting findings of this study, we feel that it is important to not normalize our changes further. Figs. S13, S19-S22 show the same details as Fig. 2 but provide sensitivity tests to our methodology. Figure S13 examines potential FIMS measurement artifacts by only using data from the first 50% of each flight leg when particle concentrations are increasing, which lessons response-time-artifacts of the FIMS during transitions from high to low concentration regions. Figure S20 tests our assumed in-plume CO threshold value by increasing it from 150 ppbv to 200 pbbv (;Fig. S19). Figure S21 tests $\Delta CO$ percentile spacing by changing the bins from 5-15%, 15-50%, 50-90%, and 90-100% to 5-25%, 25-75%, and 75-100%. Figure S22 tests assumed background region by increasing data used from the lowest 10% to the lowest 25% of CO measurements. Although these figures show slight variability, the findings discussed below remain robust, and we constrain the rest of our discussion to the original assumptions made for the FIMS measurements, in-plume CO threshold value, and $\Delta CO$ percentiles used in Fig. 2.

In general, both the cores and edges do not show any positive or negative trend in $\Delta OA/\Delta CO$ with respect to
physical aging. The correlation coefficients, $R_{\Delta OA,initial}$ and $R_{age,}$show very weak correlations of 0.02 and +0.03 (with
$R_{\Delta OA,initial}$ and $R_{age}$ ranging between -0.25 to +0.17 and 0 to 0.07, respectively, when individual flights are left out
sequentially; Table S2). The absolute variability in $\Delta OA/\Delta CO$ is dominated by differences between plumes. Many previous
field campaigns similarly show little change in $\Delta OA/\Delta CO$ with aging (Hodshire et al., 2019a and references therein; Palm et
al., 2020). This may be due to a balance between evaporation and condensation over the period of time that the plume is
observed (Hodshire et al., 2019a).This hypothesis is supported by the observed $\Delta f60$ and $\Delta f44$: The fractional components
$\Delta f_{60}$ and $\Delta f_{44}$ show clear signs of changes with aging, consistent with previous studies (Cubison et al. 2011; May et al. 2015;
Garofalo et al. 2019; Forrister et al. 2015; Lee et al. 2020). $\Delta f_{60}$ generally decreases with plume age ($R_{age}$ = -0.26; a weak
correlation), consistent with the hypotheses that compounds containing species that can fragment to m/z 60  in the SP-AMS
may be evaporating because of dilution, undergoing heterogeneous oxidation to new forms that do not appear at m/z 60,
and/or having a decreasing fractional contribution due to condensation of other compounds. In contrast, $\Delta f_{44}$ generally
increases with age ($R_{age}$ = +0.5; a moderate correlation) for all plumes with available data. It appears for the plumes in this
study that although there is little change in $\Delta OA/\Delta CO$, loss of compounds such as those that contribute to $f_{60}$ fragments (as
captured by the SP-AMS) is roughly balanced by condensation of more-oxidized compounds, including those that contain
compounds with $f_{44}$ fragments, such as carboxylic acids. This observation also suggests the possibility of heterogeneous or
particle-phase oxidation that would alter the balance of $\Delta f_{60}$ and $\Delta f_{44}$. However, estimates of heterogeneous mass losses
indicate that after three hours of aging (the range of time the BBOP measurements were taken in) for a range of OH
concentrations and reactive uptake coefficients, less than 10% of aerosol mass is lost to heterogeneous reactions (Fig. S23;
see SI text S2 for more details on the calculation). These calculations indicate that heterogeneous loss has limited effect on
aerosol composition or mass . Hence, the evaporation of compounds that contribute to m/z 60 in the SP-AMS being balanced
by gas-phase production of compounds that contribute to m/z 44 in the SP-AMS may be the more likely pathway. When
individual flights are left out sequentially, $R_{age}$ ranges from -0.21 to -0.38 and +0.4 to +0.57 for $\Delta f60$ and $\Delta f44$, respectively
(Table S2).
Two more important features of $\Delta f_{60}$ and $\Delta f_{44}$ can be seen within Fig. 2: (1) $\Delta f_{60}$ and $\Delta f_{44}$ depend on $\Delta OA_{initial}$
(moderate correlations of $R_{\Delta OA,initial}$ = +0.43 and -0.55, respectively), with plumes with higher $\Delta OA initial$ having
consistently higher $\Delta f_{60}$ and lower $\Delta f_{44}$. (2) The differences in $\Delta f_{60}$ and $\Delta f_{44}$ are apparent even for the nearest-to-source
measurements that are ~15 minutes after the time of emission. Prior studies have shown that $f_{60}$ and $f_{44}$ at the time of
emissions correlate with OA emissions factors through variability in burn conditions (Hennigan et al. 2011; Cubison et al.
2011; McClure et al. 2020), and this relationship might also contribute to our observed correlation between $\Delta f60$ and $\Delta f44$
with $\Delta OA_{initial}$. For this emissions relationship to be an important factor, the variability in the OA emission factor needs to be
a significant contributor to the variability in $\Delta OA_{initial}$. The relative variability in the OA emission factor is much smaller than
the relative variability in $\Delta OA_{initial}$, and other factors contributing to variability in $\Delta OA_{initial}$ will negate an emissions-based
covariance between $\Delta OA_{initial}$ with $\Delta f_{60}$ and $\Delta f_{44}$. While our observed $\Delta OA_{initial}$ in Figure 2 spans nearly a factor of 100,
Andreae (2019) shows that the OA emission factors have a -1σ to +1σ range of around a factor 3. Hence, variability in fuel
consumption rates and dilution prior to the first transect likely dominate the variability in ΔOAinitial, and the relationships of
$\Delta f_{60}$ and $\Delta f_{44}$ with $\Delta OA_{initial}$ are unlikely to be influenced much by variability in burn conditions. We conclude that
evaporation and/or chemistry prior to the first measurement appears to drive the initial relationship between $\Delta f_{60}$ and $\Delta f_{44}$
with $\Delta OA_{initial}$, consistent with (1) the theoretical work of Hodshire et al. (2019a), (2) an analysis of what chemistry would be
missed in laboratory experiments if the initial 10-60 minutes of chemistry was not considered, following field experiments
(Hodshire et al., 2019b), and (3)  recent field analysis indicating that up to one-third of primary OA from biomass burning
evaporates and subsequently reacts to form biomass burning SOA(Palm et al. 2020) . We include in the supporting
information scatter plots of each parameter of Fig. 1 as a function of ΔOAinitial (Fig. S24), and observe no trends other than
the cores of the plumes generally having a higher $\Delta OA_{initial}$  than the edges of the plumes, as expected.The amount of
evaporation and/or chemistry appear to depend on $\Delta OA_{initial}$, with higher rates of evaporation and chemistry occurring for
lower values of $\Delta OA_{initial}$. This result is consistent with the hypothesis that aircraft observations are missing evaporation and
chemistry prior to the first aircraft observation (Hodshire et al., 2019b). The differences in $\Delta OA_{initial}$ between plumes may be
due to different emissions fluxes (e.g., due to different fuels or combustion phases) or plume widths, where larger/thicker
plumes dilute more slowly than smaller/thinner plumes. These larger plumes have been predicted to have less evaporation
and may undergo relatively less photooxidation (Bian et al., 2017; Hodshire et al., 2019a, 2019b). When individual flights
are left out sequentially, $R_{\Delta OA,initial}$ ranges from +0.3 to +0.58 and -0.42 to -0.63 for $\Delta f60$  and $\Delta f_{44}$, respectively (Table S2).
Garofalo et al.(2019) segregated smoke data from the WE-CAN field campaign by distance from the center of a
given plume and showed that the edges of one of the fires studied have less fractional  $f_{60}$ and more fractional $f_{44}$ (not
background-corrected) than the core of the plume. Lee et al. (2020) saw similar patterns in a southwestern United States
wildfire. Similarly, we find that the 730b flight shows a very similar pattern in  $f_{60}$ and $f_{44}$ (Figs. S25-S26) to that shown in
Fig. 6 of Garofalo et al. (2019). The 821b and 809a flights also hint at elevated $f_{44}$ and decreased $f_{60}$ at the edges but the
remaining plumes do not show a clear trend from the physical edges to cores in $f_{60}$ and $f_{44}$. This could be as CO
concentrations (and thus presumably other species) do not evenly increase from the edge to the core for many of the plume
transects studied (Figs. S2-S6). To more clearly see this, Fig. S27 provides the same style of figure as Figs. S26-S27 for in-
plume CO concentrations. Generally CO peaks around the centerline and is highest in the most fresh transect, but shows
variability across transects. We do not have UV measurements that allow us to calculate photolysis rates but the in-plume
SPN1 shortwave measurements in the visible show a dimming in the fresh cores that has a similar pattern to $f_{44}$ and the
inverse of $f_{60}$ (Fig. S28; the rapid oscillations in this figure could be indicative of sporadic cloud cover above the plumes).
Lee et al. (2020) similarly saw indications of enhanced photochemical bleaching at the edges of a southwestern United States
wildfire when examining aerosol optical properties.
We also plot core and edge ΔH/ΔC and ΔO/ΔC as a function of physical age (Fig. 2d-e). Similar to $\Delta f_{44}$, ΔO/ΔC
increases with physical age and is moderately correlated to both physical age and $\Delta OA_{initial}$ (moderate correlations of $R_{age}$ =+
0.561 and $R_{\Delta OA,initial}$ = -0.45). When individual flights are left out sequentially, $R_{age}$ for ΔO/ΔC ranges between +0.46 and
+0.63 and $R_{\Delta OA,initial}$ ranges between -0.21 and -0.54 (Table S2). Given that $\Delta f_{44}$ and $\Delta O/\Delta C$ are both metrics for OA aging
(Sect. 2), it is unsurprising that we see similar trends between them. Conversely, $\Delta H/\Delta C$ is poorly correlated to physical age
and $\Delta OA_{initial}$.

Both physical age and $\Delta OA_{initial}$ appear to influence $\Delta f_{60}$, $\Delta f_{44}$, and $\Delta O/\Delta C$: oxidation reactions and evaporation

promoted by dilution occur with aging, and the extent of photochemistry and dilution should depend on plume thickness.
Being able to predict biomass burning aerosol aging parameters can provide a framework for interstudy-comparisons and can
aid in modeling efforts. We construct mathematical fits for predicting $\Delta f_{60}$, $\Delta f_{44}$, and $\Delta O/\Delta C$:

$$X = a\, log_{10}(\Delta OA_{initial}) + b\,(Physical\ age)\ + c \qquad\qquad \text{Eq. 4}$$

where $X$ is $\Delta f_{60}$, $\Delta f_{44}$, or $\Delta O/\Delta C$, physical age is in hours, and $a$, $b$, and $c$ are fit coefficients. The measured versus fit data are
shown in Fig. 3a-c. The values of $a$, $b$, and $c$ are provided in Table S3. The Pearson and Spearman coefficients of
determination ($R_p^2$ and $R_s^2$, respectively) are also summarized in Fig. 3 and indicate weak-moderate goodness of fits ($R_p^2$ and
$R_s^2$ of 0.28 and 0.25 for $\Delta f_{60}$, $R_p^2$ and $R_s^2$ of 0.58 and 0.6 for $\Delta f_{44}$, and $R_p^2$ and $R_s^2$ of 0.45 and 0.55 for $\Delta O/\Delta C$). We show $R^2$
here to indicate the fraction of variability captured by these fits, whereas calculating R for the trends in Fig. 2 indicate the
direction of the correlation. We do not constrain our fits to go through the origin. To provide further metrics of goodness-of-
fit, we also include the normalized mean bias (NMB) and normalized mean error (NME) in percent for each metric of Fig. 3.
The NMB values are very close to zero (which is anticipated as linear fits seek to minimize the sum of squared residuals).
The NME is larger, at 19.8% for $\Delta f_{60}$, 14.9% for $\Delta f_{44}$, and 10.2% for $\Delta O/\Delta C$. The p-values for each fit are less than 0.01.
Although no models that we are aware of currently predict aerosol fractional components (e.g. $f_{60}$ or $f_{44}$), O/H and H/C are
predicted by some models (e.g., Cappa and Wilson (2012) and these fit parameters may assist in modeling of aging biomass
burning aerosol. Other functional forms for fits were explored, with the following form showing similar results as Eq. 4:

$$ln(\Delta X) = a\, ln(\Delta OA_{initial}) + b\, ln(Physical\ age)\ + c \qquad\qquad \text{Eq. 5}$$

(Fig. S29 and Table S4 for the fit coefficients) and $\Delta N_{initial}$ in the place of $\Delta OA_{initial}$ in Eq. 4 (Fig. S30 and Table S5 for the fit
coefficients) providing similar correlation values and NMB and NME values for $\Delta f_{60}$, $\Delta f_{44}$, and $\Delta O/\Delta C$.

The aging values of $\Delta f_{60}$, $\Delta f_{44}$, and $\Delta O/\Delta C$ show scatter in time (Figs. S14-18), which likely contributes to the

limited predictive power of our mathematical fits. The scatter is likely due to variability in emissions due to source fuel or
combustion conditions, instrument noise and responses under the large concentration ranges encountered in these smoke
plumes, inhomogeneous mixing within the plume, variability in background concentrations not captured by our background
correction method, inaccurate characterizations of physical age due to variable wind speed, and/or deviations from a true
Lagrangian flight path. Eqs. 4-5 performed the best out of the mathematical fits that we tested. These equations do not have a
direct physical interpretation due to their indirect relations to age and initial aerosol mass. But they may be used as a starting
point for modeling studies as well as for constructing a more physically based fit. There may be another variable not
available to us in the BBOP measurements that can improve these mathematical fits, such as photolysis rates. We do not
know whether these fits may well-represent fires in other regions around the world, given variability in fuels and burn
conditions. We also do not know how these fits will perform under nighttime conditions, as our fits were made for daytime
conditions with different chemistry than would happen at night. We encourage these fits to be tested with further data sets
and modeling. These equations are a first step towards parameterizations appropriate for regional and global modeling and
need extensive testing to separate influences of oxidation versus dilution-driven evaporation.

### 3.2 Aerosol size distribution properties: $\Delta N/\Delta CO$ and $\overline{D_p}$

The observations of the normalized number concentration between 40-262 nm, $\Delta N/\Delta CO$ (Fig. 2f), show that plume
edges and cores generally show decreases in $\Delta N/\Delta CO$ with physical age, with a weak correlation of $R_{age}$ = -0.27 (-0.13 to -
0.43 when individual flights are left out, sequentially; Table S2). Although we would anticipate that plume regions with
higher initial $\Delta OA$ would have lower normalized number concentrations due to coagulation (Sakamoto et al. 2016), a few
dense cores have normalized number concentrations comparable or higher than the thinner edges, leading to no correlation
with $\Delta OA_{initial}$. We note that variability in number emissions (e.g., due to burn conditions) adds unexplained variability not
captured by the R values.
The mean particle size between 40-262 nm, $\overline{D_p}$ (Eq. 3), is shown to statistically increase with aging when
considered across the BBOP dataset (Fig. 2g) (a moderate correlation of $R_{age}$ = +0.53, with $R_{age}$ ranging between +0.43 to
+0.63 when individual flights are left out sequentially; Table S2). Coagulation and SOA condensation will increase $\overline{D_p}$. OA
evaporation will decrease $\overline{D_p}$ if the particles are in quasi-equilibrium (where evaporation is independent of surface area)
(Hodshire et al. 2019b). However, if evaporation is kinetically limited, smaller particles will preferentially evaporate more
rapidly than larger particles, which may lead to an increase in $\overline{D_p}$ if the smallest particles evaporate below 40 nm (Hodshire
et al. 2019b). The plumes do not show significant changes in $\Delta OA/\Delta CO$ (Fig. 2a), indicating that coagulation is likely
responsible for the majority of increases in $\overline{D_p}$. (We acknowledge that $\Delta OA/\Delta CO$ may be impacted by measurement artifacts
as discussed in Sect. 2. For instance, if the collection efficiency of the AMS is actually decreasing with age, then $\Delta OA/\Delta CO$
would be increasing and the increases in number mean diameter will be due to SOA condensation as well as coagulation.)
We do not have measurements for the volatility of the smoke aerosol, and so cannot refine these conclusions further. We also
perform the functional fit analysis following Sect. 3.1 (Eq. 4; where $X$ is $\overline{D_p}$ in this case). The fit can also predict greater than
30 percent of the variance in $\overline{D_p}$ ($R_P^2$ and $R_s^2$ of 0.37 and 0.33, NME of 5.5%, and p-value less than 0.01; Fig. 3d) but does
not predict $\Delta N/\Delta CO$ well (not shown). We show the functional fit for $\overline{D_p}$ for the alternative fit equation (Eq. 5) in Fig. S29
and Table S4. We also show the functional fit for $\overline{D_p}$ for Eq. 4 with $\Delta N_{initial}$ in place of $\Delta OA_{initial}$ in Fig. 30 and Table S5.
Sakamoto et al. (2016) provide fit equations for modeled $\overline{D_p}$ as a function of age, but they include a known initial $\overline{D_p}$ at the
time of emission in their parameterization (rather than 15 minutes or greater, as available to us in this study), which is not
available here. $\Delta N_{initial}$ in the place of $\Delta OA_{initial}$ in Eq. 4 predicts $\overline{D_p}$ similarly (Fig. S30). As discussed in Section 3.1, scatter
in number concentrations limits our prediction skill.
Particles appear in the 20-40 nm size range in the FIMS measurements independently of plume OA concentrations
(Figs S7-S11), implying that nucleation events may be occuring for some of the transects . Some pseudo-Lagrangian sets of
transects also show nucleation-mode particles downwind of fires in between transects (Figs. S7, S8, S9, and S11).
Nucleation-mode particles appear to be approximately one order of magnitude less concentrated than the larger particles, and
primarily occur in the outer portion of plumes, although one set of transects did show nucleation-mode particles within the
core of the plume (Fig. S11). Nucleation at edges could be due to increased photooxidation from higher total irradiance
relative to the core (Fig. S26). As well, nucleation is more favorable when the total condensation sink is lower (e.g. reduced
particle surface area; Dal Maso et al., 2002), which may occur for outer portions of plumes with little aerosol loading.
However, given the relatively small number of data points showing nucleation mode particles and limited photooxidation
and gas-phase information, we do not have confidence in the underlying source of the nucleation-mode particles.
**4 Summary and outlook**
The BBOP field campaign provided high time resolution (1 s) measurements of gas- and particle-phase smoke
measurements downwind of western U.S. wildfires along pseudo-Lagrangian transects. These flights have allowed us to
examine near-field (<4 hours) aging of smoke particles to provide analyses on how select species vary across a range of initial
organic aerosol mass loadings ($\Delta OA_{initial}$; a proxy for the relative rates at which the plume is anticipated to dilute as dilution
before the first observation is not a measurable quantity). We have also examined how the species studied vary between the
edges and cores of each plume. We find that although $\Delta OA/\Delta CO$ does not correlate with $\Delta OA_{initial}$ or physical age, $\Delta f_{60}$ (a
marker for evaporation) is moderately correlated with $\Delta OA_{initial}$ (Spearman rank-order correlation tests correlation coefficient,
$R_{\Delta OA,initial}$, of +0.43) and weakly correlated with physical age (Spearman rank-order correlation tests correlation coefficient,
$R_{age}$, of -0.26). $\Delta f_{44}$ and $\Delta O/\Delta C$ (markers for photochemical aging) increases with physical aging (moderate correlations of $R_{age}$
of +0.5 and +0.56, respectively) and are inversely related to $\Delta OA_{initial}$ (moderate correlations of $R_{\Delta OA,initial}$ of -0.55 and -0.45,
respectively). $\Delta N/\Delta CO$ decreases with physical aging, likely through coagulation. Mean aerosol diameter increases with age
primarily due to coagulation, as normalized organic aerosol mass does not change significantly, and is moderately correlated
with physical age ($R_{age}$ = +0.53). Nucleation is observed within a few of the fires and appears to occur primarily on the edges
of the plumes. Differences in initial values of $\Delta f_{60}$, $\Delta f_{44}$, and $\Delta O/\Delta C$ are evidence that evaporation and/or chemistry has
occurred before the time of initial measurement and that plumes or plume regions with lower initial aerosol loading can undergo
these changes more rapidly than thicker plumes. We have developed fit equations that can weakly to moderately predict $\Delta f_{60}$
, $\Delta f_{44}$, $\Delta O/\Delta C$, and mean aerosol diameter given a known initial (at the time of first measurement) total organic aerosol mass
loading and physical age. We were unable to quantify the impact on potential inter-fire variability in the emission values of
the metrics studied here (such as variable emissions of species that can contribute to m/z 60 and m/z 44). We anticipate that

being able to capture this additional source of variability may lead to stronger fits and correlation. We encourage future studies to attempt to quantify these chemical and physical changes before the initial measurement using combinations of modeling and laboratory measurements, where sampling is possible at the initial stages of the fire and smoke. We also suggest further refinement of our fit equations, as additional variables (such as photolysis rates) and better quantification of inter-fire variability (such as variable emission rates) are anticipated to improve these fits. We finally urge future near-field (<24 hours) analyses of recent and future biomass burning field campaigns to include differences in initial plume mass concentrations and location within the plume as considerations for understanding chemical and physical processes in plumes.

## Acknowledgements

We would like to thank Lauren Garofalo, Emily Fischer, Jakob Lindaas, and Ilana Pollack for useful conversations. We thank Charles Long for use of irradiation data. This work is supported by the U.S. NOAA, an Office of Science, Office of Atmospheric Chemistry, Carbon Cycle, and Climate Program, under the cooperative agreement awards NA17OAR4310001 and NA17OAR4310003; the U.S. NSF Atmospheric Chemistry program, under Grants AGS-1559607 and AGS-1950327; and the US Department of Energy's (DOE) Atmospheric System Research, an Office of Science, Office of Biological and Environmental Research program, under grant DE-SC0019000. Work conducted by LIK, AJS, JW was performed under sponsorship of the U.S. DOE Office of Biological & Environmental Sciences (OBER) Atmospheric System Research Program (ASR) under contracts DE-SC0012704 (BNL; LIK, AJS) and DE-SC0020259 (JW). Researchers recognize the DOE Atmospheric Radiation Measurement (ARM) Climate Research program and facility for both the support to carry out the BBOP campaign and for use of the G-1 research aircraft. TBO acknowledges support from the DOE ARM program during BBOP and the DOE ASR program for BBOP analysis (contract DE-SC0014287). DKF acknowledges funding from NOAA Climate Program Office's Atmospheric Chemistry, Carbon Cycle, and Climate program (Grant NA17OAR4310010). We thank the anonymous reviewers for their constructive feedback.

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

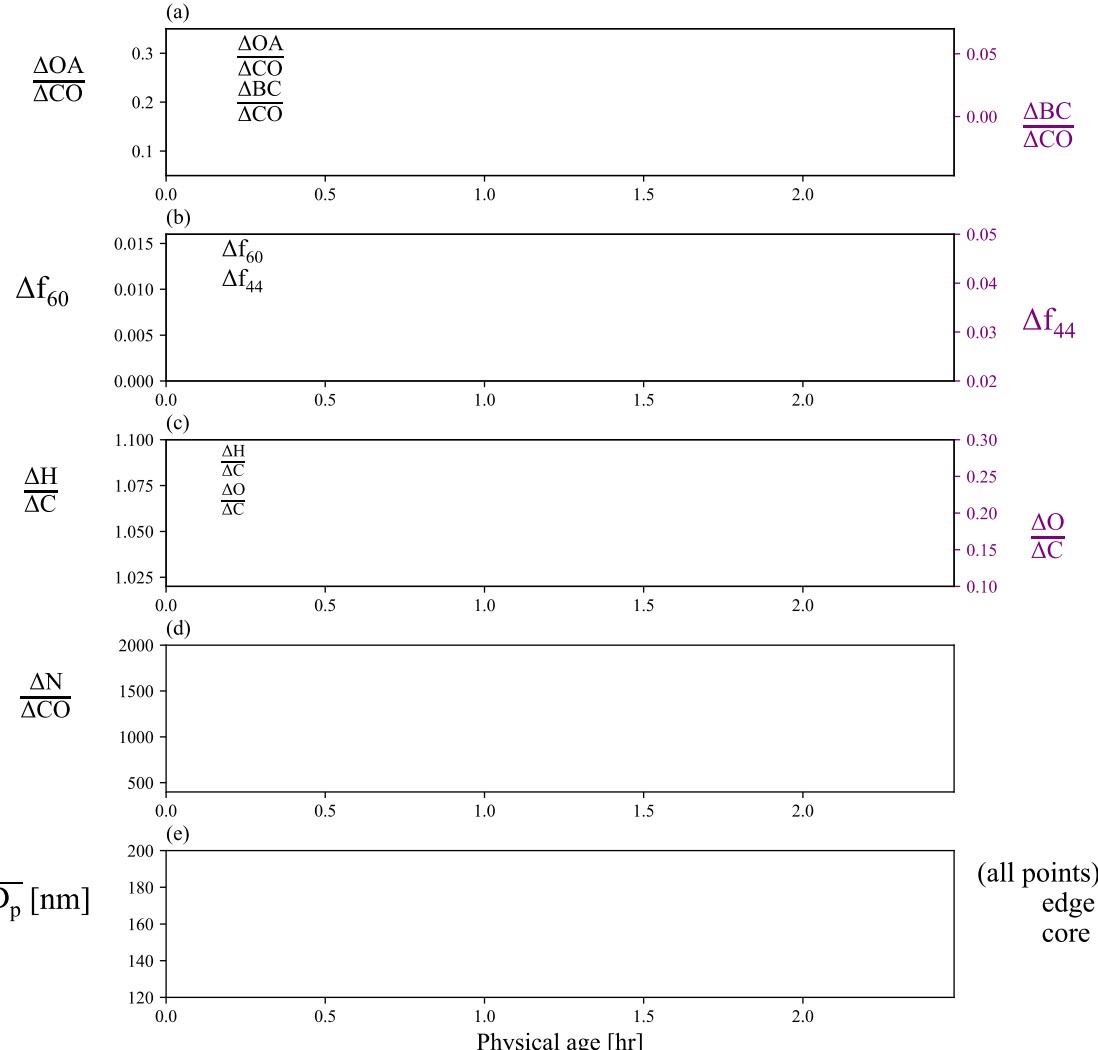


**Figure 1: Aerosol properties from the first set of pseudo-Lagrangian transects from the Colockum fire on flight '730b' (a) ΔOA/ΔCO (right y-axis) and ΔBC/ΔCO (left y-axis), (b) Δf$_{60}$ (right y-axis) and Δf$_{44}$ (left y-axis), (c) ΔH/ΔC (right y-axis) and ΔO/ΔC (left y-axis), (d) ΔN/ΔCO, and (e) $\overline{D_p}$ against physical age. For each transect, the data is divided into edge (the lowest 5-15% of ΔCO data; red points) and core (90-100% of ΔCO data; blue points).**

831

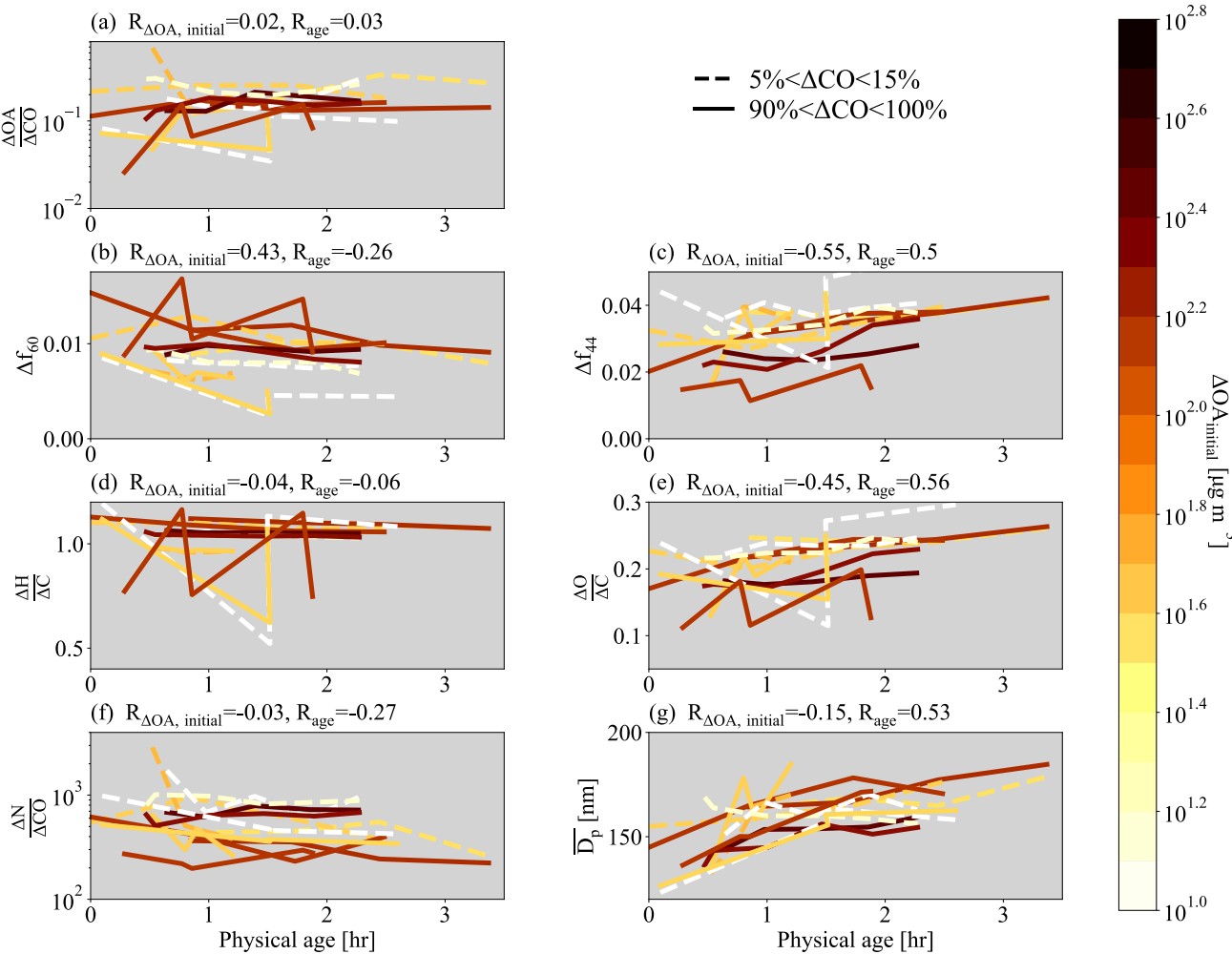

**Figure 2.** Various normalized parameters as a function of physical age for the 7 sets of pseudo-Lagrangian transects. Separate lines are shown for the edges (lowest 5-15% of ΔCO; dashed lines) and cores (highest 90-100% of ΔCO; solid lines). (a) ΔOA/ΔCO, (b) Δ$f_{60}$, (c) Δ$f_{44}$, (d) ΔH/ΔC, (e) ΔO/ΔC, (f) ΔN/ΔCO, and (g) $\overline{D_p}$ between 40-262 nm against physical age for all flights, colored by ΔOA$_{initial}$. Some flights have missing data. Also provided is the Spearman correlation coefficient, R, between each variable and ΔOA$_{initial}$ and physical age for each variable. Note that panels (a) and (f) have a log y-axis.

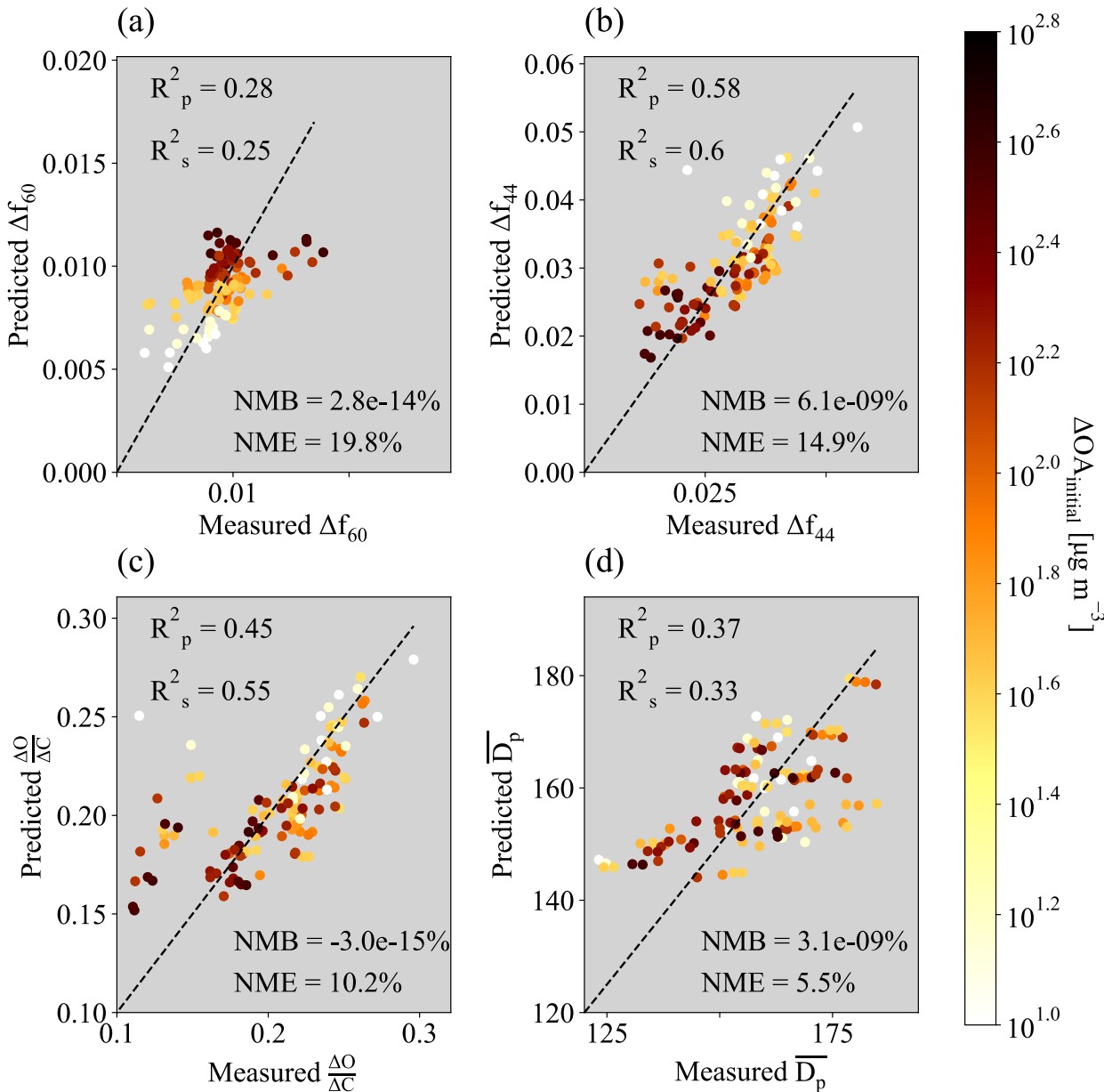

**Figure 3. Measured versus predicted (a) $\Delta f_{60}$, (b) $\Delta f_{44}$, (c) $\Delta O/\Delta C$, and (d) $\overline{D_p}$ between 40-262 nm. The predicted values are from**
**the equation $X = a \log_{10}(OA_{initial}) + b$ (Physical age) $+ c$ where $X = \Delta f_{60}$, $\Delta f_{44}$, $\Delta O/\Delta C$, or $\overline{D_p}$. The values of $a$, $b$, and $c$ are provided in**
**Table S3. The Pearson and Spearman coefficients of determination ($R_p^2$ and $R_s^2$, respectively) are provided in each panel, along**
**with the normalized mean bias (NMB) and normalized mean error (NME). Note that Fig. 2 provides R values rather than $R^2$ to**
**provide information upon the trend of the correlation. Included in the fit and figure are points from all four $\Delta CO$ regions within**
**the plume (the 5-15%, 15-50%, 50-90%, and 90-100% of $\Delta CO$), all colored by the mean $\Delta OA_{initial}$ of each $\Delta CO$ percentile range.**