# Peer review of "Dilution impacts on smoke aging: Evidence in BBOP data"

_Atmospheric Chemistry and Physics, 2020_

## Referee Comment (RC1) · Anonymous Referee #1 · 16 Apr 2020

Overall, I find this an interesting paper that addresses an important topic and builds nicely on previous work by the authors. However, I have a number of concerns regarding the inherent assumptions made or implied throughout and how thoroughly they are justified, and regarding the consistency of the interpretations provided. I find there are also a number of areas where more detail is required. I think that this work might be publishable after substantial revision. My specific comments and questions follow below.

L54: It is not clear to me how plume thickness controls gas-particle partitioning or particle coagulation rates. Both depend on concentrations, not thickness. I suggest the authors clarify whether they really mean "thickness" here and on L58.

L65: Do oxidant concentrations not also depend on the composition of the plume?

L67: The authors cite Formenti et al. (2003) as support of dilution occurring. However, they might note that the particular conclusion in Formenti et al. (2003) really derives from the observation of a single, high concentration point for the "fresh" samples that controls the linear regression. If that point is excluded, the slopes of the fresh and aged EC vs. OC curves are nearly identical.

L79: Much of this paragraph seems redundant with material already presented. I suggest it be streamlined. The only new information is the slightly greater information regarding coagulation.

L94: I suggest that the authors here define what they mean by "initial." This is a critical feature of this study. Only later is it clear that "initial" means "the closest we got to the fire for a given flight."

L112: The authors should note the size range of the SP-AMS measurements, and the size range of the SP2 measurements (L126).

L125: The authors might also note that the atomic ratios are strongly affected by mixing of different air masses and the co-oxidation of different VOC precursors, which start at different points on a van Krevelen diagram. Different VOCs in the plumes will age on a variety of timescales, giving rise to an evolving O:C and H:C regardless of "aging" of the sort implied here. Mixing and co-oxidation affect the H:C, especially, making inferences of the "types of reactions occurring" challenging. This is discussed in (Chen et al., 2015). See later comment on the same subject.

L130: The authors note that the supporting info provides "more details on the instruments used." I find this misleading. The information provided in the SI is extremely limited, hardly greater than that provided in this paragraph. I suggest the authors provide in the SI some discussion at least of instrumental uncertainties.

L138: I suggest it be clarified how f60 and f44 are background corrected. Presumably this is not a straight difference, as the denominators ([OA]) differ. Is it, for example

f60_corrected = (f60_plume*[OA]_plume – f60_bgd*[OA]_bgd)/[OA]_plume? If the authors used a straight difference, this must be justified as it does not seem appropriate to me. Similarly, more details on how the other intensive properties (O:C, H:C) are corrected are needed.

L140: It would be helpful if in Figs. S2-S6 and S7-S11 the authors would number each plume so that the the two can be related to each other. It would also help if the time-series were shown as an additional panel with the spatial plots, again so comparisons can be made. I think this is important because the authors discuss "plumes" here, but they do not discuss how it is, for example, that in a given transect there can be multiple maxima in CO. Does this imply there are two plumes? Or is this the same plume? What drives this behavior, and what might it indicate about the evolution of the plumes? What does it mean to define a "centerline" of the plume if there are clearly two distinct maxima on either side (see Fig. S3, for example).

From Figs. S7-S11, it appears that the background [CO] varies from flight-to-flight. For example, in Fig. S7 the background is clearly lower than the 150 ppb threshold the authors have used, but in Fig. S9 it is barely sufficient. Why not define a flight-specific background [CO] based on the observations?

L156: The authors note that the instruments had various time lags, but it is not clear whether they were all adjusted to account for these varying time lags. This should be clarified. Also, it would be helpful if the authors clarified whether they really mean a "lag" but with a fast response time (i.e., two instruments both show sharp changes but are offset) or whether they are referring to some amount of smearing in which previous measurements affect the current measurement. From the FIMS discussion, it sounds as if they are actually talking about smearing (related to instrument response time) and not a lag.

L165: Further details regarding how the FIMS data were used to establish the centerline are needed. How were the number distributions used specifically? How were these

determined for different transects to give a single straight line? Also, is wind speed as measured by the aircraft?

Fig. 1: The figure lacks error bars. Given the analysis, it would seem that precision-based propagated uncertainties would be appropriate, as the authors seem interested more in characterizing changes than they are absolute values. I suggest appropriate error bars are added.

L182: While it seems that the 5-15 percentile values are primarily found at the physical edges of the plumes shown in the supplemental, as often as not the 90-100 percentile values exhibit bimodal behavior across a transect, often occurring relatively close to the physical edge. From what is shown, I do not believe it is justified to say that the 90-100 percentile "core" corresponds to the physical "core" of the plume as observed. I strongly suggest the authors to define a quantitative metric to relate the percentiles to the spatial distribution. Perhaps a normalized distance from the centerline.

L191: I suggest the authors be more precise in their claims. The normalized number concentration in the "core" does not change with age, and at the edge the entirety of the change is observed from the first transect to the second. And there is perhaps an increase in diameter from the first transect to the next, but the diameter is constant (within variability) for all transects further downwind. Also, the deltaO/C does not increase with aging. The authors indicate that the delta_f44 changes with age, but it is not clear how this was determined. Was some sort of linear fit done? Is this just the difference between the first point and the last? Visually, the points look scattered about a flat line. Overall, for this discussion I think that the authors need to be more specific and precise and quantitative. As currently written, it is not always clear how the authors came to the conclusions that they did.

L203: I find it exceptionally difficult to understand exactly what the authors have done with the Spearman rank-order correlation tests. The authors need to be much more specific. The authors have one value for (e.g.,) initial plume OA mass but then have

multiple values for the deltaOA/deltaCO for each transect of a given plume. Then there are multiple plumes. How are the data merged to allow comparison across all plumes? Physical age makes more sense, as (for example) deltaOA/deltaCO can be regressed versus physical age for each plume. But, to me, how the other parameters are used (OA initial and deltaOAinitial) is unclear. Are all the initial OA values repeated for a given flight? Are the authors using only the initial values for the other parameters to compare with initial OA?

L213: What does it mean for something to "evaporate off through heterogeneous aging?" Things can evaporate, or they can be heterogeneously oxidized. These are distinct processes.

L210: The authors note that the changes in deltaOA/deltaCO with aging are small. A recent review by the authors (Hodshire et al., 2019) indicates a variety of reasons for such behavior. Another recent paper (Lim et al., 2019) introduces another potential reason for this behavior, specifically potential biases in the measurement of OA as the particle composition evolves. Have the authors considered this?

With reporting the Spearman's correlation coefficient I suggest the authors use consistent language that links to typical interpretation of the level of significance (that a relationship is monotonic). For example, a value of -0.25 (as determined for f60) might be considered "weak" while a value of 0.54 (for f44) is "moderate." Also, the authors might note when introducing the Spearman's test that it is a test for monotonicity.

There appears to be a good deal of flight-to-flight variability in behavior, from Fig. 2. This raises a question of how much of the inferred behavior (from the Spearman's test) derives from fairly strong changes in one flight. The authors might consider testing the sensitivity to their analysis by determining Spearman's coefficients when systematically leaving out individual flights or transects one at a time. This would give a broader sense of the robustness of the results, given the notable scatter.

L217: Nitpicky, but compounds do not "contain f44." Certain compounds fragment in

such a way that they show up at m/z 44 in the AMS. But overall this sentence is a run on with a second half that does not logically follow from the first. The sentence starts by talking about a balance between condensation and evaporation but shifts abruptly to note something about heterogeneous oxidation or particle-phase reactions. I suggest the authors clarify the point they are aiming to make here.

L219: The authors note that deltaOA/deltaCO does not change much. This would be consistent with the little mass loss that the authors note from heterogeneous oxidation here, correct? Are the authors aiming to make a point more specifically about the efficiency with which heterogeneous oxidation might degrade the f60 signal and not about mass loss? I find it unclear.

Laboratory observations (Cubison et al., 2011;Hennigan et al., 2011;Hodshire et al., 2019;McClure et al., 2020) have demonstrated that the f60 and f44 of freshly emitted particles vary over large ranges dependent on the fuel type and specific burn condition. Is it not possible that the differences in deltaf60 and deltaf44 between flights result from intrinsic differences in the emitted particle properties? The authors seem to discount this without explicit justification when they state that their interpretation assumes that "emitted deltaf60 and deltaf44 do not correlate with deltaOAinitial." Might there not be an initial correlation, as this might indicate some difference in the burn conditions or the particular fuel mix? I can certainly believe that "evaporation and/or chemistry likely occurred before the time of" the first measurements, however it is not clear to me that the observations as presented here demonstrate this conclusively. Also, given that different sources produce particles that have different initial f60 and f44, would they be expected to exhibit the same deltaf60 and deltaf44 even if initial OA and dilution were identical? Is there evidence that this is expected?

L243: I disagree with the authors interpretation of the van Krevelen diagram here. The authors interpret this in a process based way related to chemistry. However, this does not account for the fact that this is, likely, ultimately a mixing experiment wherein primary OA is being increasingly mixed with secondary OA. This cannot be interpreted in

terms of functional group addition. Additionally, it is not clear that a plot of deltaO/deltaC vs deltaH/deltaC should behave in the same way as a plot of O/C vs H/C. The authors must demonstrate the equivalency of these.

deltaO/deltaC ratios: I am somewhat surprised that these values are positive. O:C ratios of fresh biomass burning tend to be around 0.3-0.4 whereas O:C of background OA are typically large. (The same is true for f44.) The authors should comment on the very fact that their deltaO/deltaC values are positive.

Eqn. 2: First, what is the justification for this functional form? Is there some other form that would better explain the data? Second, in terms of utility, is it really most useful to predict the delta values, as these will depend explicitly on the background, which may vary between locations? Do the authors expect these relationships will prove robust and applicable to other regions? Would these be appropriate at night as well as during the day? The authors have not been able to distinguish between dilution-driven changes and oxidation-driven changes, so there may be distinct day/night differences? When would they expect them applicable? How could these parameters assist specifically in biomass burning models? Presumably such models would aim to be processed based, differentiating between oxidation and dilution.

When the authors report the Pearson's coefficients, are these constrained to go through the origin? The authors show only the 1-1 lines, but visually it seems that any linear fit to the calculated vs. observed relationship will have a non-zero intercept unless constrained. In this context, having a good $r^2$ value is simply an indication of a linear relationship but it is not an indication of the goodness of the calculated vs. observed. Instead, the authors would need to provide some metric such as normalized mean bias. As presented, I am not convinced that the $r^2$ values are particularly meaningful.

L263: It is not clear to me what the authors are getting at when they state that aged deltaf60 and deltaf44 show scatter, limiting the predictive skill of measurements available from BBOP. They had just discussed how there are "moderate goodness of fits." It

seems now that they are contradicting themselves. Or perhaps they are just providing more context for what "moderate" means.

L273: While the authors state here that highest initial deltaOA generally has the lowest normalized number concentrations, this seems to contradict their near zero Spearman's coefficient reported in Fig. 2. In fact, the authors state this two lines later. This needs to be revised. Either there is a correlation or there is not.

L276: Is variability in number emissions really "noise?" It seems like an inherent feature.

L278: Does the particle size really increase for "all" plumes, or does it statistically increase when considered across all plumes? There seem to be some lines in the graph that are basically flat when considered individually; thus, I am not certain that the "all" applies.

L280: As mentioned above, have the authors considered other potential artifacts in their deltaOA/deltaCO that might lead to this parameter remaining flat while the apparent particle size increases? I suggest this be discussed in the context of the authors' conclusion that coagulation drives the size change.

L283: The authors have been assuming that it is acceptable to use as an "initial" OA and particle concentration the value measured in the closest transect for each flight. Given this assumption, it is unclear why the authors now indicate it is essentially inappropriate to estimate an initial particle diameter from the closest transect to use for comparison with the model of Sakamoto et al. (2016). If the assumption is poor for one variable how is it justified that it is okay for two other variables?

Equation 2: What units must the time have?

L290: Nucleation is generally more favorable when existing particle surface area is smaller, as the condensation sink is reduced. Might this also be an explanation for the greater incidence of nucleation near plume edges?

L294: The authors note that the nucleation mode "appears to be coagulating or evaporating away as the plumes travel downwind." It would be useful if they show this explicitly in some way. Which figures should the reader look at specifically and which intersects? I find this overall too vague and suggest that it needs to be made more explicit.

L303: Again, does "thicker" here mean "more concentrated"? Thickness, which I would interpret to mean some spatial thickness, is not discussed in this paper as best I can tell. Regardless, the authors cannot conclude that deltaN/deltaCO is lower for "thicker" plumes since their Spearman's coefficient is essentially zero.

L308: Again, how can the authors rule out differences in the initial conditions that are independent of physical or chemical aging? This seems to be an underlying assumption throughout this entire study, but I do not find that the authors have really justified this assumption. Given how central it is to everything, I strongly suggest that an explicit discussion must be included wherein the authors review the evidence for and against their assumption.

Minor:

L47: It might be more accurate to say that the smoke plumes dilute through entrainment of background air rather than that they dilute and entrain background air.

Chen, Q., Heald, C. L., Jimenez, J. L., Canagaratna, M. R., Qi, Z., Ling-Yan, H., Xiao-Feng, H., Campuzano-Jost, P., Palm, B. B., Poulain, L., Kuwata, M., Martin, S. T., Abbatt, J. P. D., Lee, A. K. Y., and Liggio, J.: Elemental composition of organic aerosol: the gap between ambient and laboratory measurements, Geophysical Research Letters, 42, 4182-4189, https://doi.org/10.1002/2015gl063693, 2015. Cubison, M. J., Ortega, A. M., Hayes, P. L., Farmer, D. K., Day, D., Lechner, M. J., Brune, W. H., Apel, E., Diskin, G. S., Fisher, J. A., Fuelberg, H. E., Hecobian, A., Knapp, D. J., Mikoviny, T., Riemer, D., Sachse, G. W., Sessions, W., Weber, R. J., Weinheimer, A. J., Wisthaler, A., and Jimenez, J. L.: Effects of aging on organic aerosol from open biomass burning smoke in aircraft and laboratory studies, Atmos. Chem. Phys., 11, 12049-12064, https://doi.org/10.5194/acp-11-12049-2011, 2011. Hennigan, C. J., Miracolo, M. A., Engelhart, G. J., May, A. A., Presto, A. A., Lee, T., Sullivan, A. P., McMeeking, G. R., Coe, H., Wold, C. E., Hao, W. M., Gilman, J. B., Kuster, W. C., de Gouw, J., Schichtel, B. A., Collett Jr, J. L., Kreidenweis, S. M., and Robinson, A. L.: Chemical and physical transformations of organic aerosol from the photo-oxidation of open biomass burning emissions in an environmental chamber, Atmospheric Chemistry and Physics, 11, 7669-7686, https://doi.org/10.5194/acp-11-7669-2011, 2011. Hodshire, A. L., Akherati, A., Alvarado, M. J., Brown-Steiner, B., Jathar, S. H., Jimenez, J. L., Kreidenweis, S. M., Lonsdale, C. R., Onasch, T. B., Ortega, A. M., and Pierce, J. R.: Aging Effects on Biomass Burning Aerosol Mass and Composition: A Critical Review of Field and Laboratory Studies, Environmental Science & Technology, 53, 10007-10022, https://doi.org/10.1021/acs.est.9b02588, 2019. Lim, C. Y., Hagan, D. H., Coggon, M. M., Koss, A. R., Sekimoto, K., de Gouw, J., Warneke, C., Cappa, C. D., and Kroll, J. H.: Secondary organic aerosol formation from the laboratory oxidation of biomass burning emissions, Atmos. Chem. Phys., 19, 12797-12809, https://doi.org/10.5194/acp-19-12797-2019, 2019. McClure, C. D., Lim, C. Y., Hagan, D. H., Kroll, J. H., and Cappa, C. D.: Biomass-burning-derived particles from a wide variety of fuels – Part 1: Properties of primary particles, Atmos. Chem. Phys., 20, 1531-1547, https://doi.org/10.5194/acp-20-1531-2020, 2020.

---

## Referee Comment (RC2) · Anonymous Referee #2 · 19 Apr 2020

I am uploading this review as a supplement.

Please also note the supplement to this comment:
https://www.atmos-chem-phys-discuss.net/acp-2020-300/acp-2020-300-RC2-supplement.pdf

---

## Referee Comment (RC3) · Anonymous Referee #2 · 1 May 2020

**Review of "Dilution impacts on smoke aging: Evidence in BBOP data"**

By Anna L. Hodshire et al.

Anonymous Reviewer

*Summary:*

This manuscript uses airborne data of wildfire smoke plumes, measured as pseudo-lagrangian transects of the plumes during the 2013 BBOP field campaign. Physical ages of the plumes ranged from approximately 15 minutes to 2-4 hours.

The authors analyze the oxidation state (through f44, f60, O/C, and H/C) as well as mean particle diameter and the OA/CO emission ratio of aerosol in terms of physical plume age and the aerosol's proximity to the plume core. They demonstrate enhanced chemical aging/oxidation at the edges of plumes that they argue is related to enhanced photolysis in more dilute BBOA-containing air.

Only a couple studies have discussed the effects of chemical aging in terms of plume thickness and edge-to-core position. This is a very informative and fascinating approach and is a great use of archived data BBOP data to build upon previous modeling research. The paper is well cited and the figures are generally aesthetically pleasing. Please don't be dismayed by the criticism to follow as I tend to focus on the things that need to be fixed. There are a lot of good observations and analysis in this paper which I don't, but maybe should, highlight.

I believe that many of the conclusions are likely true, however the way the data was analyzed does not always support this and I have made quite a few comments regarding this. In my opinion, a focus should be made on comparisons within transect sets regarding how things evolve with physical age and generalizations of plume cores vs plume edges instead of on bulk regressions (Spearman's correlations) which are not particularly convincing (either low R-values or R-values reflective of outlier data). Additionally, there seems to be a lot of contradicting statements made in interpreting the results. This is potentially a very good and interesting paper relevant to the subject areas of ACP and eventually should be published, but obviously will require significant edits.

*General Comments:*

1) Figures are aesthetically pleasing but could use some minor changes.
2) Format of citations need to be fixed.
3) There are a lot of typos and issues with word choice which will need to be fixed before final publication.
4) I am curious, how wide were the plumes and how long did it take to fly through them? It seems like you explored whether instrument lags affected your results, but during a

transect did the physical age of the leading the plume edge vary significantly from the edge when you left the plume?

5) I think you can better clarify how you estimate physical age. In the supplementary files, the "core" trajectory is a straight line, presumably because you use a single wind speed and direction, but the core of the transect frequently does not lie on that line. Could this be improved with Hysplit/WRF models? Would that help the core of the transect fall along the dashed line?

6) Data are broken down into physical age and further into fringe-vs-core (such as shown in Figure 1). These data-points represent a range of data subsample in time and space and therefore should include error bars representing the variance in data represented by each data point as well as the measurement uncertainty.

7) $\Delta f60$ and $\Delta f44$ are known to vary in primary emissions, even in laboratory experiments where nascent soot can be analyzed (i.e. not after 10+ minutes of aging). However, a key assumption in many of the conclusions seems to be that all primary BBOA has the same initial $\Delta f60$ and $\Delta f44$. This is a problem when the authors try to support their conclusions.

8) The use of Spearman's rank-correlation is fine as you may not expect linearly increasing/decreasing values with physical (or even chemical) age. But it needs to be clearly stated that this is a test of monotonically increasing/decreasing values, which does not give the same predictive interpretations as a Pearson's correlation.

Interpretation of Spearman's correlation coefficients and the strength of these coefficients, in many cases, do not support the interpretations presented in this work. Part of this is because the authors chose to combine all data from all flights together for the regressions. This means that data representing older physical age of a plume with high initial concentrations is mixed together with data representing young physical age but low concentrations. The result is that there is not a strong relationship between these parameters (e.g. $\Delta N/\Delta CO$) and physical age (or $\Delta OA_{initial}$). If these transects were normalized in some other way, maybe these statements may be more supportive of the conclusions.

9) The supplementary text provides very little additional information. There seems to be some confusion regarding methodology which could be explained in more detail here. I would suggest a cartoon of a flight path showing how you chose your background for a transect.

10) Were all supplementary sections/figures referenced in the text? I lost count.

*Specific Comments:*

L30: Be more specific about what you mean by "smoke concentrations… aging markers, number, diameter."

L34-35: You state that it is not quantifiable how diluted a plume is when first measured; does this contradict the next statement that (hence) the initially measured (number?) concentration is a proxy for dilution?

L37: Do you mean "increases in oxidative tracers" or that the oxidation-state of OA at the edges was higher?

L44-47: "...rapidly evolve as smoke travels downwind, diluting and entraining background air." I think you mean that dilution and entrainment can rapidly cause aerosol & vapor evolution, but that is not how it reads.

L49: I think you mean "dilution at time of measurement".

L54: Does this refer to radiative fluxes?

L 55-57: Please fix the brackets around citations.

L93: Should read "aging and oxidation of OA mass and aerosol number concentration and mean diameter."

L112: 20-262 nm size range is not ideal, but I guess it is what you have.

L134-135: Also background correct m/z=44 and m/z=60?

L 136: Conceptually, where does the lowest 10% of CO occur? Just outside of the plume as the plane circles back through? Is the background fairly constant for a flight leg? Do you adjust background each time the plane turns around and goes back to transect the plume again?

L137: Is elemental O, H, and C calculated from O/C, H/C & OA or is H/C and O/C calculated from the elemental O, H, C concentrations? Aiken et al (2007) estimate it in the later (Eqn 1).

L 139: Typo ("…, we but do not…")

L164-165: Sentence grammar

L165-167: Why use the FIMS # distribution to determine plume center? Why not [CO], [mrBC], total number concentration, etc? In the supplemental figures, it says the center-flow is determined by number concentration (not distribution).

L170: Fix heading

L189: Measurement uncertainty should be plotted in Figures (sum of variance in data represented by each data point + uncertainty in each instrumental recording)

L189-190: Changes in f60 and f44 should be provided as fractional (as displayed on axis of Figure 1, etc). Relative changes (%) are confusing.

L192: Replace "number concentration" with either "normalized number concentration" or "$\Delta N_{40\text{-}262\ nm}/\Delta CO$".

L192: I only see a decrease in $\Delta N_{40\text{-}262\ nm}/\Delta CO$ between ~0.6 and 1.0 hours physical age. Saying that it decreases with age implies a consistent trend. For Dp, this trend is hard to tell if it is statistically significant.

L197: What do you mean by "available …"?

L197-199: Really long sentence. I have had to read it 6-7 times to parse out what is shown.

L200-201: Physical age is the distance between the transect-center to the fire-center divided by the average windspeed? So does 0 physical age imply infinite or 0 windspeed?

L203: The "…correlation coefficients (R) with initial plume OA mass,…" is not shown. Do you mean to say that this is represented by $\Delta OA_{initial}$?

L202-204: Is the Spearman coefficient for concatenation of all data points from all transects? If so, I am not sure it would make sense to do this way. Spearman's test tests for monotonically increasing/decreasing values. Given that each transect set starts at a different initial value you wouldn't expect the grouped transect sets to display a strong R-value. If you want to use Spearman's test in this way, for $R_{age}$ you could normalize each normalized value to the initial normalized value to get a % change and plot that in Figure 2 and relevant supplementary figures.

L206: Spell out "Figs." And lower case.

L207-208: Type in list "…FIMS measurements AND BACKGROUND and $\Delta CO$ percentile spacings…"

L209: Previous line said you would only discuss FIMS, background and $\Delta CO$.

L209-210: $R_{\Delta OA,initial}$ just says 0 in figure.

L209-210: This figure shows orders of magnitude changes in $\Delta OA/\Delta CO$ with age. I think you mean there is not a clear positive or negative trend (as stated in the first clause of the next sentence), not that there is no change.

L212: Here and elsewhwere, spell out "vs." Check grammar.

L213: For positive R values, consider putting a "+" sign in front of the value.

L214-218: Consider breaking this into multiple, shorter sentences. Check for redundancy with L212-214, i.e. a negative R value means there is a decreasing trend.

L214-218: Is it only evaporation or condensation (phase changes) happening or does O attack volatile and semivolatile species (levoglucosan) changing its molecular composition to more oxidized/refractory species without a phase change?

L218-220: If you didn't expect a change in normalized-OA anyway based on your model, why do you suggest a balance between evaporation particle mass loss and condensation mass gain?

L221: Those are not very strong R values to base your interpretations on, but I wouldn't expect them to be for the reasons discussed above. This statement is not particularly true for f60.

L224: But you just said that $\Delta$f60 and $\Delta$f44 correlate with $\Delta OA_{initial}$. Differences in your initial $\Delta$f60 or $\Delta$f44 don't necessary need a mechanistic explanation. We see variance these parameters in fresh emission in laboratory experiments and would expect to also see variance in primary emissions of wildfires. This is not good support for your next conclusion (that aircraft observations are missing evaporation and/or condensation).

L227: Is this logic circular? That differences in $\Delta OA_{initial}$ is due to different emission fluxes?

L228: should not be a comma after the bracket.

L231 & 234: Reference format needs to be changed.

L234: Grammar. Reference to figure in Garofalo should be something like "(Fig. 6 in Garofalo et al, 2019)"

L235-236: Isn't that why you normalize?

L237-239: You imply that patterns of f60 and f44 compared to shortwave irradiance is related by photolysis rates. I don't necessarily agree with this interpretation. If the plume is thicker it means that a higher fraction of aerosol mass is from the fire and because fire-emitted aerosol has higher f60 and lower f44 than background a simple mechanism of mixing explains your observations.

L242-243: $\Delta O/\Delta C$ and f44 are both proxies for OOA and would be expected to have the same trends. $\Delta H/\Delta C$ and f60, while not conceptually the same, both reflect primary BBOA and would also be expected to show the same trends It is a little redundant to analyze both sets.

L242-243: See issues raised earlier regarding interpreting Spearman's test results for these data sets.

L249-264: You should provide explanation for why you used these equations to try and fit f44 and f60. Is there a conceptual justification for them? Do they have meaning outside of a mathematical fit?

L263-268: What do you mean by "Aged $\Delta$f60 and $\Delta$f44"? Does "limiting the predictive skill" mean that your fits are not particularly informative?

L264-265: typos/grammar

L271-272: The decrease in normalized number concentration with physical age mostly appears to be caused by 2-3 outlier measurements (the initial points for leg 730b edge, the initial value of another edge, and the tailing value of leg 726a 1). This does not seem like a statistically robust claim and I think the R value verifies it. Lines 275-277 seem to agree with my assessment.

L273-274: "generally have lower normalized … by the time of the first measurement". This implies that there was a measurement made before the first measurement. Please explain.

L273-274: "plume edges and cores with the highest $\Delta$OA generally have lower normalized number concentrations…" This is not true based on figure 2f. The two lowest $\Delta$OA$_{initial}$ values (white dashed lines) have two of the highest $\Delta$N/$\Delta$CO values.

L279: Evaporation (mass loss/time) is, partially, a function of available surface area. Since small particles have a higher surface area-to-volume, it is plausible that evaporation will decrease the number of small particles more than large particles and therefore increase the mean particle size.  You state this possibility of preferential loss of small particles on lines 293-295.

L282-283: should be $R_P{}^2$ instead of $R^2{}_P$.

L282-283: you were previously using R and not $R^2$ (L272, Fig 2, etc). In my opinion, this is fine and depends on how you use them, but I have been reviewed differently. Did you intend to calculate R and R2? Please check to make sure that you they are used and calculated correctly. I only state this because there are a number of typos in the manuscript and want to make sure that this is not one.

L287: Do you mean "legs" instead of "days"?

L294: Replace "~" with "approximately"

L301-302: As mentioned above, I do not agree that the data supports the statement regarding correlation. I think there is a lot of good analysis in this paper and I don't think you need to make this statement.

L302-304: I also do not agree that the data supports the statement regarding $\Delta$N/$\Delta$CO.

L304: You don't need to keep specifying that diameter size range of 40-262.

L306-308: I don't like saying this, I don't agree that your data support this statement. The only way that differences in $\Delta f44_{initial}$, $\Delta f60_{initial}$ and $\Delta O/C_{initial}$ support this statement is if all primary OA from all wildfires have the same value which has been shown to not be true.

Figure 1: Change "BC" to "rBC" in the legend and axis. Also in Figures S14-S18

Figure 1: Change $\Delta N/\Delta CO$ to $\Delta N_{40\text{-}262\,nm}/\Delta CO$ to be consistent with text.

Figure 2: Caption should be "function of physical age"

Figure 2: This figure is pretty confusing. If I look at Figure S2, I see that for leg 726a there were 2 sets of transects with each comprising of 4 transects. So, theoretically, the same air mass was sampled 4 times corresponding to 4 different physical ages. So a line in figure 2 contains ~4 data points which correspond with either the edge or core of a transect in the transect set? Am I reading this correct?

   How does the white dashed line in 2a go backwards in physical age?

Figure 2: Change to $R_{\Delta OA,initial}$ instead of double subscript to be consistent with that used in text.

Figure S1: I don't see a black star or dashed line.

Figure S1: Leg number not indicated. ("The numbers are the leg number")

Figure S1: I would suggest that you use a different symbol and symbol color for the MODIS thermal anomalies so that it contrasts with the color code of the # concentration.

Figure S1: Please change the colorcode to a color-blind friendly one.

Figure S5: Is the black star the fire center for 8/9/2013 or 8/8/2013? The caption does not say what symbol is used for 8/8/2013, only that "The black star indicates the approximate center of the fire…"

Figure S24-S25: The y-axis scale changes between graphs, with a wide range for data that do not look like they have much variation (leg 730a) and a smaller range for others (730b). Is this why there is not consistent patterns in 730a and 730b?

Figure S26: is shortwave irradiance a measure of photo-chemical rate, the amount of scattering/absorbing aerosol above you, or a combination of both?

Figure S27: Please complete the drawing of the Van Krevelen diagram with the 1:1, 2:1, and 0.5:1 lines.

---

## Author Response (AR1)

**Cover letter for responses and track changes documents for 'Dilution impacts on smoke aging: Evidence in BBOP data' (Hodshire et al., 2020)**

Please note--due to substantial updates to our supporting information, we provide the track changes version of the SI as well. All of our changes have been documented in the reviewer responses and track-changes documents.

The reviewer responses start on page 2 of this document.

Our track-changes main text starts on page 51 of this document.

Our track-changes SI starts on page 82 of this document.

Reviewer responses for 'Dilution impacts on smoke aging: Evidence in BBOP data'

We thank the reviewers for their helpful comments. To aid the review process, we are placing reviewer comments in black text, our responses in blue text, any changes to the text in red, and, in some instances, reproduce text from the previously submitted manuscript (*italic magenta*). We have numbered the reviewer comments to assist the conversation.

Due to the length of the reviews and responses we provide here the page numbers of the start of each review:
Review 1 and responses: page 3
Review 2 and responses: page 24

First we would like to note that we found a minor error in our code that calculates the locations of the lowest 10% of out-of-plume CO that we use to determine our background region. This error led to us not including all of the locations (indexes) of this background region for each flight. Fortunately, when we fixed the error, none of our conclusions changed and all values shifted only slightly. We have updated all figures, tables, and text that depends on background corrections and note that the changes in our moderate and strong correlation coefficients (see Fig. 2 for instance) do not exceed 8%.

We note the recent publication of Lee et al. (2020) that focuses on aerosol optical properties in a southwestern US wildfire that has also looked at differences between edge and core. We have added the following text in Sect. 3.1 (new text underlined for emphasis)

(Garofalo et al., 2019) segregated smoke data from the WE-CAN field campaign by distance from the center of a given plume and showed that the edges of one of the fires studied have less $f_{60}$ and more $f_{44}$ (not background-corrected) than the core of the plume; Lee et al. (2020) saw similar patterns in a southwestern United States wildfire.

And

We do not have UV measurements that allow us to calculate photolysis rates but  the in-plume SPN1 shortwave measurements in the visible show a dimming in the fresh cores that has a similar pattern to $f_{44}$ and the inverse of $f_{60}$ (Fig. S26; the rapid oscillations in this figure could be indicative of sporadic cloud cover above the plumes). (Lee et al. 2020) similarly saw indications of enhanced photochemical bleaching at the edges of a southwestern United States wildfire when examining aerosol optical properties.

Lee, J. E., Dubey, M. K., Aiken, A. C., Chylek, P., & Carrico, C. M. (2020). Optical and chemical analysis of absorption enhancement by mixed carbonaceous aerosols in the 2019

Woodbury, AZ fire plume. Journal of Geophysical Research: Atmospheres, 125, e2020JD032399. https://doi.org/10.1029/2020JD032399

We have noticed that we did not include any discussion of the fit equations that we developed (Eqs. 4-5 in the revised manuscript), despite spending significant time on them in the text. We have included the following statements in the conclusions:

"We have developed fit equations that can weakly to moderately predict $\Delta f_{60}$, $\Delta f_{44}$, $\Delta O/\Delta C$, and mean aerosol diameter given a known initial (at the time of first measurement) total organic aerosol mass loading and physical age."

"We also suggest further refinement of our fit equations, as further variables (such as photolysis rates) and better quantification of interfire variability (such as variable emission rates) are anticipated to improve these fits."

Finally, we note that we have made updates to many SI figures. In order to hopefully keep this document more navigable, we only rarely have included an updated SI figure here and instead point the reviewers to our marked-up SI document to assess these changes. We have also made many small edits to the main text to improve sentence structure, readability, and grammar (as noted a few times specifically by reviewer 2).

**Review 1**

Overall, I find this an interesting paper that addresses an important topic and builds nicely on previous work by the authors. However, I have a number of concerns regarding the inherent assumptions made or implied throughout and how thoroughly they are justified, and regarding the consistency of the interpretations provided. I find there are also a number of areas where more detail is required. I think that this work might be publishable after substantial revision. My specific comments and questions follow below.

R1.1) L54: It is not clear to me how plume thickness controls gas-particle partitioning or particle coagulation rates. Both depend on concentrations, not thickness. I suggest the authors clarify whether they really mean "thickness" here and on L58.

We agree that "thickness" is vague and that "concentration" is more clear. We have changed "thickness" to "aerosol concentration" in both instances as we are really referring to the aerosol concentration.

R1.2) L65: Do oxidant concentrations not also depend on the composition of the plume?

Yes, this was an oversight on our part. We have updated the text to read:

"In turn, oxidant concentrations depend on shortwave fluxes (Tang et al., 1998; Tie, 2003; Yang et al., 2009) and the composition of the plume (Yokelson et al. 2009; Akagi et al. 2012; Hobbs et al. 2003; Alvarado et al. 2015)."

R1.3) L67: The authors cite Formenti et al. (2003) as support of dilution occurring. However, they might note that the particular conclusion in Formenti et al. (2003) really derives from the observation of a single, high concentration point for the "fresh" samples that controls the linear regression. If that point is excluded, the slopes of the fresh and aged EC vs. OC curves are nearly identical.

This point is a subtlety that we did not capture with our original statement. Upon re-review of Formenti et al. (2003), we see that the authors state "...as our data for the elemental versus organic carbon ratio suggest that organic carbon might have evaporated while in the atmosphere." (Sect 3.4) However, the authors do not directly explicitly connect evaporation with dilution in their manuscript, and we have chosen to remove this citation. We replace it with (Garofalo et al. 2019; Grieshop et al. 2009).

R1.4) L79: Much of this paragraph seems redundant with material already presented. I suggest it be streamlined. The only new information is the slightly greater information regarding coagulation.

We respectfully disagree and believe this paragraph stands alone--it connects prior discussion to
climate-relevant aerosol properties, which have not been discussed yet.

R1.5)  L94: I suggest that the authors here define what they mean by "initial." This is a critical
feature of this study. Only later is it clear that "initial" means "the closest we got to the fire for a
given flight."

This is another oversight on our part--we have updated the text to read:

"A range of initial (at the time of the first plume pass in the aircraft) plume OA mass
concentrations were captured within these flights and sufficiently fast (1 second) measurements
of aerosols and key vapors were taken."

R1.6) L112: The authors should note the size range of the SP-AMS measurements, and the size
range of the SP2 measurements (L126).

We have added the following text (with added underlines as guides)  for the SP-AMS:

"A Soot Photometer Aerosol Mass Spectrometer (SP-AMS) provided organic and inorganic
(sulfate, chlorine, nitrate, ammonium) $PM_1$ aerosol masses (Canagaratna et al. 2007), select
fractional components (the fraction of the AMS OA spectra at a given mass-to-charge ratio)
(Onasch et al., 2012), and elemental analysis (O/C and H/C) (Aiken et al., 2008; Canagaratna et
al., 2015). The SP-AMS had the highest sensitivity between 70-500 nm, dropping to 50%
transmission efficiency by 1000 nm (Liu et al. 2007). "

And for the SP2:

" A Single-Particle Soot Photometer (SP2; Droplet Measurement Technologies) was used to
measure refractory black carbon (rBC) between 80-500 nm (Schwarz et al. 2010) …"

R1.7) L125: The authors might also note that the atomic ratios are strongly affected by mixing
of different air masses and the co-oxidation of different VOC precursors, which start at different
points on a van Krevelen diagram. Different VOCs in the plumes will age on a variety of
timescales, giving rise to an evolving O:C and H:C regardless of "aging" of the sort implied here.
Mixing and co-oxidation affect the H:C, especially, making inferences of the "types of reactions
occurring" challenging. This is discussed in (Chen et al., 2015). See later comment on the same
subject.

We agree that we did not expand upon this discussion as much as we could have, and we thank
the reviewer for the helpful reference. We have expanded this discussion as follows:

"O/C tends to increase with oxidative aging (Decarlo et al., 2008) whereas H/C ranges from
increasing to decreasing with oxidative aging, depending on the types of reactions occurring
(Heald et al., 2010). Changes in O/C and H/C are also influenced by mixing of different air
masses and co-oxidation of different VOC precursors (Chen et al. 2015). Thus, tracking H/C
with aging may provide clues upon the types of reactions that may be occuring; however,
variable oxidation timescales can make inferences of this type difficult (Chen et al. 2015)."
In our analysis, we background-correct C, O, and H (creating ΔC, ΔO, and ΔH) and present the
ratios as ΔO:ΔC and ΔH:ΔC. The mixing of background OA into the plume should have no
direct impact on ΔO:ΔC and ΔH:ΔC (although there may be indirect impacts through changing
chemistry).

R1.8) L130: The authors note that the supporting info provides "more details on the instruments
used." I find this misleading. The information provided in the SI is extremely limited, hardly
greater than that provided in this paragraph. I suggest the authors provide in the SI some
discussion at least of instrumental uncertainties.

We agree that our SI is sparse on details of the BBOP instrumentation. Our coauthor Lawrence
Kleinman's current ACPD paper also on BBOP aerosol properties has a significant amount of
detail on the SP-AMS, the SP2, the FIMS, and trace gas instruments. We will refer the reader to
this text for those details. As well, we flushed out our discussion in the SI:

The Fast Integrated Mobility Spectrometer (FIMS) characterizes particle sizes based on
electrical mobility as in scanning mobility particle sizer (SMPS). Because FIMS measures
particles of different sizes simultaneously instead of sequentially as in traditional SMPS, it
provides aerosol size distribution with a much higher time resolution at 1 Hz (Wang et al.,
2017). The relative humidity of the aerosol sample was reduced to below ~25% using a Nafion
dryer before being introduced into the FIMS. Therefore, the measured size distributions
represented that of the dry aerosol particles. The particle number concentration integrated from
FIMS size distribution typically agrees with the CPC 3010 (Condensation Particle Counter)
measurement (Kleinman et al. 2020) within ~ 15% when size distribution suggests that particles
smaller than 10 nm contribute negligibly to the total number concentration. Thus, we estimate
the uncertainty in the FIMS number concentration to be ~15%. The uncertainty in measured
particle size is about 3% (Wang et al. 2017).
The Soot Particle Aerosol Mass Spectrometer (SP-AMS) is thoroughly detailed in
Kleinman et al. (2020). Although it was not directly characterized for uncertainties during the
BBOP campaign, we estimate uncertainties as follows. The AMS uncertainty is estimated following the methods in (Bahreini et al. 2009) (first equation of their supplemental information), leading to 37% uncertainty for organics. The laser vaporizer adds additional uncertainty up to 20%. Thus summing the uncertainties in quadrature leads to a 42% uncertainty in organics. The Soot Photometer (SP2) had an uncertainty of 20%.

CO measurement uncertainties are detailed in Kleinmen et al. (2020): the Off-Axis Integrated Cavity Output Spectroscopy was found to have an accuracy of 1-2%, and the precision at ambient backgrounds of 90 ppb was 0.5 ppbv RMS (using a 1 second averaging).

An SPN1 radiometer (Badosa et al. 2014; Long et al. 2010) provided total shortwave irradiance, with a shaded mask applied following (Badosa et al. 2014). The data was corrected for tilt up to 10 degrees of tilt, following (Long et al. 2010). For tilt greater than 10 degrees these values are set to "bad". Instrument uncertainties are detailed in (Badosa et al. 2014).

Badosa, Jordi, John Wood, Philippe Blanc, Charles N. Long, Laurent Vuilleumier, Dominique Demengel, and Martial Haeffelin. 2014. "Solar Irradiances Measured Using SPN1 Radiometers: Uncertainties and Clues for Development." *Atmospheric Measurement Techniques* 7: 4267–83.

Bahreini, R., Ervens, B., Middlebrook, a. M., Warneke, C., de Gouw, J. a., DeCarlo, P.F., Jimenez, J.L., Brock, C. a., Neuman, J. a., Ryerson, T.B., Stark, H., Atlas, E., Brioude, J., Fried, A., Holloway, J.S., Peischl, J., Richter, D., Walega, J., Weibring, P., Wollny, a. G., and Fehsenfeld, F.C. (2009). Organic aerosol formation in urban and industrial plumes near Houston and Dallas, Texas. J. Geophys. Res., 114:D00F16.

Kleinman, L. I., Sedlacek III, A. J., Adachi, K., Buseck, P. R., Collier, S., Dubey, M. K., Hodshire, A. L., Lewis, E., Onasch, T. B., Pierce, J. R., Shilling, J., Springston, S. R., Wang, J., Zhang, Q., Zhou, S., and Yokelson, R. J.: Rapid Evolution of Aerosol Particles and their Optical Properties Downwind of Wildfires in the Western U.S., Atmos. Chem. Phys. Discuss., https://doi.org/10.5194/acp-2020-239, in review, 2020.

Wang, J., Pikridas, M., Spielman, S. R., and Pinterich, T.: A fast integrated mobility spectrometer for rapid measurement of sub-micrometer aerosol size distribution, Part I: Design and model evaluation, J. Aerosol Sci., 108, 44-55, 10.1016/j.jaerosci.2017.02.012, 2017.

R1.9) L138: I suggest it be clarified how f60 and f44 are background corrected. Presumably this is not a straight difference, as the denominators ([OA]) differ. Is it, for example f60_corrected = (f60_plume*[OA]_plume – f60_bgd*[OA]_bgd)/[OA]_plume? If the authors used a straight difference, this must be justified as it does not seem appropriate to me. Similarly, more details on how the other intensive properties (O:C, H:C) are corrected are needed.

We calculated the background corrected f60 and f44 as follows (where $f = f_{60}$ or $f_{44}$):

$$\Delta f \ = \ \frac{(f_{in}*OA_{in}) - (f_{out}*OA_{out})}{\Delta OA} \qquad \text{Eq. R1}$$

Similar, the $\Delta O/\Delta C$ and $\Delta H/\Delta C$ are calculated through (where $X=$ O or H):

$$\frac{\Delta X}{\Delta C} \ = \ \frac{(X_{in\ plume} - X_{out\ of\ plume}\ )}{(C_{in\ plume} - C_{out\ of\ plume})} \qquad \text{Eq. R2}$$

We've added Eqs. R1-R2 as Eqs. 1 and 2 in the main text and have updated other equation
numbers and references.

R1.10) L140: It would be helpful if in Figs. S2-S6 and S7-S11 the authors would number each
plume so that the two can be related to each other. It would also help if the time-series were
shown as an additional panel with the spatial plots, again so comparisons can be made. I think
this is important because the authors discuss "plumes" here, but they do not discuss how it is, for
example, that in a given transect there can be multiple maxima in CO. Does this imply there are
two plumes? Or is this the same plume? What drives this behavior, and what might it indicate
about the evolution of the plumes? What does it mean to define a "centerline" of the plume if
there are clearly two distinct maxima on either side (see Fig. S3, for example).

We have included subplots for figures S2-S6 that show both the flight tracks colored by time in
minutes as well as the leg numbers as designated in the BBOP database (as designated by the
flight team). We've updated the x-axis of figures S7-S11 to be in minutes to allow for easier
comparisons between the two. We agree that the "centerline" is an imperfect metric and is a
limitation of this study. However, the centerlines have been determined using the most-
concentrated portion of the aerosol number concentration, which did tend to be more clear (see
e.g. Fig. S1). We added more text about the centerline, also following comments from R2.25:

"To estimate the physical age of the plume, we use the estimated fire center as well as the total
FIMS number concentration to determine an approximate centerline of the plume as the smoke
travels downwind (Figs. S1-S6). The centerline is subjectively placed to attempt to capture the
most-concentrated portion of the total number concentration for each plume pass, as we focus on
aerosol properties and their relations to dilution in this study. We use  mean wind speed and this
estimated centerline to get an estimated physical age for each transect. We did not propagate
uncertainty in fire location, wind speed, or centerline through to the physical age, which is a
limitation of this study."

We have also added the following text to the first paragraph of section 3 to discuss the potential
of multiple plumes (underlines for the new material):

"We have divided each transect into four regions: between the 5-15 (edge), 15-50 (intermediate,
outer), 50-90 (intermediate, inner), and 90-100 (core) percentile of ΔCO within each transect.

Fig. 1 shows the edge and core data, both averaged per transect, with Figs. S14-18 providing all four percentile bins for each flight. These percentile bins correspond with the thinnest to thickest portions of the plume, respectively, and if a fire has uniform emissions ratios across all regions and dilutes evenly downwind, these percentile bins would correspond to the edges, intermediate regions, and the core of the diluting plume. We use this terminology in this study but note that uneven emissions, mixing, and/or dilution lead to the percentile bins not corresponding physically to our defined regions in some cases. We note that some plumes show more than one maxima in CO concentrations within a given plume crossing, which implies that there may be more than one fire or fire front, and that these plumes from separate fires or fronts are not perfectly mixing. As well, in at least one of the fires (in flights '730a' and '730b'), the fuels vary between different sides of the fire, as discussed in Kleinman et al., (2020)."

R1.11) From Figs. S7-S11, it appears that the background [CO] varies from flight-to-flight. For example, in Fig. S7 the background is clearly lower than the 150 ppb threshold the authors have used, but in Fig. S9 it is barely sufficient. Why not define a flight-specific background [CO] based on the observations?

We agree that the background CO is variable from flight to flight. However, we performed a sensitivity analysis on the background CO cutoff (using a cutoff of 200 ppbv instead of 150 ppbv), shown in Fig. S20, and the results do not qualitatively change our conclusions. This is briefly discussed in lines 205-208, "*Figs. S13, S19-S21 show the same details as Fig. 2 but provide sensitivity tests to potential FIMS measurement artifacts (Fig. S13) and our assumed background CO and ΔCO percentile spacing (Figs. S19-S21). Although these figures show slight variability, the findings discussed below remain robust, and we constrain the rest of our discussion to the FIMS measurements, background and ΔCO percentile spacings used in Fig. 2.*"

R1.12) L156: The authors note that the instruments had various time lags, but it is not clear whether they were all adjusted to account for these varying time lags. This should be clarified. Also, it would be helpful if the authors clarified whether they really mean a "lag" but with a fast response time (i.e., two instruments both show sharp changes but are offset) or whether they are referring to some amount of smearing in which previous measurements affect the current measurement. From the FIMS discussion, it sounds as if they are actually talking about smearing (related to instrument response time) and not a lag.

The data was not time lag corrected, and we clarify this in the text now. Kleinman et al. (2020) provides further details on time lags--they did correct the data but note that "Time-shifts of 1-2 seconds are readily apparent as a degradation in correlation when comparing instruments. Maximizing correlations, however, does not accurately compensate for varying response time." From coauthor Kleinman's careful work and analyses, we believe that most of the instruments display only a time lag, but that the FIMS displays both a time lag and some smearing. Given
that analysis only using the first half of the FIMS data for each leg did not change our
conclusions (see the methods section, specifically *"To test if these lags impact our results, we*
*perform an additional analysis where we only consider the first half of each in-plume transect,*
*when concentrations are generally rising with time  (Figure S12-S13), and our main conclusions*
*are unaffected."*) We have clarified in the text that the FIMS had additional smearing.

Kleinman, L. I., Sedlacek III, A. J., Adachi, K., Buseck, P. R., Collier, S., Dubey, M. K.,
Hodshire, A. L., Lewis, E., Onasch, T. B., Pierce, J. R., Shilling, J., Springston, S. R., Wang, J.,
Zhang, Q., Zhou, S., and Yokelson, R. J.: Rapid Evolution of Aerosol Particles and their Optical
Properties Downwind of Wildfires in the Western U.S., Atmos. Chem. Phys. Discuss.,
https://doi.org/10.5194/acp-2020-239, in review, 2020.

R1.13) L165: Further details regarding how the FIMS data were used to establish the centerline
are needed. How were the number distributions used specifically? How were these determined
for different transects to give a single straight line? Also, is wind speed as measured by the
aircraft?

The centerline was subjectively determined to approximately capture the most-concentrated
portion of the total number concentration for each plume pass, as we are focused on aerosol
properties in this study (and their relation to concentration and dilution). We have added this text
to the main document and do include this as a limitation of the study in the original text (new
text underlined for clarity and along with some fixes to errors pointed out in R2):

"To estimate the physical age of the plume, we use the estimated fire center as well as the total
FIMS number concentration to determine an approximate centerline of the plume as the smoke
travels downwind (Figs. S1-S6). The centerline is subjectively placed to attempt to capture the
most-concentrated portion of the total number concentration for each plume pass, as we focus on
aerosol properties and their relations to dilution in this study. We use  mean wind speed and this
estimated centerline to get an estimated physical age for each transect. We did not propagate
uncertainty in fire location, wind speed, or centerline through to the physical age, which is a
limitation of this study."

R1.14) Fig. 1: The figure lacks error bars. Given the analysis, it would seem that precision-
based propagated uncertainties would be appropriate, as the authors seem interested more in
characterizing changes than they are absolute values. I suggest appropriate error bars are added.

We are quite hesitant to put forth a precision-based analysis. We are cautious to apply a known
precision under ambient conditions to the sometimes extremely concentrated conditions of
smoke plumes. For instance, our initial analyses included ozone measurements and UHSAS

particle size distribution measurements, but we had to remove both instruments due to unresolvable issues with interferences under plume conditions. The UHSAS became saturated-- this saturation level may be changing as both a function of particle size and concentration (as was discovered from careful analysis of a UHSAS during strong pollution events during an indoor campaign and seen again during a controlled burn study; Erin Boedicker [Colorado State University; Farmer group], personal communication). Another issue is that propagating uncertainties assumes that precision is equivalent in all of the measurements. We are using multiple instruments so this assumption breaks down, as many instruments define and calculate precision differently. This makes a true apples-to-apples comparison (which is needed for propagation of errors) tricky or impossible. As discussed in response to other comments by this and the other reviewer, we have weakened the language of our results throughout due to these uncertainties.

R1.15) L182: While it seems that the 5-15 percentile values are primarily found at the physical edges of the plumes shown in the supplemental, as often as not the 90-100 percentile values exhibit bimodal behavior across a transect, often occurring relatively close to the physical edge. From what is shown, I do not believe it is justified to say that the 90-100 percentile "core" corresponds to the physical "core" of the plume as observed. I strongly suggest the authors to define a quantitative metric to relate the percentiles to the spatial distribution. Perhaps a normalized distance from the centerline.

We agree that the 5-15 and 90-100 percentiles do not perfectly line up to the physical edge and core, and state in the original manuscript (lines 177-182): "*These percentile bins correspond with the thinnest to thickest portions of the plume, respectively, and if a fire has uniform emissions ratios across all regions and dilutes evenly downwind, these percentile bins would correspond to the edges, intermediate regions, and the core of the diluting plume. We use this terminology in this study but note that uneven emissions, mixing, and/or dilution lead to the percentile bins not corresponding physically to our defined regions in some cases. However, the lowest two $\Delta CO$ bins tend more towards the physical edges of the plume and the highest two tend more towards the physical center of the plume (Figs. S2-S6).*"

(We note that we have added more material to the above quoted section, following comment R1.10). We argue that our 5-15, 15-50, 50-90, and 90-100 $\Delta CO$ percentile bins are our quantitative metric and that due to variable mixing between different smoke plumes as well as variable plume widths, defining a spatial relationship is not necessarily particularly informative. We add the following reminder to the manuscript in sect 3.1:

"We use the simple 'edge' and 'core' terminology throughout the following discussion but note that the 5-15 and 90-100 $\Delta CO$ percentile bins do not necessarily correspond to the physical (spatial) edges and cores of each plume. They instead correspond to the most CO-dense and least CO-dense portions of the plume.

R1.16) L191: I suggest the authors be more precise in their claims. The normalized number
concentration in the "core" does not change with age, and at the edge the entirety of the change
is observed from the first transect to the second. And there is perhaps an increase in diameter
from the first transect to the next, but the diameter is constant (within variability) for all transects
further downwind. Also, the deltaO/C does not increase with aging. The authors indicate that the
delta_f44 changes with age, but it is not clear how this was determined. Was some sort of linear
fit done? Is this just the difference between the first point and the last? Visually, the points look
scattered about a flat line. Overall, for this discussion I think that the authors need to be more
specific and precise and quantitative. As currently written, it is not always clear how the authors
came to the conclusions that they did.

We agree that this flight shows weak trends for the majority of the metrics discussed, and that
information on trends is only gained once all of the flights have been pooled together. Figure 1's
primary purpose is to orient the reader to the different metrics and how they might look for a
flight. We have changed this paragraph to read:

"Figure 1 shows that for this specific plume, $\Delta OA/\Delta CO$ and $\Delta BC/\Delta CO$ vary little with age for
both the 5-15 and 90-100 percentile of $\Delta CO$ (p-values>0.5). A true Lagrangian flight with the
aircraft sampling the same portion of the plume and no measurement artifacts (e.g. coincidence
errors at high concentrations) would have a constant $\Delta BC/\Delta CO$ for each transect. This flight and
other flights studied here have slight variations in $\Delta BC/\Delta CO$ (Fig. 1; Figs. S14-S18), which may
be indicative of deviations from a Lagrangian flight path with temporal variations in emission
and/or measurement uncertainties. The remaining variables plotted also show some noise and
few clear trends, but it is apparent that the 5-15 and 90-100 percentiles do show a separation for
many of the individual metrics. In order to determine the existence or lack trends for these
metrics, we spend the remainder of this study examining each metric from all of the pseudo-
Lagrangian flights together."

R1.17) L203: I find it exceptionally difficult to understand exactly what the authors have done
with the Spearman rank-order correlation tests. The authors need to be much more specific. The
authors have one value for (e.g.,) initial plume OA mass but then have multiple values for the
deltaOA/deltaCO for each transect of a given plume. Then there are multiple plumes. How are
the data merged to allow comparison across all plumes? Physical age makes more sense, as (for
example) deltaOA/deltaCO can be regressed versus physical age for each plume. But, to me,
how the other parameters are used (OA initial and deltaOAinitial) is unclear. Are all the initial
OA values repeated for a given flight? Are the authors using only the initial values for the other
parameters to compare with initial OA?

We see that our original text here is confusing and misleading. We have attempted to clarify it.
We are using a single value for $\Delta OA_{initial}$ for each transect within a Lagrangian set of transects
which is obtained from the first transect of the set. If a flight has two Lagrangian sets of transects, there will be a different value of $\Delta OA_{initial}$ used for the two sets of transects, each again obtained from the first transect of each set. The original text may have been interpreted that we used $OA_{initial}$ but we did not--we have clarified that. We use the changing values of $\Delta OA/\Delta CO$,

$\Delta f_{60}$, $\Delta f_{44}$ $\Delta H/\Delta C$, and $\Delta O/\Delta C$ as they age downwind to compare with initial OA. We have updated this text (also following suggestions made in R1.20):

"Also included in Fig. 2 are the Spearman rank-order correlation tests (hereafter Spearman tests), which are tests for monotonicity. The Spearman tests show correlation coefficients for each flight set (Table S1) with the initial $\Delta OA$ of a flight set ($\Delta OA_{initial}$) against $\Delta OA/\Delta CO$,

$\Delta f_{60}$, $\Delta f_{44}$ $\Delta H/\Delta C$, and $\Delta O/\Delta C$ as each variable ages downwind. We also include Spearman tests for the calculated physical age of the smoke for each flight set against these same variables. The

R values are labeled $R_{\Delta OA,initial}$ and $R_{age}$, respectively, in Fig. 2.  For the correlations with

$\Delta OA_{initial}$, all transects in a given Lagrangian set of transects have the same $\Delta OA_{initial}$ value; for flights with two Lagrangian set of transects, each set has its own $\Delta OA_{initial}$ value. Correlating to

$\Delta OA_{initial}$ provides an estimate of how the plume aerosol concentrations at the time of the initial transect impact plume aging (aging both before and after this initial transect)."

R1.18) L213: What does it mean for something to "evaporate off through heterogeneous aging?"

Things can evaporate, or they can be heterogeneously oxidized. These are distinct processes.

We agree that the language here is misleading, and have updated the text to read:

" $\Delta f_{60}$ generally decreases with plume age ($R_{age}$ = -0.26; a weak correlation), consistent with the hypotheses that $\Delta f_{60}$ may be evaporating because of dilution, undergoing heterogeneous oxidation to new forms that do not appear at m/z 60, and/or having a decreasing fractional contribution due to condensation of other compounds."

R1.19) L210: The authors note that the changes in deltaOA/deltaCO with aging are small. A

recent review by the authors (Hodshire et al., 2019) indicates a variety of reasons for such behavior. Another recent paper (Lim et al., 2019) introduces another potential reason for this behavior, specifically potential biases in the measurement of OA as the particle composition evolves. Have the authors considered this?

We agree that variable collection efficiency and related measurement artifacts could in theory bias OA measurements. We realized that we did not include in the original manuscript the characterized collection efficiencies (CE) of the SP-AMS, found to have two different efficiencies for when the laser was on (CE=0.76) or off (CE=0.5) and we include those details in the text now. We did not characterize any changes in efficiency with aging. This is an on-going topic of debate within the AMS community (and is addressed within the SI of the abovementioned paper from our group, Hodshire et al. 2019), and we briefly address it as a limitation of this study. We have included these details in the methods section:

"It [the SP-AMS] was characterized to have a collection efficiency of 0.5 when the laser was off
and 0.76 when the laser was on during the BBOP campaign, and these corrections have been
applied to the data. We do not attempt to characterize whether the collection efficiency of the
SP-AMS changes as the aerosol ages. This may be a limitation of this study, as collection
efficiency has been recently observed to decrease with aging within a laboratory study of
biomass burning (Lim et al. 2019). However, no consistent evidence of changing collection
efficiencies in field studies exist yet."

R1.20) With reporting the Spearman's correlation coefficient I suggest the authors use
consistent language that links to typical interpretation of the level of significance (that a
relationship is monotonic). For example, a value of -0.25 (as determined for f60) might be
considered "weak" while a value of 0.54 (for f44) is "moderate." Also, the authors might note
when introducing the Spearman's test that it is a test for monotonicity.

Thank you for these suggestions. We now note in the text that the Spearman tests are a test for
monotonicity when we first mention it in the text, and have added the following definitions that
we use throughout the text each time we discuss an R value (and we also have updated our
language for $R^2$ to reflect these categories as well as emphasizing that $R^2$ is explaining a given
fraction of the variance):

"We define the following categories of correlation for the absolute value of R: 0.0-0.19 is 'very
weak', 0.2-0.39 is 'weak', 0.4-0.59 is 'moderate', 0.6-0.79 is 'strong', and 0.8-1.0 is 'very
strong' (Evans, 1996)."

Evans, J. D. (1996). Straightforward statistics for the behavioral sciences. Thomson Brooks/Cole
Publishing Co.

R1.21) There appears to be a good deal of flight-to-flight variability in behavior, from Fig. 2.
This raises a question of how much of the inferred behavior (from the Spearman's test) derives
from fairly strong changes in one flight. The authors might consider testing the sensitivity to
their analysis by determining Spearman's coefficients when systematically leaving out individual
flights or transects one at a time. This would give a broader sense of the robustness of the results,
given the notable scatter.

We have performed the Spearman's test for $R_{age}$ and $R_{\Delta OA, initial}$ for all metrics of Figure 2 leaving
one flight out at a time. The results are summarized in Table S2. We add the following text when
we first introduce the R values:

"As individual flights show scatter in the metrics of Fig. 2 (Figs. 1, Figs. S14-S18), we also
include $R_{\Delta OA,initial}$ and $R_{age}$ for each metric of Fig. 2 systematically sequentially removing one flight from the statistical analysis. These results are summarized in Table S2. In general,
removing single flights does not change our conclusions, particularly when correlations are
moderate or stronger."

We provide the range of these results within the text as each metric is discussed.

R1.22) L217: Nitpicky, but compounds do not "contain f44." Certain compounds fragment in
such a way that they show up at m/z 44 in the AMS. But overall this sentence is a run on with a
second half that does not logically follow from the first. The sentence starts by talking about a
balance between condensation and evaporation but shifts abruptly to note something about
heterogeneous oxidation or particle-phase reactions. I suggest the authors clarify the point they
are aiming to make here.

This is a reasonable point and we have updated the text to here to read (including updates as
suggested by reviewer 2's comment R2.43):

"$\Delta f_{60}$ generally decreases with plume age ($R_{age}$ = -0.26), consistent with the hypotheses that $\Delta f_{60}$
may be evaporating because of dilution, undergoing heterogeneous oxidation, and/or having a
decreasing fractional contribution due to condensation of other compounds.. In contrast, $\Delta f_{44}$
generally increases with age ($R_{age}$ = +0.5) for all plumes with available data. It appears for the
plumes in this study that although there is little change in $\Delta OA/\Delta CO$, loss of compounds that
contain $f_{60}$ fragments (as captured by the SP-AMS) is roughly balanced by condensation of
more-oxidized compounds, including those that contain compounds with $f_{44}$ fragments, such as
carboxylic acids. This observation suggests the possibility of heterogeneous or particle-phase
oxidation that would alter the balance of $\Delta f_{60}$ and $\Delta f_{44}$."

R1.23) L219: The authors note that deltaOA/deltaCO does not change much. This would be
consistent with the little mass loss that the authors note from heterogeneous oxidation here,
correct? Are the authors aiming to make a point more specifically about the efficiency with
which heterogeneous oxidation might degrade the f60 signal and not about mass loss? I find it
unclear.

We are trying to note that heterogeneous chemistry is relatively slow (for near-field aging) and
shouldn't significantly contribute to evaporative or compositional changes. We have added text
to emphasize that point more clearly:

"However, estimates of heterogeneous mass losses indicate that after three hours of aging for a
range of OH concentrations and reactive uptake coefficients, over 90% of aerosol mass is anticipated to remain, indicating that heterogeneous loss has limited effect on aerosol composition or mass (Fig. S23; see SI text S2 for more details on the calculation). Hence, the evaporation of $f_{60}$ being balanced by gas-phase production of $f_{44}$ may be the more likely pathway."

 R1.24) Laboratory observations (Cubison et al., 2011;Hennigan et al., 2011;Hodshire et al., 2019;McClure et al., 2020) have demonstrated that the f60 and f44 of freshly emitted particles vary over large ranges dependent on the fuel type and specific burn condition. Is it not possible that the differences in deltaf60 and deltaf44 between flights result from intrinsic differences in the emitted particle properties? The authors seem to discount this without explicit justification when they state that their interpretation assumes that "emitted deltaf60 and deltaf44 do not correlate with deltaOAinitial." Might there not be an initial correlation, as this might indicate some difference in the burn conditions or the particular fuel mix? I can certainly believe that "evaporation and/or chemistry likely occurred before the time of" the first measurements, however it is not clear to me that the observations as presented here demonstrate this conclusively. Also, given that different sources produce particles that have different initial f60 and f44, would they be expected to exhibit the same deltaf60 and deltaf44 even if initial OA and dilution were identical? Is there evidence that this is expected?

The reviewer makes reasonable points here and we agree that these are alternative hypotheses that should be explicitly discussed in the manuscript. Reviewer 2 made similar comments in R2.47. Unfortunately, lacking direct measurements of the emissions, we cannot explore this hypothesis in any detail. And we do find it compelling that less-dense plumes do show higher f44/lower f60 than more-dense plumes, which supports our hypothesis of aging prior to the transect. We have added the following text to Sect. 3.1:

"We note that each fire may emit particles with variable initial $f_{44}$ and $f_{60}$ values, as has been observed in laboratory studies (Hennigan et al. 2011; Cubison et al. 2011; McClure et al. 2020), which adds to scatter within the data. It is possible that variability in f44 and f60 may also contribute to the observed correlations with $\Delta OA_{intial}$; however, this would require that higher $f_{44}$ emissions are correlated with lower emissions rates and/or faster dilution rates (and visa versa for $f_{60}$). Lacking direct emissions measurements, this hypothesis cannot be further explored in this work."

To the reviewer's last query ("Also, given that different sources produce particles that have different initial f60 and f44, would they be expected to exhibit the same deltaf60 and deltaf44 even if initial OA and dilution were identical? Is there evidence that this is expected?"), we would not expect the same $\Delta f_{44}$ and $\Delta f_{60}$ under those circumstances and thus variability from emissions likely contributes to the noise of our fit parameters. We do include a brief discussion on this in the text in Sect. 3.1 within the discussion of our fit parameters (with new minor edits
addressing comments from reviewer 2):

"The scatter is likely due to variability in emissions due to source fuel or combustion conditions,
instrument noise and responses under the large concentration ranges encountered in these smoke
plumes, inhomogeneous mixing within the plume, variability in background concentrations not
captured by our background correction method, inaccurate characterizations of physical age due
to variable wind speed, and/or deviations from a true Lagrangian flight path."

R1.25) L243: I disagree with the authors interpretation of the van Krevelen diagram here. The
authors interpret this in a process based way related to chemistry. However, this does not account
for the fact that this is, likely, ultimately a mixing experiment wherein primary OA is being
increasingly mixed with secondary OA. This cannot be interpreted in terms of functional group
addition. Additionally, it is not clear that a plot of deltaO/deltaC vs deltaH/deltaC should behave
in the same way as a plot of O/C vs H/C. The authors must demonstrate the equivalency of these.

We think that the reviewer has interpreted our work to mean that we have calculated delta(H/C)
and delta(O/C) (we did not calculate this), rather than delta(H)/delta(C) and delta(O)/delta(C)
(which is the calculation we did do). We hope that our response to the earlier reviewer comment
R1.9 clarifies this matter. We remind the reader in the text of this here, (underlines for new
additions):

A Van Krevelen diagram of $\Delta H/\Delta C$ versus $\Delta O/\Delta C$ (Fig. S27) indicates that oxygenation
reactions or a combination of oxygenation and hydration reactions are likely dominant (Heald et
al., 2010) (recalling that $\Delta H/\Delta C$ and $\Delta O/\Delta C$ are calculated by background-correcting the
individual elements before ratioing; Eq. 1)
It is true that any non-linear changes in chemistry and composition will mean that our
delta(H)/delta(C) and delta(O)/delta(C) method will not perfectly isolate the elemental factors
from smoke, and we add this disclaimer in the methods:
"We note that any non-linear changes in chemistry and composition between the plume and
background will not perfectly isolate the elemental factors in smoke."

R1.26) deltaO/deltaC ratios: I am somewhat surprised that these values are positive. O:C ratios
of fresh biomass burning tend to be around 0.3-0.4 whereas O:C of background OA are typically
large. (The same is true for f44.) The authors should comment on the very fact that their
deltaO/deltaC values are positive.

We think that the reviewer has interpreted our work to mean that we have calculated delta(H/C)
and delta(O/C) (we did not calculate this), rather than delta(H)/delta(C) and delta(O)/delta(C)

(which is the calculation we did do). delta(H)/delta(C) and delta(O)/delta(C) represent estimates
of the H:C and O:C of the smoke OA, which cannot physically be negative; and it would be
highly unlikely that delta(H), delta(C), and delta(O) are negative as this would require the
background concentration of these elements to be higher than the plume concentrations We hope
that our response to the earlier reviewer comment R1.9 clarifies this matter and explains the
positive values.

R1.27) Eqn. 2: First, what is the justification for this functional form? Is there some other form
that would better explain the data? Second, in terms of utility, is it really most useful to predict
the delta values, as these will depend explicitly on the background, which may vary between
locations? Do the authors expect these relationships will prove robust and applicable to other
regions? Would these be appropriate at night as well as during the day? The authors have not
been able to distinguish between dilution-driven changes and oxidation-driven changes, so there
may be distinct day/night differences? When would they expect them applicable? How could
these parameters assist specifically in biomass burning models? Presumably such models would
aim to be processed based, differentiating between oxidation and dilution.

Reviewer 2 had similar questions in comment R2.56). We do not agree with the comment about
delta values here. The delta values mean that the background has been subtracted off in an
attempt to isolate the smoke contributions. Hence, in the absence of non-linear interactions
between the smoke and background species, the delta values do not depend on the background.
The non-delta values (the smoke+background values) much more explicitly depend on the
background.

We do agree that it's as yet unclear whether these fits are appropriate for other regions of the
world as well as day/night differences. We tried a large number of mathematical fits and these
equations (Eqs. 2-3 in the original text; Eqs. 4-5 in the updated text) performed the best. They do
not have a direct physical meaning. The parameters would need significantly more testing to be
applicable for models, and we have added the following text to address these comments:

"Eqs. 4-5 performed the best out of the mathematical fits that we tested. These equations do not
have a direct physical interpretation but may be used as a starting point for modeling studies as
well as for constructing a more physically based fit. There may be another variable not available
to us in the BBOP measurements that can improve these mathematical fits, such as photolysis
rates. We do not know whether these fits may well-represent fires in other regions around the
world, given variability in fuels and burn conditions. We also do not know how these fits will
perform under nighttime conditions, as our fits were made during daytime conditions with
different chemistry than would happen at night. We encourage these fits to be tested out with
further data sets and modeling. These equations are a first step towards parameterizations
appropriate for regional and global modeling and need extensive testing to separate influences of
oxidation versus dilution-driven evaporation."

R1.28) When the authors report the Pearson's coefficients, are these constrained to go through
the origin? The authors show only the 1-1 lines, but visually it seems that any linear fit to the
calculated vs. observed relationship will have a non-zero intercept unless constrained. In this
context, having a good r^2 value is simply an indication of a linear relationship but it is not an
indication of the goodness of the calculated vs. observed. Instead, the authors would need to
provide some metric such as normalized mean bias. As presented, I am not convinced that the r^2
values are particularly meaningful.

We do not constrain the Pearson's coefficients to go through the origin. We have now calculated
the normalized mean bias (NMB) and normalized mean error (NME), as the normalized mean
bias is likely to be small given that we're minimizing the linear fit. We include the NMB and
NME values in our Figures 3 and S28-29. We have updated figures and figure captions
accordingly. We add the following sections of text to Sect. 3.1 and 3.2:

1. (Section 3.1) "We do not constrain our fits to go through the origin. To provide further metrics
of goodness-of-fit, we also include the normalized mean bias (NMB) and normalized mean error
(NME) in percent for each metric of Fig. 3. The NMB values are very close to zero (which is
anticipated as linear fits seek to minimize the sum of squared residuals).  The NME is more
variable, at 18.8% for $\Delta f_{60}$, 14.9% for $\Delta f_{44}$, and 10.4% for $\Delta O/\Delta C$."
2. (Section 3.1) "Other functional fits were explored, with
$$ln(\Delta X) = a\ ln(\Delta OA_{initial}) + b\ ln(Physical\ age)\ + c \qquad\qquad \text{Eq. 5}$$
(Fig. S28 and Table S4 for the fit coefficients) and $\Delta N_{initial}$ in the place of $\Delta OA_{initial}$ in Eq. 42
(Fig. S29 and Table S5 for the fit coefficients) providing similar correlation values and NMB and
NME values for $\Delta f_{60}$, and $\Delta f_{44}$, and $\Delta O/\Delta C$."
3. (Section 3.2) We also perform the functional fit analysis following Sect. 3.1 (Eq. 4; where X is
$D_p$ in this case). The fit can also weakly predict greater than 30 percent of the variance in $D_p$
($R_p^2$ and $R_s^{2s}$ of 0.36 and 0.31 and NME of 5.6%; Fig. 3d) but does not well-predict $\Delta N_{40\text{-}300}$
$_{nm}/\Delta CO$ (not shown). We show the functional fit for $D_p$ for the alternative fit equation (Eq. 5) in
Fig. S28 and Table S4. We also show the functional fits for $D_p$ for Eq. 4 with $\Delta N_{initial}$ in place of
$\Delta OA_{initial}$ in Fig. 29 and Table S5.

R1.29) L263: It is not clear to me what the authors are getting at when they state that aged
deltaf60 and deltaf44 show scatter, limiting the predictive skill of measurements available from
BBOP. They had just discussed how there are "moderate goodness of fits." It seems now that they are contradicting themselves. Or perhaps they are just providing more context for what "moderate" means.

We have updated our language when discussing the correlation metrics to be consistent throughout, following comment R1.20). Reviewer 2 had similar comments in R2.57), and we provide our response to that comment here:

We were referring to the aging values of $\Delta$f60 and $\Delta$f44, we were not careful in our language here though. "Limiting the predictive skill" was perhaps not the best phrase to use--we are trying to argue that the scatter in the measurement data is likely contributing to the limited predictive power of our current mathematical fits. We note that the p-values for these fits for $\Delta$f60 and $\Delta$f44 (as well as the other variables in Fig. 3, mean Dp $\Delta O/\Delta C$) and are both less than 0.01 and we argue that our fits provide valuable information on how physical age and a metric for plume size (here, initial OA at the time of the first measurement) impact $\Delta$f60 and $\Delta$f44. We now note in the text that the p-values are <0.01 for all fits and we have updated this section to read:

"The aging values of $\Delta f_{60}$ , $\Delta f_{44}$ , and $\Delta O/\Delta C$ show scatter (Figs. S14-18), which likely contributes to the limited predictive power of our mathematical fits. The scatter is likely due to variability in emissions due to source fuel or combustion conditions, instrument noise and responses under the large concentration ranges encountered in these smoke plumes, inhomogeneous mixing within the plume, variability in background concentrations not captured by our background correction method, inaccurate characterizations of physical age due to variable wind speed, and/or deviations from a true Lagrangian flight path. Eqs. 4-5 performed the best out of the mathematical fits that we tested. These equations do not have a direct physical interpretation but may be used as a starting point for modeling studies as well as for constructing a more physically based fit. There may be another variable not available to us in the BBOP measurements that can improve these mathematical fits, such as photolysis rates. We do not know whether these fits may well-represent fires in other regions around the world, given variability in fuels and burn conditions. We also do not know how these fits will perform under nighttime conditions, as our fits were made during daytime conditions with different chemistry than would happen at night. We encourage these fits to be tested out with further data sets and modeling. These equations are a first step towards parameterizations appropriate for regional and global modeling and need extensive testing to separate influences of oxidation versus dilution-driven evaporation.."

R1.30) L273: While the authors state here that highest initial deltaOA generally has the lowest normalized number concentrations, this seems to contradict their near zero Spearman's coefficient reported in Fig. 2. In fact, the authors state this two lines later. This needs to be revised. Either there is a correlation or there is not.

This is a good point--we have omitted this statement as it is not consistent with the observations.
Instead we state:

"Although we would anticipate that plume regions with higher initial $\Delta$OA would have lower
normalized number concentrations due to coagulation, a few dense cores have normalized
number concentrations comparable or higher than the thinner edges, leading to no correlation
with $\Delta$OA$_{initial}$."

R1.31) L276: Is variability in number emissions really "noise?" It seems like an inherent
feature.

We have changed "noise" to "unexplained variability" in the text.

R1.32) L278: Does the particle size really increase for "all" plumes, or does it statistically
increase when considered across all plumes? There seem to be some lines in the graph that are
basically flat when considered individually; thus, I am not certain that the "all" applies.

This is a good point. We have deleted the 'all' reference and have modified the text to read:

"The mean particle size between 40-262 nm, $D_p$ (Eq. 31), is shown to statistically increase with
aging when considered across the BBOP dataset…"

R1.33) L280: As mentioned above, have the authors considered other potential artifacts in their
deltaOA/deltaCO that might lead to this parameter remaining flat while the apparent particle size
increases? I suggest this be discussed in the context of the authors' conclusion that coagulation
drives the size change.

We agree that this caveat is appropriate to discuss here. We have added the following
parenthetical remark:

"(We acknowledge that $\Delta$OA/$\Delta$CO may be impacted by measurement artifacts as discussed in
Sect. 2. For instance, if the collection efficiency of the AMS is actually decreasing with age, then
$\Delta$OA/$\Delta$CO would be increasing and the increases in mean diameter will be due to SOA
condensation as well as coagulation.)"

R1.34) L283: The authors have been assuming that it is acceptable to use as an "initial" OA and
particle concentration the value measured in the closest transect for each flight. Given this
assumption, it is unclear why the authors now indicate it is essentially inappropriate to estimate
an initial particle diameter from the closest transect to use for comparison with the model of
Sakamoto et al. (2016). If the assumption is poor for one variable how is it justified that it is
okay for two other variables?

The reviewer has a good point that our logic seems inconsistent here. We add the following text
when we first introduce the concept of ΔOA$_{initial}$ in Sect. 3.1:

We note that ΔOA$_{initial}$ does not actually represent the true initial emitted OA from each fire, but
instead serves as a proxy for the general fire size, intensity, and emission rate (as presumably
larger, more intensely burning fires will have larger mass fluxes than smaller, less intensely
burning fires). Thus, ΔOA$_{initial}$ and other "initial" metrics referred to in this study are not to be
taken as emission values, and direct comparison to studies with direct emissions values is not
appropriate, as dilution and chemistry may occur before the initial flight transect, which we
discuss further below.

We also modify the specifically mentioned section:

"Sakamoto et al. (2016) provide fit equations for modeled $D_p$ as a function of age, but they
include a known initial $D_p$ at the time of emission in their parameterization (rather than 15
minutes or greater, as available to us in this study)" (underline added to point out new text)

R1.35) Equation 2: What units must the time have?

Good call--for the fit coefficients, time is in hours. We have now included this when introducing
the fit equation.

R1.36) L290: Nucleation is generally more favorable when existing particle surface area is
smaller, as the condensation sink is reduced. Might this also be an explanation for the greater
incidence of nucleation near plume edges?

Yes absolutely--we have added this possibility to the discussion.

"As well, nucleation is more favorable when the total condensation sink is lower (e.g. reduced
particle surface area) (Dal Maso et al. 2002), which may occur for outer portions of plumes with
little aerosol loading."

R1.37) L294: The authors note that the nucleation mode "appears to be coagulating or
evaporating away as the plumes travel downwind." It would be useful if they show this explicitly
in some way. Which figures should the reader look at specifically and which intersects? I find
this overall too vague and suggest that it needs to be made more explicit.

We have examined this statement and Figs. S7-S11 (number size distribution plots) and upon
further consideration do not think that it's strictly apparent what is happening to the smallest
particles downwind--quite often the nucleation mode appears to be persistent even at final transects. We have removed the statement and have moved the first half of this sentence to
earlier in the paragraph (underlines to emphasis text that has been moved into this sentence):

Nucleation-mode particles appear to be approximately one order of magnitude less concentrated
than the larger particles, and primarily occur in the outer portion of plumes, although one day did
show nucleation-mode particles within the core of the plume (Fig. S11).

R1.38) L303: Again, does "thicker" here mean "more concentrated"? Thickness, which I would
interpret to mean some spatial thickness, is not discussed in this paper as best I can tell.
Regardless, the authors cannot conclude that deltaN/deltaCO is lower for "thicker" plumes since
their Spearman's coefficient is essentially zero.

The reviewer is correct--following similar comments above, we delete this statement.

R1.39) L308: Again, how can the authors rule out differences in the initial conditions that are
independent of physical or chemical aging? This seems to be an underlying assumption
throughout this entire study, but I do not find that the authors have really justified this
assumption. Given how central it is to everything, I strongly suggest that an explicit discussion
must be included wherein the authors review the evidence for and against their assumption.

We have added more text and qualifiers to section 3 addressing this issue, following comments
R1.24 and R2.47. We add the following text to this discussion:

"We were unable to quantify the impact on potential interfire variability in the emission values
of the metrics studied here (such as variable $f_{60}$ and $f_{44}$).  We anticipate that being able to capture
this additional source of variability may lead to stronger fits and correlation."

And

"We also suggest further refinement of our fit equations, as further variables (such as photolysis
rates) and better quantification of interfire variability (such as variable emission rates) are
anticipated to improve these fits."

Minor:

R1.40) L47: It might be more accurate to say that the smoke plumes dilute through entrainment
of background air rather than that they dilute and entrain background air.

Thank you--this is similar to comment R2.14) and we have clarified this sentence, addressing
both reviewer comments:

"Dilution through entrainment of regional background air can cause vapors and particles emitted
from fires to rapidly evolve as smoke travels downwind"

 **Review 2**
*Summary*:
This manuscript uses airborne data of wildfire smoke plumes, measured as pseudo-lagrangian
transects of the plumes during the 2013 BBOP field campaign. Physical ages of the plumes
ranged from approximately 15 minutes to 2-4 hours.
The authors analyze the oxidation state (through f44, f60, O/C, and H/C) as well as mean particle
diameter and the OA/CO emission ratio of aerosol in terms of physical plume age and the
aerosol's proximity to the plume core. They demonstrate enhanced chemical aging/oxidation at
the edges of plumes that they argue is related to enhanced photolysis in more dilute BBOA-
containing air.
Only a couple studies have discussed the effects of chemical aging in terms of plume thickness
and edge-to-core position. This is a very informative and fascinating approach and is a great use
of archived data BBOP data to build upon previous modeling research. The paper is well cited
and the figures are generally aesthetically pleasing. Please don't be dismayed by the criticism to
follow as I tend to focus on the things that need to be fixed. There are a lot of good observations
and analysis in this paper which I don't, but maybe should, highlight.
I believe that many of the conclusions are likely true, however the way the data was analyzed
does not always support this and I have made quite a few comments regarding this. In my
opinion, a focus should be made on comparisons within transect sets regarding how things
evolve with physical age and generalizations of plume cores vs plume edges instead of on bulk
regressions (Spearman's correlations) which are not particularly convincing (either low R-values
or R-values reflective of outlier data). Additionally, there seems to be a lot of contradicting
statements made in interpreting the results. This is potentially a very good and interesting paper
relevant to the subject areas of ACP and eventually should be published, but obviously will
require significant edits.
*General Comments*:
R2.1) Figures are aesthetically pleasing but could use some minor changes.
We have followed both reviewers' specific suggestions in ensuing comments to the best of our
abilities and scientific agreement.
R2.2) Format of citations need to be fixed.
We agree that a number of our in-text citations came through poorly. We apologize and have
fixed these to the best of our abilities.

R2.3) There are a lot of typos and issues with word choice which will need to be fixed before
final publication.

We have responded to specific comments in both reviews and have done a thorough check of our
document before resubmission.

R2.4) I am curious, how wide were the plumes and how long did it take to fly through them? It
seems like you explored whether instrument lags affected your results, but during a
transect did the physical age of the leading the plume edge vary significantly from the
edge when you left the plume?

The plumes are approximately 5-50 km wide (using the Haversine method; this can be observed
in Figs. S2-S6). We now note this in the methods section:

"The plumes spanned from approximately 5-50 km wide (Figs. S2-6)."

If the flights were perfectly Lagrangian, the physical age would be the same from leading plume
edge to trailing plume edge. The plane was travelling at 100 m s$^{-1}$ on average, and thus took ~50-
500 s (0.8 to 8.3 minutes) to cross, and general uncertainty in physical age is larger than this. We
note this at the end of the section:

"We use the mean wind speed and this estimated centerline to calculate an estimated physical
age for each transect, and this physical age is assumed to be constant across the transect, as
plume crossings took between 50-500 seconds."

R2.5) I think you can better clarify how you estimate physical age. In the supplementary files,
the "core" trajectory is a straight line, presumably because you use a single wind speed
and direction, but the core of the transect frequently does not lie on that line. Could this
be improved with Hysplit/WRF models? Would that help the core of the transect fall
along the dashed line?

There were a few other comments on our physical age estimate (see R1.10, R1.13, and R2.25).
We have modified the text to read:

"To estimate the physical age of the plume, we use the estimated fire center as well as the total
FIMS number concentration to determine an approximate centerline of the plume as the smoke
travels downwind (Figs. S1-S6). The centerline is subjectively placed to attempt to capture the most-concentrated portion of the total number concentration for each plume pass, as we focus on
aerosol properties and their relations to dilution in this study."

We agree that our approximate center-line is not perfect. However, the resolution and uncertainty
of models like Hysplit or WRF are great enough that we do not have confidence that they would
perform any better as the model/reanalysis meteorology may have errors.

R2.6) Data are broken down into physical age and further into fringe-vs-core (such as shown in
Figure 1). These data-points represent a range of data subsample in time and space and
therefore should include error bars representing the variance in data represented by each
data point as well as the measurement uncertainty.

Reviewer 1 had similar comments in R1.14) for figure 1. We provide our response to them here
and argue that these comments are appropriate for figure 2 as well.

We are quite hesitant to put forth a precision-based analysis. We are cautious to apply a known
precision under ambient conditions to the sometimes extremely concentrated conditions of
smoke plumes. For instance, our initial analyses included ozone measurements and UHSAS
particle size distribution measurements, but we had to remove both instruments due to
unresolvable issues with interferences under plume conditions. The UHSAS became saturated--
this saturation level may be changing as both a function of particle size and concentration (as
was discovered from careful analysis of a UHSAS during strong pollution events during an
indoor campaign and seen again during a controlled burn study; Erin Boedicker [Colorado State
University; Farmer group], personal communication). Another issue is that propagating
uncertainties assumes that precision is equivalent in all of the measurements. We are using
multiple instruments so this assumption breaks down, as many instruments define and calculate
precision differently. This makes a true apples-to-apples comparison (which is needed for
propagation of errors) tricky or impossible. As discussed in response to other comments by this
and the other reviewer, we have weakened the language of our results throughout due to these
uncertainties.

R2.7) Df60 and Df44 are known to vary in primary emissions, even in laboratory experiments
where nascent soot can be analyzed (i.e. not after 10+ minutes of aging). However, a key
assumption in many of the conclusions seems to be that all primary BBOA has the same
initial Df60 and Df44. This is a problem when the authors try to support their conclusions.

Reviewer 1 had similar concerns in comment R1.24. We did not intend to make that assumption,
but it is possible that a reading of our manuscript gives the impression that we implicitly are
making that assumption. We do not expect the same $\Delta f_{44}$ and $\Delta f_{60}$ for each fire, and thus variability from emissions likely contributes to the unexplained variability of our fit parameters.
We do include two more brief discussions on this in the text in Sect. 3.1 within the discussion of
our fit parameters:
"We note that each fire may emit particles with variable initial $f_{44}$ and $f_{60}$ values, as has been
observed in laboratory studies (Hennigan et al. 2011; Cubison et al. 2011; McClure et al. 2020),
which adds to scatter within the data. It is possible that variability in f44 and f60 may also
contribute to the observed correlations with $\Delta OA_{intial}$; however, this would require that higher $f_{44}$
emissions are correlated with lower emissions rates and/or faster dilution rates (and visa versa for
$f_{60}$). Lacking direct emissions measurements, this hypothesis cannot be further explored in this
work."(this comment and R1.24)
"The scatter is likely due to variability in emissions due to source fuel or combustion conditions,
instrument noise and responses under the large concentration ranges encountered in these smoke
plumes, inhomogeneous mixing within the plume, variability in background concentrations not
captured by our background correction method, inaccurate characterizations of physical age due
to variable wind speed, and/or deviations from a true Lagrangian flight path." (this comment and
R1.24)
R2.8) The use of Spearman's rank-correlation is fine as you may not expect linearly
increasing/decreasing values with physical (or even chemical) age. But it needs to be
clearly stated that this is a test of monotonically increasing/decreasing values, which does
not give the same predictive interpretations as a Pearson's correlation.
Interpretation of Spearman's correlation coefficients and the strength of these
coefficients, in many cases, do not support the interpretations presented in this work. Part
of this is because the authors chose to combine all data from all flights together for the
regressions. This means that data representing older physical age of a plume with high
initial concentrations is mixed together with data representing young physical age but low
concentrations. The result is that there is not a strong relationship between these
parameters (e.g. DN/DCO) and physical age (or DOAinitial). If these transects were
normalized in some other way, maybe these statements may be more supportive of the
Conclusions.
We now note in the text that the Spearman tests are a test for monotonicity when we first
mention it in the text. We agree that mixing data in the fashion described may limit our statistical
analysis. However, the fit equations and results of Figure 3 do get at the combined effects of
age/concentration. Given that those fits show initial promise and that the results of Figure 1 do
show some moderate trends, we argue that there is value in our methods. The reviewer asks more
specific questions regarding normalization in comment R2.35, and we refer the reviewer to our
response there for further details.

R2.9) The supplementary text provides very little additional information. There seems to be
some confusion regarding methodology which could be explained in more detail here. I
would suggest a cartoon of a flight path showing how you chose your background for a
Transect.

We agree that the original SI was too sparse. We have expanded the SI section to include more
information about the campaign instrumentation, following reviewer 1's comment R1.8, and we
refer reviewer 2 to comment R1.8. We have included the locations of each flight's background
(lowest 10% of CO) in Figures S2-S6.

R2.10) Were all supplementary sections/figures referenced in the text? I lost count.

We verified before submission that all SI figures and text were referenced in the text; we have
re-verified before our current re-submission.

*Specific Comments*:

R2.11) L30: Be more specific about what you mean by "smoke concentrations… aging
markers,number, diameter."

We have updated the text to read:

"Here, we use observational data from the BBOP field campaign and show that initial smoke
organic aerosol mass concentrations can help predict changes in smoke aerosol aging markers,
number concentration, and mean diameter between 40-262 nm."

R2.12) L34-35: You state that it is not quantifiable how diluted a plume is when first measured;
does this contradict the next statement that (hence) the initially measured (number?)
concentration is a proxy for dilution?

We agree that this text is confusing and have clarified it:

"However, the extent to which dilution has occurred prior to the first observation is not a directly
measurable quantity. Hence, initial observed plume concentrations can serve as a rough indicator
of the extent of dilution prior to the first measurement, which impacts photochemistry and
aerosol evaporation."

R2.13) L37: Do you mean "increases in oxidative tracers" or that the oxidation-state of OA at
the edges was higher?
The latter--we've clarified the text (and split the original long sentence into two):
"We further find that on the edges, the oxidation state of organic aerosol has increased and has
undergone more decreases in a marker for primary biomass burning organic aerosol. "
R2.14) L44-47: "...rapidly evolve as smoke travels downwind, diluting and entraining
background air." I think you mean that dilution and entrainment can rapidly cause aerosol &
vapor evolution, but that is not how it reads.
Thank you--this is similar to comment R1.40) and we have clarified this sentence, addressing
both reviewer comments:
"Dilution through entrainment of regional background air can cause vapors and particles emitted
from fires to rapidly evolve as smoke travels downwind"
R2.15) L49: I think you mean "dilution at time of measurement".
Thank you--we have added this.
R2.16) L54: Does this refer to radiative fluxes?
Yes--we have updated this phrase to "shortwave radiative fluxes"
R2.17) L 55-57: Please fix the brackets around citations.
Fixed.
R2.18) L93: Should read "aging and oxidation of OA mass and aerosol number concentration
and mean Diameter."
We agree that this sentence is hard to parse; we've updated it:
"Here, we present smoke plume observations from the Biomass Burning Observation Project
(BBOP) campaign of aerosol properties from five research flights sampling wildfires downwind
in seven pseudo-Lagrangian sets of transects to investigate the evolution of OA mass and
oxidation state, aerosol number, and aerosol mean diameter."

R2.19) L112: 20-262 nm size range is not ideal, but I guess it is what you have.

We agree that we would have preferred a larger size range.

R2.20) L134-135: Also background correct m/z=44 and m/z=60?

Reviewer 1 was also unclear on our background-corrections and calculations for $f_{60}$ and $f_{44}$

(comment R1.9). We repeat our response here:

We calculated the background corrected f60 and f44 as follows (where $f = f_{60}$ or $f_{44}$):

$\Delta f \; = \; \dfrac{(f_{in} * OA_{in}) \, - \, (f_{out} * OA_{out})}{\Delta OA}$       Eq. R1

Similar, the $\Delta O/\Delta C$ and $\Delta H/\Delta C$ are calculated through (where $X = O$ or H):

$\dfrac{\Delta X}{\Delta C} \; = \; \dfrac{(X_{in\,plume} \, - \, X_{out\,of\,plume})}{(C_{in\,plume} \, - \, C_{out\,of\,plume})}$       Eq. R2

We've added Eqs. R1-R2 as Eqs. 1 and 2 in the main text and have updated other equation numbers and references.

R2.21) L 136: Conceptually, where does the lowest 10% of CO occur? Just outside of the plume as the plane circles back through? Is the background fairly constant for a flight leg? Do you adjust background each time the plane turns around and goes back to transect the plume again?

Figures S7-S11 (white solid line in each figure)  indicate that the CO outside of the plume is fairly constant. We do not adjust the background each time but instead use the lowest 10% for the entire flight path once the plane has reached the fire until the plane leaves the fire/smoke complex. The location of the lowest 10% varies from flight to flight and from leg to leg, but often occurs on the flight portion furthest from the smoke plume of each leg. As was noted in the text, we did sensitivity analyses of our results to our assumptions about background and in- plume CO values and our conclusions were not changed.

R2.22) L137: Is elemental O, H, and C calculated from O/C, H/C & OA or is H/C and O/C

calculated from the elemental O, H, C concentrations? Aiken et al (2007) estimate it in the later (Eqn 1).

We calculate elemental O, H, and C using O/C, H/C, and OA , assuming that all of the OA mass was from O, C, and H. We have added the following: "Elemental O, H, and C are calculated using the O/C and H/C and OA data from the SP-AMS (assuming all of the OA mass is from O,
C, and H),..." (underline ours)

R2.23) L 139: Typo ("…, we but do not…")

Fixed

R2.24) L164-165: Sentence grammar

Updated to:

"The true fire location and center at the time of sampling is likely different than the MODIS
estimates, depending on the speed of the fire front."

R2.25) L165-167: Why use the FIMS # distribution to determine plume center? Why not [CO],
[mrBC], total number concentration, etc? In the supplemental figures, it says the center-flow is
determined by number concentration (not distribution).

We have made an error here--we do use the total FIMS number concentration to determine our
plume center and have updated the text to reflect that. We use aerosol number as this study is
focused on aerosol properties as a function of dilution amount. We have updated the text here
(also following points made in R1.13):

"To estimate the physical age of the plume, we use the estimated fire center as well as the total
FIMS number concentration to determine an approximate centerline of the plume as the smoke
travels downwind (Figs. S1-S6). The centerline is subjectively placed to attempt to capture the
most-concentrated portion of the total number concentration for each plume pass, as we focus on
aerosol properties and their relations to dilution in this study. We use  mean wind speed and this
estimated centerline to get an estimated physical age for each transect. We did not propagate
uncertainty in fire location, wind speed, or centerline through to the physical age, which is a
limitation of this study."

R2.26) L170: Fix heading

Fixed

R2.27) L189: Measurement uncertainty should be plotted in Figures (sum of variance in data
represented by each data point + uncertainty in each instrumental recording)

This comment is similar to comments R2.8 and R1.14, and we refer the reviewer to our
responses there.
R2.28) L189-190: Changes in f60 and f44 should be provided as fractional (as displayed on axis
of Figure 1, etc). Relative changes (%) are confusing.
Reviewer 1 pointed out in comment R1.16 that much of the discussion in this paragraph (lines
185-194 of the original document) was not well-posed. We have deleted this discussion and
replaced it with:
"Figure 1 shows that $\Delta OA/\Delta CO$ and $\Delta BC/\Delta CO$ vary little with age for both the 5-15 and 90-100
percentile of $\Delta CO$ (p-values>0.5). A true Lagrangian flight with the aircraft sampling the same
portion of the plume and no measurement artifacts (e.g. coincidence errors at high
concentrations) would have a constant $\Delta BC/\Delta CO$ for each transect. This flight and other flights
studied here have slight variations in $\Delta BC/\Delta CO$ (Fig. 1; Figs. S14-S18), which may be indicative
of deviations from a Lagrangian flight path with temporal variations in emission and/or
measurement uncertainties. The remaining variables plotted also show some noise and few clear
trends, but it is apparent that the 5-15 and 90-100 percentiles do show a separation for many of
the individual metrics. In order to determine the existence or lack trends for these metrics, we
spend the remainder of this study examining each metric from all of the pseudo-Lagrangian
flights together."
R2.29) L192: Replace "number concentration" with either "normalized number concentration"
or "DN40-262 nm /DCO".
Thank you, we have changed this to read as "normalized number concentration"
R2.30) L192: I only see a decrease in DN40-262 nm /DCO between ~0.6 and 1.0 hours physical
age. Saying that it decreases with age implies a consistent trend. For Dp, this trend is hard to tell
if it is statistically significant.
Reviewer 1 had similar issues with this paragraph in comment R1.16 (see also comment R2.28
above) and we have modified the discussion entirely:
"Figure 1 shows that $\Delta OA/\Delta CO$ and $\Delta BC/\Delta CO$ vary little with age for both the 5-15 and 90-100
percentile of $\Delta CO$ (p-values>0.5). A true Lagrangian flight with the aircraft sampling the same
portion of the plume and no measurement artifacts (e.g. coincidence errors at high
concentrations) would have a constant $\Delta BC/\Delta CO$ for each transect. This flight and other flights
studied here have slight variations in $\Delta BC/\Delta CO$ (Fig. 1; Figs. S14-S18), which may be indicative of deviations from a Lagrangian flight path with temporal variations in emission and/or
measurement uncertainties. The remaining variables plotted also show some noise and few clear
trends, but it is apparent that the 5-15 and 90-100 percentiles do show a separation for many of
the individual metrics. In order to determine the existence or lack trends for these metrics, we
spend the remainder of this study examining each metric from all of the pseudo-Lagrangian
flights together."

R2.31) L197: What do you mean by "available …"?

By available, we mean when instruments were taking measurements--we have gaps in the
measurement data. We have added the following parenthetical statement:

"(Some transects do not have data available for specific instruments.)"

R2.32) L197-199: Really long sentence. I have had to read it 6-7 times to parse out what is
shown.

We have updated this to:

"Fig. 2a-e show available $\Delta OA/\Delta CO$, $\Delta f_{60}$, $\Delta f_{44}$ $\Delta H/\Delta C$, and $\Delta O/\Delta C$ edge and core data versus
physical age for each transect for each flight of this study. We color each line by the mean $\Delta OA$
within a $\Delta CO$ percentile bin from the transect closest to the fire, $\Delta OA_{initial}$."

R2.33) L200-201: Physical age is the distance between the transect-center to the fire-center
divided by the average windspeed? So does 0 physical age imply infinite or 0 windspeed?

It would imply that the measurement is directly over the fire center (fire center - transect center =
0), we've clarified this in the text:

"We note that although some of the physical ages appear to be at ~0 hours (e.g. over the fire)..."

R2.34) L203: The "…correlation coefficients (R) with initial plume OA mass,…" is not shown.
Do you mean to say that this is represented by DOAinitial?

Reviewer 1 had similar concerns in comments R1.17 and R.20. We copy our discussion here:

We see that our original text here is confusing and misleading. We have attempted to clarify it.
We are using a single value for $\Delta OA_{initial}$ for each transect within a Lagrangian set of transects
which is obtained from the first transect of the set. If a flight has two Lagrangian sets of
transects, there will be a different value of $\Delta OA_{initial}$ used for the two sets of transects, each again obtained from the first transect of each set. The original text may have been interpreted that we
used $OA_{initial}$ but we did not--we have clarified that. We use the changing values of $\Delta OA/\Delta CO$,
$\Delta f_{60}$, $\Delta f_{44}$ $\Delta H/\Delta C$, and $\Delta O/\Delta C$ as they age downwind to compare with initial OA. We have
updated this text (also following suggestions made in R1.20):

"Also included in Fig. 2 are the Spearman rank-order correlation tests (hereafter Spearman
tests), which are tests for monotonicity. The Spearman tests show correlation coefficients for
each flight set (Table S1) with the initial $\Delta OA$ of a flight set ($\Delta OA_{initial}$) against $\Delta OA/\Delta CO$,
$\Delta f_{60}$, $\Delta f_{44}$ $\Delta H/\Delta C$, and $\Delta O/\Delta C$ as each variable ages downwind. We also include Spearman tests
for the calculated physical age of the smoke for each flight set against these same variables. The
R values are labeled $R_{\Delta OA,initial}$ and $R_{age}$, respectively, in Fig. 2. For the correlations with
$\Delta OA_{initial}$, all transects in a given Lagrangian set of transects have the same $\Delta OA_{initial}$ value; for
flights with two Lagrangian set of transects, each set has its own $\Delta OA_{initial}$ value. Correlating to
$\Delta OA_{initial}$ provides an estimate of how the plume aerosol concentrations at the time of the initial
transect impact plume aging (aging both before and after this initial transect)."

R2.35) L202-204: Is the Spearman coefficient for concatenation of all data points from all
transects? If so, I am not sure it would make sense to do this way. Spearman's test tests for
monotonically increasing/decreasing values. Given that each transect set starts at a different
initial value you wouldn't expect the grouped transect sets to display a strong R-value. If you
want to use Spearman's test in this way, for Rage you could normalize each normalized value to
the initial normalized value to get a % change and plot that in Figure 2 and relevant
supplementary figures.

We do agree that variability in emissions will lead to a different initial value of $\Delta OA_{initial}$.
However, changes to the smoke aerosol (coagulation, dilution, evaporation, chemistry, etc.)
should be occurring before the time of the first measurement, and using $\Delta OA_{initial}$ helps show
that. If the changes in the factors in Figure 2 between the time of emission and the first transect
are affected by the plume density, this would lead to an increase in the Spearman $R_{\Delta OA, initial}$. Of
course, we are still impacted by variability in emissions within our current methods, and we have
added further disclaimers throughout the text following reviewer comments. As the reviewer
mentions, this scatter at the time of the first transect does reduce the Spearman $R_{age}$, but because
plume-density-dependent aging prior to the first transect is one of the potentially interesting
findings of this study, we feel that it is important to not normalize our changes. We have added
the following text to Sect. 3.1:

"We note that scatter in $\Delta OA_{initial}$ leads to weaker $R_{age}$ values than would be obtained if we
normalized changes with aging to the first (normalized) value. However, as plume-density-
dependent aging prior to the first transect is one of the potentially interesting findings of this
study, we feel that it is important to not normalize our changes further."

R2.36) L206: Spell out "Figs." And lower case.

Fixed

R2.37) L207-208: Type in list "…FIMS measurements AND BACKGROUND and DCO

percentile Spacings…"

We have updated this section. We also have changed "background CO" to "in-plume CO

threshold value", as the latter is accurate and background CO is misleading.

"Figs. S13, S19-S21 show the same details as Fig. 2 but provide sensitivity tests to potential

FIMS measurement artifacts (Fig. S13) and our assumed in-plume CO threshold value (set to

150 ppbv for Figs. 1-3; Sect. 2) and $\Delta CO$ percentile spacing (Figs. S19-S21). Although these figures show slight variability, the findings discussed below remain robust and we constrain the rest of our discussion to the assumptions made for the FIMS measurements, in-plume CO

threshold value, and $\Delta CO$ percentiles used in Fig. 2."

R2.38) L209: Previous line said you would only discuss FIMS, background and DCO.

We see that this sentence is confusing, we intend that our assumptions used in Fig. 2 about the

FIMS measurements, CO, and delta(CO) percentiles will be used throughout the rest of the study. We have clarified the text:

"Although these figures show slight variability, the findings discussed below remain robust and we constrain the rest of our discussion to the original assumptions made for the FIMS

measurements, in-plume CO threshold value, and $\Delta CO$ percentiles used in Fig. 2."

R2.39) L209-210: RDOA,initial just says 0 in figure.

Thank you for catching this, the R value is 0 here and we have updated the text:

"In general, both the cores and edges show little change in $\Delta OA/\Delta CO$ with physical aging, with

$R_{\Delta OA,initial}$ and $R_{age}$ at 0 .02 and 0.03… "

R2.40) L209-210: This figure shows orders of magnitude changes in DOA/DCO with age. I

think you mean there is not a clear positive or negative trend (as stated in the first clause of the next sentence), not that there is no change.

We have updated the text from "show little change" to "do not show any positive or negative trend".

R2.41) L212: Here and elsewhere, spell out "vs." Check grammar.

We have fixed the vs. errors and have done a thorough grammar check. We have made many
small changes to improve readability and grammar.

R2.42) L213: For positive R values, consider putting a "+" sign in front of the value.

This does improve clarity and we have updated the positive R values to have a + sign throughout
the manuscript.

R2.43) L214-218: Consider breaking this into multiple, shorter sentences. Check for redundancy
with L212-214, i.e. a negative R value means there is a decreasing trend.

We have updated this section (including suggestions following reviewer 1's comment R1.22).
We removed the sentence in L212-214, as it is redundant, and incorporated the R values into the
updated text:

"$\Delta f_{60}$ generally decreases with plume age ($R_{age}$ = -0.26), consistent with the hypotheses that $\Delta f_{60}$
may be evaporating because of dilution, undergoing heterogeneous oxidation, and/or having a
decreasing fractional contribution due to condensation of other compounds.. In contrast, $\Delta f_{44}$
generally increases with age ($R_{age}$ = +0.5) for all plumes with available data. It appears for the
plumes in this study that although there is little change in $\Delta OA/\Delta CO$, loss of compounds that
contain $f_{60}$ fragments (as captured by the SP-AMS) is roughly balanced by condensation of
more-oxidized compounds, including those that contain compounds with $f_{44}$ fragments, such as
carboxylic acids. This observation suggests the possibility of heterogeneous or particle-phase
oxidation that would alter the balance of $\Delta f_{60}$ and  $\Delta f_{44}$."

R2.44) L214-218: Is it only evaporation or condensation (phase changes) happening or does O
attack volatile and semivolatile species (levoglucosan) changing its molecular composition to
more oxidized/refractory species without a phase change?

Reviewer 1 made similar comments in R1.23. We answer this comment and the next comment
(R2.45) as well as R1.23:

We are trying to note that heterogeneous chemistry is relatively slow (for near-field aging) and
shouldn't significantly contribute to compositional changes. We have added text to emphasize
that point more clearly:

"However, estimates of heterogeneous mass losses indicate that after three hours of aging for a
range of OH concentrations and reactive uptake coefficients, over 90% of aerosol mass is
anticipated to remain, indicating that heterogeneous loss has limited effect on aerosol
composition or mass (Fig. S23; see SI text S2 for more details on the calculation). Hence, the
evaporation of $f_{60}$ being balanced by gas-phase production of $f_{44}$ may be the more likely
pathway"

R2.45) L218-220: If you didn't expect a change in normalized-OA anyway based on your
model, why do you suggest a balance between evaporation particle mass loss and condensation
mass gain?

The evaporative loss may be driven by dilution and the condensation may be driven by
production of lower-volatility species from oxidation of either evaporated POA or more-volatile
SOA precursors.

R2.46) L221: Those are not very strong R values to base your interpretations on, but I wouldn't
expect them to be for the reasons discussed above. This statement is not particularly true for f60.

We have unified our language when discussing R and $R^2$ values throughout the text, following
reviewer comment R1.20 as well as this comment.

R2.47) L224: But you just said that Df60 and Df44 correlate with DOAinitial. Differences in
your initial Df60 or Df44 don't necessary need a mechanistic explanation. We see variance these
parameters in fresh emission in laboratory experiments and would expect to also see variance in
primary emissions of wildfires. This is not good support for your next conclusion (that aircraft
observations are missing evaporation and/or condensation).

Reviewer 1 had similar concerns in comments R1.24 and 1.39. Our response to both R1.24 and
R2.47 is:

Both reviewers makes reasonable points here and we agree that these are alternative hypotheses
that should be explicitly discussed in the manuscript. Reviewer 2 made similar comments in
R2.47. Unfortunately, lacking direct measurements of the emissions, we cannot explore this
hypothesis in any detail. And we do find it compelling that less-dense plumes do show higher
f44/lower f60 than more-dense plumes, which supports our hypothesis of aging prior to the
transect. We have added the following text to Sect. 3.1:

"We note that each fire may emit particles with variable initial $f_{44}$ and $f_{60}$ values, as has been
observed in laboratory studies (Hennigan et al. 2011; Cubison et al. 2011; McClure et al. 2020),
which adds to scatter within the data. It is possible that variability in f44 and f60 may also
contribute to the observed correlations with $\Delta OA_{intial}$; however, this would require that higher $f_{44}$

emissions are correlated with lower emissions rates and/or faster dilution rates (and visa versa for
$f_{60}$). Lacking direct emissions measurements, this hypothesis cannot be further explored in this
work."

To Reviewer 1's last query ("Also, given that different sources produce particles that have
different initial f60 and f44, would they be expected to exhibit the same deltaf60 and deltaf44
even if initial OA and dilution were identical? Is there evidence that this is expected?"), we
would not expect the same $\Delta f_{44}$ and $\Delta f_{60}$ under those circumstances and thus variability from
emissions likely contributes to the noise of our fit parameters. We do include a brief discussion
on this in the text in Sect. 3.1 within the discussion of our fit parameters (with new minor edits
addressing comments from reviewer 2):

"The scatter is likely due to variability in emissions due to source fuel or combustion conditions,
instrument noise and responses under the large concentration ranges encountered in these smoke
plumes, inhomogeneous mixing within the plume, variability in background concentrations not
captured by our background correction method, inaccurate characterizations of physical age due
to variable wind speed, and/or deviations from a true Lagrangian flight path."

We also address this issue in the conclusions. We have added more text and qualifiers to section
3 addressing this issue, following comments R1.24 and R2.47. We add the following text to this
discussion:

"We were unable to quantify the impact on potential interfire variability in the emission values
of the metrics studied here (such as variable $f_{60}$ and $f_{44}$). We anticipate that being able to capture
this additional source of variability may lead to stronger fits and correlation."

And

"We also suggest further refinement of our fit equations, as further variables (such as photolysis
rates) and better quantification of interfire variability (such as variable emission rates) are
anticipated to improve these fits."

R2.48) L227: Is this logic circular? That differences in DOAinitial is due to different emission
fluxes?

Differences in $\Delta OA_{initial}$ (which is the $\Delta OA$ of the first flight transect, not the $\Delta OA$ directly
emitted from the fire) can stem from a variety of reasons beyond emission fluxes. We include
some further reasons in our original text, copied here: "*The differences in $\Delta OA_{initial}$ between*
*plumes may be due to different emissions fluxes (e.g., due to different fuels or combustion*
*phases), or plume widths, where larger/thicker plumes dilute more slowly than smaller/thinner*

*plumes; these larger plumes have been predicted to have less evaporation and may undergo*
*relatively less photooxidation (Bian et al., 2017; Hodshire et al., 2019a, 2019b)."*

R2.49) L228: should not be a comma after the bracket.

Fixed

R2.50) L231 & 234: Reference format needs to be changed.

Fixed

R2.51) L234: Grammar. Reference to figure in Garofalo should be something like "(Fig. 6 in
Garofalo et al, 2019)"

Fixed

R2.52) L235-236: Isn't that why you normalize?

The lack of trends from physical edge to core is most likely due to inhomogeneous mixing
(which will not be improved by subtracting background concentrations), which is our next
sentence, repeated here for reference:

*This could be as CO concentrations (and thus presumably other species) do not evenly increase*
*from the edge to the core for many of the plume transects studied (Figs. S2-S6).*

We have added "... the remaining plumes do not show a clear trend from the physical edges to
cores" (underline ours) to this statement to emphasize that we are discussing the physical
transect, rather than the divisions made by ΔCO percentile bins.

R2.53) L237-239: You imply that patterns of f60 and f44 compared to shortwave irradiance is
related by photolysis rates. I don't necessarily agree with this interpretation. If the plume is
thicker it means that a higher fraction of aerosol mass is from the fire and because fire-emitted
aerosol has higher f60 and lower f44 than background a simple mechanism of mixing explains
your observations.

Our f60 and f44 values are background corrected (please see section 2 and newly added equation
2; comment R1.9), which should correct for mixing. We are also not trying to draw any firm
conclusions here, but are pointing out observational similarities (underline added for quick
reference): *We do not have UV measurements that allow us to calculate photolysis rates but the*
*in-plume shortwave measurements in the visible show a dimming in the fresh cores that has a*

*similar pattern to f₄₄ and the inverse of f₆₀ (Fig. S26; the rapid oscillations in this figure could be*
*indicative of sporadic cloud cover above the plumes).*
R2.54) L242-243: DO/DC and f44 are both proxies for OOA and would be expected to have the
same trends. DH/DC and f60, while not conceptually the same, both reflect primary BBOA and
would also be expected to show the same trends It is a little redundant to analyze both sets.
We have reviewed a significant amount of biomass burning (BB) literature and have noted that
many studies examine f44/f60 or O:C/H:C or both. Furthermore, while f44/f60 are popular
within AMS BB measurement studies, models currently can only predict O:C/H:C. We chose to
include both for completeness and ease of comparisons to other datasets in future studies. We
agree that it's unsurprising to see similarities between the DO/DC and Df44 and DH/DC and
Df60 results, given their relations, particularly for DO/DC and Df44. We have added the
following text within this paragraph:
"Given that $\Delta f_{44}$ and $\Delta O/\Delta C$ are both metrics for OA aging (Sect. 2), it is unsurprising that we
see similar trends between them."
R2.55) L242-243: See issues raised earlier regarding interpreting Spearman's test results for
these data sets.
We refer the reviewer to our responses on comments R2.8 and R.35.
R2.56) L249-264: You should provide explanation for why you used these equations to try and
fit f44and f60. Is there a conceptual justification for them? Do they have meaning outside of a
mathematical fit?
We tried a large number of mathematical fits and these equations (Eqs. 2-3 in the original text;
Eqs. 4-5 in the updated text) performed the best. They do not have a direct physical meaning, and
we have added the following to the end of this discussion:
"Eqs. 4-5 performed the best out of the mathematical fits that we tested. They do not have a
direct physical interpretation but may be used as a starting point for modeling studies as well as
for constructing a more physically-based fit."
R2.57) L263-268: What do you mean by "Aged Df60 and Df44"? Does "limiting the predictive
skill"mean that your fits are not particularly informative?
We were referring to the aging values of $\Delta f60$ and $\Delta f44$, we were not careful in our language
here though. "Limiting the predictive skill" was perhaps not the best phrase to use--we are trying to argue that the scatter in the measurement data is likely contributing to the limited predictive
power of our current mathematical fits. We note that the p-values for these fits for $\Delta f60$ and $\Delta f44$
(as well as the other variables in Fig. 3, mean Dp $\Delta O/\Delta C$) and are both less than 0.01 and we
argue that our fits provide valuable information on how physical age and a metric for plume size
(here, initial OA at the time of the first measurement) impact $\Delta f60$ and $\Delta f44$. We now note in the
text that the p-values are <0.01 for all fits and we have updated this section to read:

"The aging values of  $\Delta f_{60}$ , $\Delta f_{44}$ , and $\Delta O/\Delta C$ show scatter (Figs. S14-18), which likely
contributes to the limited predictive power of our mathematical fits. The scatter is likely due to
variability in emissions due to source fuel or combustion conditions, instrument noise and
responses under the large concentration ranges encountered in these smoke plumes,
inhomogeneous mixing within the plume, variability in background concentrations not captured
by our background correction method, inaccurate characterizations of physical age due to
variable wind speed, and/or  deviations from a true Lagrangian flight path. Eqs. 4-5 performed
the best out of the mathematical fits that we tested. These equations do not have a direct physical
interpretation but may be used as a starting point for modeling studies as well as for constructing
a more physically based fit. There may be another variable not available to us in the BBOP
measurements that can improve these mathematical fits, such as photolysis rates. We do not
know whether these fits may well-represent fires in other regions around the world, given
variability in fuels and burn conditions. We also do not know how these fits will perform under
nighttime conditions, as our fits were made during daytime conditions with different chemistry
than would happen at night. We encourage these fits to be tested out with further data sets and
modeling. These equations are a first step towards parameterizations appropriate for regional and
global modeling and need extensive testing to separate influences of oxidation versus dilution-
driven evaporation."

R2.58) L264-265: typos/grammar

Fixed

R2.59) L271-272: The decrease in normalized number concentration with physical age mostly
appears to be caused by 2-3 outlier measurements (the initial points for leg 730b edge, the initial
value of another edge, and the tailing value of leg 726a 1). This does not seem like a statistically
robust claim and I think the R value verifies it. Lines 275-277 seem to agree with my
Assessment.

We agree with this assessment--reviewer 1 has asked us to be more precise in our language for
reviewer comments (please see R1.20) and we have noted that this is a weak correlation within
these sentences. We also note that reviewer 1 asked for a test in which we leave one flight out, sequentially, to see how each R value changes (comment R1.21). We have done this and include
language in the text as well as Table S2, summarizing the results.
R2.60) L273-274: "generally have lower normalized … by the time of the first measurement".
This implies that there was a measurement made before the first measurement. Please explain.
We are merely trying to comment on our observations from the data here.We do not think that
our text is implying that there's a measurement before the first measurement--perhaps this is
made more clear by changing the phrase "by the time" to "**at** the time", and we have changed our
text thusly.
R2.61) L273-274: "plume edges and cores with the highest DOA generally have lower
normalized number concentrations…" This is not true based on figure 2f. The two lowest
DOAinitial values (white dashed lines) have two of the highest DN/DCO values.
We respectfully point out that our quoted text here is discussing "highest $\Delta OA$ and low
$\Delta N/\Delta CO$" whereas the reviewer is pointing out "lowest $\Delta OA$ and highest $\Delta N/\Delta CO$"--the two
arguments are consistent with each other.
R2.62) L279: Evaporation (mass loss/time) is, partially, a function of available surface area.
Since small particles have a higher surface area-to-volume, it is plausible that evaporation will
decrease the number of small particles more than large particles and therefore increase the mean
particle size. You state this possibility of preferential loss of small particles on lines 293-295.
This is a reasonable point--if evaporation is gas-phase mass-transfer limited, evaporation will
decrease the size of smaller particles more than larger particles. However, this case would only
lead to an increase in the mean diameter if a significant number of small particles shrunk to
below 40 nm, removing them from the calculation of the mean Dp. And if evaporation is in
quasi-equilibrium, evaporation is independent of surface area. However--the organic mass of the
plume does not change significantly, so we do not have evidence to support this hypothesis for
the increase in mean Dp. We have added the following text to this discussion:
"OA evaporation will decrease $D_p$ if the particles are in quasi-equilibrium (where evaporation is
independent of surface area) (Hodshire et al. 2019b). However, if evaporation is kinetically
limited, smaller particles will preferentially evaporate more rapidly than larger particles, which
may lead to an increase in $D_p$ if the smallest particles evaporate to below 40 nm(Hodshire et al.
2019b). The plumes do not show significant changes in $\Delta OA/\Delta CO$ (Fig. 2a), indicating that
coagulation is likely responsible for the majority of increases in $D_p$."

R2.63) L282-283: should be $R_P^2$ instead of $R^2_P$.

We've fixed this formatting here and elsewhere in the text.

R2.64) L282-283: you were previously using R and not R2 (L272, Fig 2, etc). In my opinion,
this is fine and depends on how you use them, but I have been reviewed differently. Did you
intend to calculate R and R2? Please check to make sure that you they are used and calculated
correctly. I only state this because there are a number of typos in the manuscript and want to
make sure that this is not one.

We did indeed intend to calculate $R^2$ here. Calculating R previously was useful to indicate the
sign of the correlation whereas here with $R^2$ we intend to show what fraction of the variability is
captured, since all fits are positively correlated. We have added the following text:

We show $R^2$ here to indicate the fraction of variability captured by these fits, whereas calculating
R for the trends in Fig. 2 indicate the direction of the correlation.

R2.65) L287: Do you mean "legs" instead of "days"?

We have updated this text to "Lagrangian set of transects" to match the language of our other
text.

R2.66) L294: Replace "~" with "approximately"

We have updated this instance of '~' and all others in the text for consistency.

R2.67) L301-302: As mentioned above, I do not agree that the data supports the statement
regarding correlation. I think there is a lot of good analysis in this paper and I don't think you
need to make this statement.

We update this text to be more subjective and consistent with our terminology added in response
to R1.20:

"We find that although $\Delta OA/\Delta CO$ does not correlate with $\Delta OA_{initial}$ or physical age, $\Delta f_{60}$ (a
marker for evaporation) is moderately correlated with $\Delta OA_{initial}$ (Spearman rank-order correlation
tests correlation coefficient, $R_{\Delta OA,initial}$, of +0.43) and weakly correlated with physical age
(Spearman rank-order correlation tests correlation coefficient, $R_{age}$, of -0.26). $\Delta f_{44}$ and $\Delta O/\Delta C$
(markers for photochemical aging) increases with physical aging (moderate correlations of $R_{age}$
of +0.5 and +0.56, respectively) and are inversely related to $\Delta OA_{initial}$ (moderate correlations of
$R_{\Delta OA,initial}$ of -0.55 and -0.45, respectively)."

R2.68) L302-304: I also do not agree that the data supports the statement regarding DN/DCO.

We have removed the latter half of this sentence, which is consistent with edits made previously in the manuscript.

R2.69) L304: You don't need to keep specifying that diameter size range of 40-262.

We removed this mention of the size range.

R2.70) L306-308: I don't like saying this, I don't agree that your data support this statement. The only way that differences in Df44initial, Df60initial and DO/Cinitialsupport this statement is if all primary OA from all wildfires have the same value which has been shown to not be true.

We respectfully disagree here--variability in the emitted oxidation markers from fire to fire is most likely random, and yet we see correlations despite the random variability. The only way this comment would be true is if the emitted oxidant markers are correlated with OA emission rates, fire size, and/or dilution rates prior to the first transect--there is currently no evidence for this. We choose to keep this statement as is. We note that in Sect 3.1 we have the following statement (and have added additional text to further emphasize these points, underlined here to clearly show what's been added):

Differences in $\Delta f_{60}$ and $\Delta f_{44}$ for the nearest-to-source measurements indicate that evaporation and/or chemistry likely occurred before the time of these first measurements (assuming that emitted $\Delta f_{60}$ and $\Delta f_{44}$ do not correlate with $\Delta OA_{initial}$; there is currently no evidence for this alternative hypothesis).

R2.71) Figure 1: Change "BC" to "rBC" in the legend and axis. Also in Figures S14-S18

We have changed all mentions of 'rBC' in the text to 'BC' to be consistent with our figure notation and note in the text when Fig. 1 is introduced that BC is for the refractory BC from the SP2.

R2.72) Figure 1: Change DN/DCO to DN40-262 nm /DCO to be consistent with text.

We have noticed our inconsistency of $\Delta N/\Delta CO$ vs. $\Delta N_{40\text{-}262\ nm}/\Delta CO$ throughout our figures. We had originally divided our analysis into $\Delta N_{40\text{-}262\ nm}/\Delta CO$ vs $\Delta N_{<40nm}/\Delta CO$ but did not include the $\Delta N_{<40nm}/\Delta CO$ analysis in the final paper. We apologize for these inconsistencies and have changed all instances in the text and figures to simply ΔN/ΔCO. We have done the same for $D_p$

vs $D_{p,40-262\,nm}$ (updating all mentions of the latter to the former).

R2.73) Figure 2: Caption should be "function of physical age"

Good catch, thank you. Fixed

R2.74) Figure 2: This figure is pretty confusing. If I look at Figure S2, I see that for leg 726a there were 2 sets of transects with each comprising of 4 transects. So, theoretically, the same air mass was sampled 4 times corresponding to 4 different physical ages. So a line in figure contains

~4 data points which correspond with either the edge or core of a transect in the transect set? Am

I reading this correct?

For the flights that have 2 Lagrangian sets of transects or days with 2 separate flights ('726a',

'730a', and '730b'), Figure 2 will contain one line for each Lagrangian set of transects downwind. The physical age is assumed to be constant across a given flight transect (see comment R2.4 for further discussion on this), as mentioned in the manuscript with minor edits for clarity,

"We use the mean wind speed and this estimated centerline to calculate an estimated physical age for each transect, and this physical age is assumed to be constant across the transect, as plume crossing took between 50-500 seconds".

We include the following text to clarify the reviewer's other comments on Figure 2 here:

"Flights with two sets of pseudo-Lagrangian transects ('726a' and '730b') have two separate lines in Fig. 2, one for each set."

R2.75) How does the white dashed line in 2a go backwards in physical age?

The white line in 2a is for flight '809a'. Figure S5 (S5 of the original submission) shows that 2

legs essentially overlap. We have added subpanels to Figs. S2-S6 that indicates the time-of-flight for each flight. However, the leg slightly further from the fire occurred first in the flight so it has a calculated age slightly older than the next leg, as the calculation depends in part on distance from the fire. This is a limitation of our method. We have added the following text to the first paragraph of Sect. 3.1:

"As well, two legs for flight '809a' nearly overlap (Fig. S5), with the leg that is further from the fire occurring first in the flight path, leading to an apparent slight decrease in physical age for the sequential leg (see e.g. the white dashed line in Fig. 2a)."

R2.76) Figure 2: Change to RDOA,initial instead of double subscript to be consistent with that
used in text.
Thank you for catching this--we have updated these labels.
R2.77) Figure S1: I don't see a black star or dashed line.
We have added these to the figure, thank you. We include the new version of Fig. S1 after
comment R2.80.
R2.78) Figure S1: Leg number not indicated. ("The numbers are the leg number")
We have removed this reference . Figures S2-S6 now include the leg numbers, and this is
reflected in these figure captions.
R2.79) Figure S1: I would suggest that you use a different symbol and symbol color for the
MODIS thermal anomalies so that it contrasts with the color code of the # concentration.
We have changed our color palette for the number concentration to 'plasma', which hopefully
provides enough contrast.
R2.80) Figure S1: Please change the colorcode to a color-blind friendly one.
We have changed our color palette for the number concentration to 'plasma'. We include the
updated figure and caption below, as reference.

[Figure]

Figure S1. The flight path for flight '730b', colored by the FIMS total number concentration. The
red dots are MODIS fire/thermal anomalies. The black star indicates the approximate center of
the fire and the black dashed line indicates the approximate centerline of the plume, estimated by
the number concentration.
R2.81) Figure S5: Is the black star the fire center for 8/9/2013 or 8/8/2013? The caption does not
say what symbol is used for 8/8/2013, only that "The black star indicates the approximate center
of the fire…"
We do not show the fire location on 8/8/2013 or 8/10/2013; we instead are estimating the fire
center on 8/9/2013 (black star) using MODIS images from 8/8/2013 and 8/10/2013. We have
added in a green star to this figure to indicate the approximate fire center on 8/8/2013.

R2.82) Figure S24-S25: The y-axis scale changes between graphs, with a wide range for data
that do not look like they have much variation (leg 730a) and a smaller range for others (730b).
Is this why there is not consistent patterns in 730a and 730b?

We have tightened the y axes on the subpanels that had too much whitespace. We thank the
reviewer for pointing this out.

R2.83) Figure S26: is shortwave irradiance a measure of photo-chemical rate, the amount of
scattering/absorbing aerosol above you, or a combination of both?

In this study, we're using the total shortwave irradiance as measured by an SPN1 (Long et al.,
2010). The shortwave irradiance is a function of solar angle and scattering/absorption prior to the
measurement. While it is not a measure of the UV wavelengths that drive photochemistry, we are
using it as a rough proxy for these wavelengths so that we can look at how photolysis rates may
vary across the flight path. We note this in our original text: "*We do not have UV measurements*
*that allow us to calculate photolysis rates but  the in-plume SPN1 shortwave measurements in*
*the visible show a dimming in the fresh cores that has a similar pattern to $f_{44}$ and the inverse of*
*$f_{60}$ (Fig. S26; the rapid oscillations in this figure could be indicative of sporadic cloud cover*
*above the plumes)."*

Long, C. N., A. Bucholtz, H. Jonsson, B. Schmid, A. Vogelmann, and J. Wood (2010): A
Method of Correcting for Tilt from Horizontal in Downwelling SW Measurements on Moving
Platforms, TOASJ, 4, pp.78-87, doi: 10.2174/1874282301004010078

R2.84) Figure S27: Please complete the drawing of the Van Krevelen diagram with the 1:1, 2:1,
and 0.5:1 lines

Literal 1:1, 2:1, and 0.5:1 lines are rather uninformative, as can be seen in the below figure.

[Figure]

1753 We think that the reviewer may have intended constant lines of oxidation, as shown in Figure 1
1754 of Heald et al. (2010) (their red and blue lines). Heald et al. (2010) chose a starting point of
1755 H/C=2 at O/C=0 (which is the case for long alkanes), which upon visual inspection is not an
1756 appropriate starting point for our data. We do not know exactly what the appropriate H/C and
1757 O/C starting point for primary biomass burning OA is, given variability in the emissions during
1758 BBOP and literature values. We do not add these lines of oxidation for this reason. We note that
1759 reviewer 1 had confusion with this figure, and we refer reviewer 2 to R1.7 and R1.25 for further
1760 details.

1762 Heald, C. L., Kroll, J. H., Jimenez, J. L., Docherty, K. S., Decarlo, P. F., Aiken, A. C., Chen, Q.,
1763 Martin, S. T., Farmer, D. K. and Artaxo, P.: A simplified description of the evolution of organic
1764 aerosol composition in the atmosphere, Geophys. Res. Lett., 37(8), doi:10.1029/2010GL042737,
1765 2010.

**Dilution impacts on smoke aging: Evidence in BBOP data**

Anna L. Hodshire[1], Emily Ramnarine[1], Ali Akherati[2], Matthew L. Alvarado[3], Delphine K. Farmer[4], Shantanu H. Jathar[2], Sonia M. Kreidenweis[1], Chantelle R. Lonsdale[3], Timothy B. Onasch[5], Stephen R. Springston[6], Jian Wang[6,a], Yang Wang[7,b], Lawrence I. Kleinman[6], Arthur J. Sedlacek III[6], Jeffrey R. Pierce[1]

[1] Department of Atmospheric Science, Colorado State University, Fort Collins, CO 80523, United States
[2] Department of Mechanical Engineering, Colorado State University, Fort Collins, CO 80523, United States
[3] Atmospheric and Environmental Research, Inc., Lexington, MA 02421, United States
[4] Department of Chemistry, Colorado State University, Fort Collins, CO 80523, United States
[5] Aerodyne Research Inc., Billerica, MA 01821, United States
[6] Environmental and Climate Sciences Department, Brookhaven National Laboratory, Upton, NY 11973, United States
[7] Center for Aerosol Science and Engineering, Washington University, St. Louis, MO 63130, United States
[a] Now at Center for Aerosol Science and Engineering, Washington University, St. Louis, MO 63130, United States
[b] Now at Department of Civil, Architectural and Environmental Engineering, Missouri University of Science and Technology, Rolla, Missouri 65409, United States

*Correspondence to*: Anna L. Hodshire (Anna.Hodshire@colostate.edu)

**Abstract.** Biomass burning emits vapors and aerosols into the atmosphere that can rapidly evolve as smoke plumes travel downwind and dilute, affecting climate- and health-relevant properties of the smoke. To date, theory has been unable to explain variability in smoke evolution. Here, we use observational data from the BBOP field campaign and show that initial smoke organic aerosol mass concentrations can help predict changes in smoke aerosol aging markers, number concentration, and mean diameter between 40-262 nm. Because initial field measurements of plumes are generally >10 minutes downwind, smaller plumes will have already undergone substantial dilution relative to larger plumes. However, the extent to which dilution has occurred prior to the first observation is not a directly measurable quantity. Hence, initial observed plume concentrations can serve as a rough indicator of the extent of dilution prior to the first measurement, which impacts photochemistry and aerosol evaporation. Cores of plumes have higher concentrations than edges. By segregating the observed plumes into cores and edges, we infer that particle aging, evaporation, and coagulation occurred before the first measurement. We further find that on the edges, the oxidation state of organic aerosol has increased , and we find that edges

 undergone more decreases in a marker for primary biomass burning organic aerosol.   and less coagulation than the cores.

[revised manuscript text omitted]

However, no consistent evidence of changing collection efficiencies in field studies exist yet. ▪
[revised manuscript text omitted]
 Figs. S1-S6). The centerline is subjectively placed to attempt to capture the most-concentrated portion of the total number concentration for each plume pass, as we focus on aerosol properties and their relations to dilution in this study. and We use the mean wind speed and this estimated centerline to calculate an estimated physical age for each transect, and this physical age is assumed to be constant across the transect, as plume crossings took between 50-500 seconds. We did not propagate uncertainty in fire location, wind speed, or centerline through to the physical age, which is a limitation of this study.

**3 Results and discussion**

As a case example, we examine the aging profiles of smoke from the Colockum fire during the first set of pseudo-Lagrangian transects on flight 730b (Table S1). Figure 1 provides $\Delta OA/\Delta CO$, $\Delta rBC/\Delta CO$ $\Delta f_{60}$, $\Delta f_{44}$, $\Delta H/\Delta C$, $\Delta O/\Delta C$, $\Delta N_{40\text{-}262\ nm}/\Delta CO$, and $\overline{D_p}$ as a function of the estimated physical age; Figs. S14-S18 provides this information for the other pseudo-Lagrangian transect sets studied. (Here, BC represents the refractory BC from the SP2; Sect. 2.) We have divided each transect into four regions: between the 5-15 (edge), 15-50 (intermediate, outer), 50-90 (intermediate, inner), and 90-100 (core) percentile of $\Delta CO$ within each transect. Figures S2-S6 show the locations of these CO percentile bin for each transect of individual flights. Figure 1 shows the edge and core data, both averaged per transect, with Figs. S14-18 providing all four percentile bins for each flight. These percentile bins correspond with the thinnest (least CO-dense) to thickest (most CO-dense) portions of the plume, respectively. If a fire has uniform emissions ratios across all regions and dilutes evenly downwind, these percentile bins would correspond to the edges, intermediate regions, and the core of the diluting plume. We use this terminology in this study but note that uneven emissions, mixing, and/or dilution lead to the percentile bins not physically corresponding  to our defined regions in some cases. We note that some plumes show more than one maxima in CO concentrations within a given plume crossing, which implies that there may be more than one fire or fire front, and that these plumes from separate fires or fronts are not perfectly mixing. As well, in at least one of the fires (in flights '730a' and '730b'), the fuels vary between different sides of the fire, as discussed in Kleinman et al., 2020. However, the lowest two $\Delta CO$ bins tend more towards the physical edges of the plume and the highest two tend more towards the physical center of the plume (Figs. S2-S6). We do not use the data from the lowest 5% of $\Delta CO$ to reduce uncertainty at the plume-background boundary. We do not know where the plane is vertically in the plume, which is a limitation as vertical location will also impact the amount of solar flux able to penetrate through the plume.

Figure 1 shows that for this specific plume, $\Delta OA/\Delta CO$ and $\Delta rBC/\Delta CO$ vary little with age for both the 5-15 and 90-100 percentile of $\Delta CO$ (p-values>0.5). A true Lagrangian flight with the aircraft sampling the same portion of the plume and no measurement artifacts (e.g. coincidence errors at high concentrations) would have a constant $\Delta rBC/\Delta CO$ for each transect. This flight and other flights studied here have slight variations in $\Delta rBC/\Delta CO$ (Fig. 1; Figs. S14-S18), which may be indicative of deviations from a Lagrangian flight path with temporal variations in emission and/or measurement uncertainties. The remaining variables plotted also show some noise and few clear trends, but it is apparent that the 5-15 and 90-100 percentiles do show a separation for many of the individual metrics. In order to determine the existence or lack of trends for these metrics, we spend the remainder of this study examining each metric from all of the pseudo-Lagrangian flights together. ~~For this flight, $\Delta f_{60}$ changes little (p-values>0.5approximately~±6% for both the between edge and core) while $\Delta f_{44}$ increases slightly (approximately~8% for both edge and core between the initial and final transect) with age, with edges showing the highest $\Delta f_{44}$. $\Delta H/\Delta C$ decreases while $\Delta O/\Delta C$ changes little; however, the increases with the edges showing higher values of $\Delta O/\Delta C$. Mean aerosol diameter (between 40-262 nm) increases with aging. and nThe normalized number concentration of this same size range changes little in the 90-100 percentile of $\Delta CO$ for this flight, while it decreases in the 5-15 percentile of $\Delta CO$ between 40-262 nm with aging. The decrease in normalized number concentration is presumably due to coagulation, as little dry deposition would occur within these timescales (<2.5 hours). These trends are discussed for all flights in the following sections.~~

[revised manuscript text omitted]

**Text S1. Further details on BBOP instrumentation**

The Fast Integrated Mobility Spectrometer (FIMS) characterizes particle sizes based on electrical mobility as in scanning mobility particle sizer (SMPS). Because the FIMS measures particles of different sizes simultaneously instead of sequentially as in traditional SMPS, it provides aerosol size distribution with a much higher time resolution at 1 Hz (Wang et al., 2017). The relative humidity of the aerosol sample was reduced to below ~25% using a Nafion dryer before being introduced into the FIMS. Therefore, the measured size distributions represented that of the dry aerosol particles. The particle number concentration integrated from FIMS size distribution typically agrees with the CPC 3010 (Condensation Particle Counter) measurement (Kleinman et al., 2020) within ~ 15% when size distribution suggests that particles smaller than 10 nm contribute negligibly to the total number concentration. Thus, we estimate the uncertainty in the FIMS number concentration to be ~15%. The uncertainty in measured particle size is about 3% (Wang et al., 2017).

The Soot Particle Aerosol Mass Spectrometer (SP-AMS) is thoroughly detailed in Kleinman et al. (2020). Although it was not directly characterized for uncertainties during the BBOP campaign, we estimate uncertainties as follows. The AMS uncertainty is estimated following the methods in (Bahreini et al. 2009) (first equation of their supplemental information), leading to 37% uncertainty for organics. The laser vaporizer adds additional uncertainty up to 20%. Thus summing the uncertainties in quadrature leads to a 42% uncertainty in organics. The Soot Photometer (SP2) had an uncertainty of 20%.

CO measurement uncertainties are detailed in Kleinmen et al. (2020): the Off-Axis Integrated Cavity Output Spectroscopy was found to have an accuracy of 1-2%, and the precision at ambient backgrounds of 90 ppb was 0.5 ppbv RMS (using a 1 second averaging).

An SPN1 radiometer (Badosa et al., 2014; Long et al., 2010) provided total shortwave irradiance, with a shaded mask applied following (Badosa et al., 2014). The data was corrected for tilt up to 10 degrees of tilt, following (Long et al., 2010). For tilt greater than 10 degrees these values are set to "bad". Instrument uncertainties are detailed in (Badosa et al. 2014).

References

**Text S2. Heterogeneous chemistry calculations**

We test the impact of heterogeneous chemistry on aerosol mass loss within the smoke plume. We performed a simple calculation of OH molecules collision to the surface of a single particle ranging from 1 nm to 1 μm size in diameter. The following parameters assumed for the calculations:

- OH diffusivity = 3.5e-5 [m2 s-1]
- Constant OH concentration varied from 1e5 to 5e7 [molecules cm-3]
- Molecular weight of organics = 200 [g mol-1]
- Density of organics = 1.4 [g cm-3]
- Total run time = 3 [hours]

As an upper bound calculation, we assume each collision results in removing an organic molecule on the surface of the particle (assumed to be 200 amu), fragmenting and removing the molecule from the particle. The fragmentation products are not assumed to participate in further reaction. Figure S23a shows the resulting final:initial mass ratios after four hours of aging, indicating that for all aerosol sizes captured in this study (>10 nm) and under a range of OH concentrations, >90% of the aerosol mass remains. As a lower bound, we also include a case in which only 10% of all OH collisions result in a mass loss of 200 amu (Figure S23c).(Slade and Knopf, 2013)

Badosa, J., Wood, J., Blanc, P., Long, C. N., Vuilleumier, L., Demengel, D. and Haeffelin, M.: Solar irradiances measured using SPN1 radiometers: uncertainties and clues for development, Atmospheric Measurement Techniques, 7, 4267–4283, 2014.

Bahreini, R., Ervens, B., Middlebrook, a. M., Warneke, C., de Gouw, J. a., DeCarlo, P.F., Jimenez, J.L., Brock, C. a., Neuman, J. a., Ryerson, T.B., Stark, H., Atlas, E., Brioude, J., Fried, A., Holloway, J.S.,

Peischl, J., Richter, D., Walega, J., Weibring, P., Wollny, a. G., and Fehsenfeld, F.C.: Organic aerosol formation in urban and industrial plumes near Houston and Dallas, Texas. J. Geophys. Res., 114:D00F16, 2009.

Kleinman, L. I., Sedlacek, A. J., III, Adachi, K., Buseck, P. R., Collier, S., Dubey, M. K., Hodshire, A. L., Lewis, E., Onasch, T. B., Pierce, J. R., Shilling, J., Springston, S. R., Wang, J., Zhang, Q., Zhou, S. and Yokelson, R. J.: Rapid Evolution of Aerosol Particles and their Optical Properties Downwind of Wildfires in the Western U.S, Aerosols/Field Measurements/Troposphere/Physics (physical properties and processes), doi:10.5194/acp-2020-239, 2020.

Long, C. N., Bucholtz, A., Jonsson, H., Schmid, B., Vogelmann, A. and Wood, J.: A Method of Correcting for Tilt from Horizontal in Downwelling Shortwave Irradiance Measurements on Moving Platforms, The Open Atmospheric Science Journal, 4(1), 78–87, doi:10.2174/1874282301004010078, 2010.

Slade, J. H. and Knopf, D. A.: Heterogeneous OH oxidation of biomass burning organic aerosol surrogate compounds: assessment of volatilisation products and the role of OH concentration on the reactive uptake kinetics, Phys. Chem. Chem. Phys., 15(16), 5898–5915, 2013.

Wang, J., Pikridas, M., Spielman, S. R. and Pinterich, T.: A fast integrated mobility spectrometer for rapid measurement of sub-micrometer aerosol size distribution, Part I: Design and model evaluation, J. Aerosol Sci., 108, 44–55, 2017.

Table S1. Flight description table.

| Flight name, date | Number of sets of pseudo-Lagrangian transects | Fire name | Fuel[1] | Missing data[2] |
|---|---|---|---|---|
| '726a', 07-26-2013 | 2 | Mile Marker 28 | grasslands, shrub brush, timber, and timber litter | |
| '730a', 07-30-2013 | 1 | Colockum Tarps | grass, trees | |
| '730b', 07-30-2013 | 2 | Colockum Tarps | grass, trees | |
| '809a', 08-09-2013 | 1 | Colockum Tarps | grass, trees | $NO_x$ |
| '821b', 08-21-2013 | 1 | Government Flats | | $O_3$ |

[1]When known
[2]Instruments relevant to this study

Table S2. Calculated $R_{\Delta OA, initial}$ and $R_{age}$ values for $\Delta OA/\Delta CO$, $\Delta f_{60}$, $\Delta f_{44}$ $\Delta H/\Delta C$, $\Delta O/\Delta C$, $\Delta N/\Delta CO$, and $D_p$ when one flight is left out of the statistical analysis. We include the original R values as the first row for comparison. Red values indicate that the correlation has improved compared to all flights in the statistical analysis (closer to ±1). Blue values indicate that the correlation has worsened (closer to 0) compared to all flights in the statistical analysis. Black values denote no change in the correlation compared to all flights in the statistical analysis. Note that for flights '726a' and '730b' both sets of Lagrangian transects have been left out.

| $\Delta OA/\Delta CO$ | | |
|---|---|---|
| Flight left out, date | Resulting $R_{\Delta OA, initial}$ | Resulting $R_{age}$ |
| None | +0.02 | +0.03 |
| '726a', 07-26-2013 | +0.12 | 0.0 |
| '730a', 07-30-2013 | +0.02 | +0.07 |
| '730b', 07-30-2013 | +0.17 | 0.0 |
| '809a', 08-09-2013 | -0.25 | +0.02 |
| '821b', 08-21-2013 | +0.05 | +0.03 |
| $\Delta f_{60}$ | | |
| Flight left out, date | Resulting $R_{\Delta OA, initial}$ | Resulting $R_{age}$ |
| None | +0.43 | -0.26 |
| '726a', 07-26-2013 | +0.58 | -0.38 |
| '730a', 07-30-2013 | +0.39 | -0.37 |
| '730b', 07-30-2013 | +0.52 | -0.19 |
| '809a', 08-09-2013 | +0.3 | -0.21 |
| '821b', 08-21-2013 | +0.4 | -0.26 |

| $\Delta f_{44}$ | | |
|---|---|---|
| Flight left out, date | Resulting $R_{\Delta OA, initial}$ | Resulting $R_{age}$ |
| None | -0.55 | +0.5 |
| '726a', 07-26-2013 | -0.63 | +0.4 |
| '730a', 07-30-2013 | -0.62 | +0.54 |
| '730b', 07-30-2013 | -0.45 | +0.46 |
| '809a', 08-09-2013 | -0.54 | +0.54 |
| '821b', 08-21-2013 | -0.42 | +0.57 |
| **$\Delta H/\Delta CO$** | | |
| Flight left out, date | Resulting $R_{\Delta OA, initial}$ | Resulting $R_{age}$ |
| None | -0.04 | -0.06 |
| '726a', 07-26-2013 | -0.04 | -0.12 |
| '730a', 07-30-2013 | -0.13 | -0.2 |
| '730b', 07-30-2013 | 0.0 | -0.16 |
| '809a', 08-09-2013 | 0.02 | -0.01 |
| '821b', 08-21-2013 | -0.01 | -0.05 |
| **$\Delta O/\Delta CO$** | | |
| Flight left out, date | Resulting $R_{\Delta OA, initial}$ | Resulting $R_{age}$ |
| None | -0.45 | +0.56 |
| '726a', 07-26-2013 | -0.54 | +0.46 |
| '730a', 07-30-2013 | -0.52 | +0.55 |
| '730b', 07-30-2013 | -0.21 | +0.54 |
| '809a', 08-09-2013 | -0.5 | +0.61 |
| '821b', 08-21-2013 | -0.32 | +0.63 |
| **$\Delta N/\Delta CO$** | | |

| Flight left out, date | Resulting $R_{\Delta OA,\ initial}$ | Resulting $R_{age}$ |
| --- | --- | --- |
| None | -0.03 | -0.27 |
| '726a', 07-26-2013 | -0.03 | -0.13 |
| '730a', 07-30-2013 | -0.03 | -0.3 |
| '730b', 07-30-2013 | -0.21 | -0.43 |
| '809a', 08-09-2013 | -0.07 | -0.2 |
| '821b', 08-21-2013 | 0.0 | -0.37 |

$$\overline{D_p}$$

| Flight left out, date | Resulting $R_{\Delta OA,\ initial}$ | Resulting $R_{age}$ |
| --- | --- | --- |
| None | -0.15 | +0.53 |
| '726a', 07-26-2013 | -0.18 | +0.43 |
| '730a', 07-30-2013 | -0.17 | +0.57 |
| '730b', 07-30-2013 | +0.19 | +0.63 |
| '809a', 08-09-2013 | -0.28 | +0.52 |
| '821b', 08-21-2013 | -0.18 | +0.52 |

Table S3. Fit coefficients *a, b,* and *c* for the fits shown in Fig. 3 , equation 4. The units of *a* are (metric); the units of *b* are (metric)/hr, and the units of *c* are (metric), where (metric) = the units of $\Delta f_{60}$, $\Delta f_{44}$, $\Delta$O/$\Delta$C, or $\overline{D_p}$ , respectively.

| Metric | *a* | *b* | *c* |
|---|---|---|---|
| $\Delta f_{60}$ | 2.8e-03 | -6.4e-04 | 4.7e-03 |
| $\Delta f_{44}$ | -1.1e-02 | 5.8e-03 | 4.4e-02 |
| $\Delta$O/$\Delta$C | -3.6e-02 | 2.6e-02 | 0.24 |
| $\overline{D_p}$ | -1.5 | 10 | 150 |

Table S4. Fit coefficients *a, b,* and *c* for the fits shown in Fig. S28 , equation 5. The units of *a* are (metric); the units of *b* are (metric)/hr, and the units of *c* are (metric), where (metric) = the units of $\Delta f_{60}$, $\Delta f_{44}$, $\Delta$O/$\Delta$C, or $\overline{D_p}$ , respectively.

| Metric | *a* | *b* | *c* |
|---|---|---|---|
| $\Delta f_{60}$ | 0.14 | -6.6e-02 | -5.3 |
| $\Delta f_{44}$ | -0.14 | 0.11 | -2.9 |
| $\Delta$O/$\Delta$C | -7.3e-02 | 6.1e-02 | -1.3 |
| $\overline{D_p}$ | -6.3e-03 | 4.0e-02 | 5.1 |

Table S5. Fit coefficients $a$, $b$, and $c$ for the fits shown in Fig. S29 , equation 4 (but with $\Delta N_{initial}$ in place of $\Delta OA_{initial}$). The units of $a$ are (metric); the units of $b$ are (metric)/hr, and the units of $c$ are (metric), where (metric) = the units of $\Delta f_{60,}$ $\Delta f_{44}$, $\Delta O/\Delta C$, or $\overline{D_p}$ , respectively.

| Metric | $a$ | $b$ | $c$ |
|---|---|---|---|
| $\Delta f_{60}$ | 2.0e-03 | -5.4e-04 | -1.5e-03 |
| $\Delta f_{44}$ | -1.1e-02 | 5.3e-03 | 8.4e-02 |
| $\Delta O/\Delta C$ | -4.1e-02 | 2.4e-02 | 0.4 |
| $\overline{D_p}$ | -3.5 | 10 | 160 |

[Figure]

[Figure]

Figure S1. The flight path for flight '730b', colored by the FIMS total number concentration. The red dots are MODIS fire/thermal anomalies. The black star indicates the approximate center of the fire and the black dashed line indicates the approximate centerline of the plume, estimated by the number concentration.

[Figure]

[Figure]

Figure S2. The flight path for '726a'. Top two panels: t̶The legs used in this study are colored by each ΔCO percentile bin used in the main text analyses. The green traces indicate the locations of the lowest 10% of CO, used to compute averaged backgrounds for this flight. Bottom two panels: the flight track colored by time since take-off in minutes. The numbers indicate the leg numbers as identified in the BBOP database. There were two complete flight paths for this day. The red dots are MODIS fire/thermal anomalies. The black star indicates the approximate center of the fire and the black dashed line indicates the approximate centerline of the plume, estimated by the number concentration.

[Figure]

[Figure]

Figure S3. The flight path for '730a'. Top panel: tThe legs used in this study are colored by each ΔCO percentile bin used in the main text analyses. The green traces indicate the locations of the lowest 10% of CO, used to compute averaged backgrounds for this flight. Bottom panel: the flight track colored by time since take-off in minutes. The numbers indicate the leg numbers as identified in the BBOP database. The red dots are MODIS fire/thermal anomalies. The black star indicates the approximate center of the fire and the black dashed line indicates the approximate centerline of the plume, estimated by the number concentration.

[Figure]

[Figure]

Figure S4. The flight path for '730b'. Top two panels: tThe legs used in this study are colored by each ΔCO percentile bin used in the main text analyses. The green traces indicate the locations of the lowest 10% of CO, used to compute averaged backgrounds for this flight. Bottom two panels: the flight track colored by time since take-off in minutes. The numbers indicate the leg numbers as identified in the BBOP database. There were two complete flight paths for this flight. The red dots are MODIS fire/thermal anomalies. The black star indicates the approximate center of the fire and the black dashed line indicates the approximate centerline of the plume, estimated by the number concentration.

[Figure]

[Figure]

Figure S5. The flight path for '809a'. Top panel: tThe legs used in this study are colored by each ΔCO percentile bin used in the main text analyses. The green traces indicate the locations of the lowest 10% of CO, used to compute averaged backgrounds for this flight. Bottom panel: the flight track colored by time since take-off in minutes. The numbers indicate the leg numbers as identified in the BBOP database. The Worldview image for this day had clouds over the fire location at the time of the satellite passover. Thus we estimate a fire center using Worldview and MODIS images for this region on the previous day (8-08-2013) (light green star) and the following day (8-10-2013) (salmon-colored star). The black star indicates our estimated the approximate center of the fire on 8-09-2013 and the black dashed line indicates the approximate centerline of the plume, estimated by the number concentration.

[Figure]

[Figure]

Figure S6. The flight path for '821b'. Top panel: tThe legs used in this study are colored by each ΔCO percentile bin used in the main text analyses. The green traces indicate the locations of the lowest 10% of CO, used to compute averaged backgrounds for this flight. Bottom panel: the flight track colored by time since take-off in minutes. The numbers indicate the leg numbers as identified in the BBOP database. The red dots are MODIS fire/thermal anomalies. The black star indicates the approximate center of the fire and the black dashed line indicates the approximate centerline of the plume, estimated by the number concentration.

[Figure]

[Figure]

Figure S7. Number size distribution data, dN/dlogD$_p$, from the FIMS; CO (white solid line); and total short wave (SW) irradiance (black dots) data for the '726a' flight. The dotted dashed line indicates CO=150 ppb, our cutoff for in-plume/out-of-plume. The second set of Lagrangian transects for this flight start at the plume at approximately 86 minutes into the flight.

[Figure]

[Figure]

Figure S8. Number size distribution data, dN/dlogD$_p$, from the FIMS; CO (white solid line); and total short wave (SW) irradiance (black dots) data for the '730a' flight. The dotted dashed line indicates CO=150 ppb, our cutoff for in-plume/out-of-plume.

[Figure]

[Figure]

Figure S9. Number size distribution data, dN/dlogD$_p$, from the FIMS; CO (white solid line); and total short wave (SW) irradiance (black dots) data for the '730b' flight. The dotted dashed line indicates CO=150 ppb, our cutoff for in-plume/out-of-plume. For this figure, the top panel contains all of the first Lagrangian set of flight transects, and the bottom panel contains all of the second Lagrangian set of flight transects.

[Figure]

Figure S10. Number size distribution data, dN/dlogD$_p$, from the FIMS; CO (white solid line); and total short wave (SW) irradiance (black dots) data for the '809a' flight. The dotted dashed line indicates CO=150 ppb, our cutoff for in-plume/out-of-plume.

[Figure]

[Figure]

Figure S11. Number size distribution data, dN/dlogD$_p$, from the FIMS; CO (white solid line); and total short wave (SW) irradiance (black dots) data for the '821b' flight. The dotted dashed line indicates CO=150 ppb, our cutoff for in-plume/out-of-plume.

[Figure]

Figure S12. FIMS data for '809a' for the two legs that ~overlap (Figure S5) for the 51, 106, and 219 nm size bins. The solid line is from the plane flying north to south (right to left in this figure) and the dashed line is from the plane flying south to north (left to right in this figure). In the absence of FIMS measurement artifacts, we expect these two lines to roughly match each other. Each y axis is number in bin.

[Figure]

[Figure]

Figure S13. Same as Figure 2 but using only the first 50% of data for each leg of the FIMS and CO data for panels f-g.

[Figure]

[Figure]

Figure S14. Aerosol properties for the first set (left-hand column) and second set (right-hand column) of pseudo-Lagrangian transects from flight '726a' (a-b) ΔOA/ΔCO (right y-axis) and ΔrBC/ΔCO (left y-axis), (c-d) $\Delta f_{60}$ (right y-axis) and $\overline{\Delta f_{44}}$ (left y-axis), (e-f) ΔH/ΔC (right y-axis) and ΔO/ΔC (left y-axis), (g-h) ΔN/ΔCO, and (i-j) $\overline{D_p}$ against physical age. For each transect, the data is divided into edge (the lowest 5-15% of ΔCO data; red points), core (90-100% of ΔCO data; blue points), and intermediate regions (15-50% and 50-90% of ΔCO data; light green and dark green points). ΔrBC/ΔCO is shown in log scale and the x-axis for the right-hand column has been shifted backwards to improve clarity. Note that the left-hand and right-hand columns do not always have the same y-axis limits.

[Figure]

[Figure]

Figure S15. Aerosol properties for the set of pseudo-Lagrangian transects from flight '730a' (a) ΔOA/ΔCO (right y-axis) and ΔrBC/ΔCO (left y-axis), (b) $\Delta f_{60}$ (right y-axis) and $\Delta f_{44}$ (left y-axis), (c) ΔH/ΔC (right y-axis) and ΔO/ΔC (left y-axis), (d) ΔN/ΔCO, and (e) $\overline{D_p}$ against physical age. For each transect, the data is divided into edge (the lowest 5-15% of ΔCO data; red points), core (90-100% of ΔCO data; blue points), and intermediate regions (15-50% and 50-90% of ΔCO data; light green and dark green points). ΔrBC/ΔCO is shown in log scale to improve clarity.

[Figure]

Figure S16. Aerosol properties for the first set (left-hand column) and second set (right-hand column) of pseudo-Lagrangian transects from flight '730b' (a-b) ΔOA/ΔCO (right y-axis) and

$\Delta$rBC/$\Delta$CO (left y-axis), (c-d) $\Delta f_{60}$ (right y-axis) and $\Delta f_{44}$ (left y-axis), (e-f) $\Delta H/\Delta C$ (right y-axis) and $\Delta O/\Delta C$ (left y-axis), (g-h) $\Delta N/\Delta CO$, and (i-j) $\overline{D_p}$ against physical age. For each transect, the data is divided into edge (the lowest 5-15% of $\Delta CO$ data; red points), core (90-100% of $\Delta CO$ data; blue points), and intermediate regions (15-50% and 50-90% of $\Delta CO$ data; light green and dark green points). $\Delta$rBC/$\Delta$CO is shown in log scale to improve clarity. Note that the left-hand and right-hand columns do not always have the same y-axis limits.

[Figure]

[Figure]

Figure S17. Aerosol properties for the set of pseudo-Lagrangian transects from flight '809a' (a) ΔOA/ΔCO (right y-axis) and ΔrBC/ΔCO (left y-axis), (b) $\Delta f_{60}$ (right y-axis) and $\Delta f_{44}$ (left y-axis), (c) ΔH/ΔC (right y-axis) and ΔO/ΔC (left y-axis), (d) ΔN/ΔCO, and (e) $\overline{D_p}$ against physical age. For each transect, the data is divided into edge (the lowest 5-15% of ΔCO data; red points), core (90-100% of ΔCO data; blue points), and intermediate regions (15-50% and 50-90% of ΔCO data; light green and dark green points). ΔrBC/ΔCO is shown in log scale and the x-axis for the right-hand column has been shifted backwards to improve clarity.

[Figure]

[Figure]

Figure S18. Aerosol properties for the set of pseudo-Lagrangian transects from flight '821b' (a) $\Delta OA/\Delta CO$ (right y-axis) and $\Delta rBC/\Delta CO$ (left y-axis), (b) $\Delta f_{60}$ (right y-axis) and $\Delta f_{44}$ (left y-axis), (c) $\Delta H/\Delta C$ (right y-axis) and $\Delta O/\Delta C$ (left y-axis), (d) $\Delta N/\Delta CO$, and (e) $\overline{D_p}$ against physical age. For each transect, the data is divided into edge (the lowest 5-15% of $\Delta CO$ data; red points), core (90-100% of $\Delta CO$ data; blue points), and intermediate regions (15-50% and 50-90% of $\Delta CO$ data; light green and dark green points). $\Delta rBC/\Delta CO$ is shown in log scale and the x-axis for the right-hand column has been shifted backwards to improve clarity.

[Figure]

[Figure]

Figure S19. Various normalized parameters as a function of age for the 7 sets of pseudo-Lagrangian transects. Separate lines are shown for the edges (lowest 5-15% of ΔCO; dashed lines) cores (highest 90-100% of ΔCO; solid lines), and intermediate regions (15-50% and 50-90%; dotted and dashed-dot lines). (a) ΔOA/ΔCO, (b) $\Delta f_{60}$, (c) $\Delta f_{44}$, (d) ΔH/ΔC, (e) ΔO/ΔC, (f) $\Delta N_{40\text{-}262 \text{ nm}}$/ΔCO, and (g) $\overline{D_p}$ between 40-262 nm against physical age for all flights, colored by $\Delta OA_{initial}$. Some flights have missing data. Also provided is the Spearman correlation coefficient, R, between each variable and $\Delta OA_{initial}$ and physical age for each variable. Note that panels (a), (d), and (g) have a log y-axis.

[Figure]

[Figure]

Figure S20. Various normalized parameters as a function of age for the 7 sets of pseudo-Lagrangian transects. Separate lines are shown for the edges (lowest 5-15% of ΔCO; dashed lines) and cores (highest 90-100% of ΔCO; solid lines). (a) ΔOA/ΔCO, (b) $\Delta f_{60}$, (c) $\Delta f_{44}$, (d) ΔH/ΔC, (e) ΔO/ΔC, (f) $\Delta N_{40\text{-}262\,nm}$/ΔCO, and (g) $\overline{D_p}$ between 40-262 nm against physical age for all flights, colored by $\Delta OA_{initial}$. Some flights have missing data. Also provided is the Spearman correlation coefficient, R, between each variable and $\Delta OA_{initial}$ and physical age for each variable. Note that panels (a), (d), and (g) have a log y-axis. This figure is identical to Figure 2 but uses an in-plume CO cutoff of 200 ppb.

[Figure]

[Figure]

Figure S21. Various normalized parameters as a function of age for the 7 sets of pseudo-Lagrangian transects. Separate lines are shown for the edges (lowest 5-25% of ΔCO; dashed lines) and cores (highest 75-100% of ΔCO; solid lines). (a) ΔOA/ΔCO, (b) $\Delta f_{60}$, (c) $\Delta f_{44}$, (d) ΔH/ΔC, (e) ΔO/ΔC, (f) $\Delta N_{40\text{-}262\ nm}$/ΔCO, and (g) $\overline{D_p}$ between 40-262 nm against physical age for all flights, colored by $\Delta OA_{initial}$. Some flights have missing data. Also provided is the Spearman correlation coefficient, R, between each variable and $\Delta OA_{initial}$ and physical age for each variable. Note that panels (a), (d), and (g) have a log y-axis. ▬This figure is identical to Figure 2 but uses different ΔCO percentile widths.

[Figure]

[Figure]

Figure S22. Various normalized parameters as a function of age for the 7 sets of pseudo-Lagrangian transects. Separate lines are shown for the edges (lowest 5-15% of ΔCO; dashed lines) and cores (highest 90-100% of ΔCO; solid lines). (a) ΔOA/ΔCO, (b) $\Delta f_{60}$, (c) $\Delta f_{44}$, (d) ΔH/ΔC, (e) ΔO/ΔC, (f) $\Delta N_{40\text{-}262\ nm}$/ΔCO, and (g) $\overline{D_p}$ between 40-262 nm against physical age for all flights, colored by ΔOA$_{initial}$. Some flights have missing data. Also provided is the Spearman correlation coefficient, R, between each variable and ΔOA$_{initial}$ and physical age for each variable. Note that panels (a), (d), and (g) have a log y-axis. This figure is identical to Figure 2 except that it uses the location of the lowest 25% of CO data to determine the background concentrations of each species.

[Figure]

Figure S23. Calculated (final aerosol mass):(initial aerosol mass) ratios for mass loss through heterogeneous chemistry over a range of aerosol diameters and OH concentrations. As an upper-bound case, (a) it is assumed that for each OH collision, 200 amu of mass is lost. As a middle-bound, (b) it is assumed that 50% of OH collisions result in a 200 amu mass loss. As a more-realistic loss rate, (c) assumes that 10% of all OH collisions result in an 200 amu mass loss. See SI text S2 for more details.

[Figure]

[Figure]

Figure S24. Raw $f_{60}$ data for each flight along each transect included in this study. The titles indicate the flight. The black color indicates the earliest transect, with increasingly lighter colors indicating increasingly downwind transects. The centerline was estimated from the number size distribution and the estimated center of the fire (Figures S1-S6).

[Figure]

[Figure]

Figure S25. Raw $f_{44}$ data for each flight along each transect included in this study. The titles indicate the flight. The black color indicates the earliest transect, with increasingly lighter colors indicating increasingly downwind transects. The centerline was estimated from the number size distribution and the estimated center of the fire (Figures S1-S6).

[Figure]

Figure S26. Total in-plume shortwave (SW) irradiance for each flight along each transect included in this study. The titles indicate the flight. The black color indicates the earliest transect, with increasingly lighter colors indicating increasingly downwind transects. The centerline was estimated from the number size distribution and the estimated center of the fire (Figures S1-S6).

[Figure]

Figure S27. The Van Krevelen diagram of ΔH/ΔC versus ΔO/ΔC for all points in the 7 sets of pseudo-Lagrangian transects, colored by $\Delta OA_{initial}$.

[Figure]

[Figure]

Figure S28. Measured versus predicted (a) $\Delta f_{60}$, (b) $\Delta f_{44}$, and (c) $\overline{D_p}$ between 40-262 nm, using the equation $ln(X) = a\ ln(\Delta OA_{initial}) + b\ ln(Physical\ age) + c$ (Eq. 5) where $X=\Delta f_{60}$, $\Delta f_{44}$, or $\overline{D_p}$. The values of a, b, and c are provided in Table S4. The Pearson and Spearman coefficients of determination ($R^2_p$ and $R^2_s$, respectively) are provided in each panel, along with the normalized mean bias (NMB) and normalized mean error (NME).  Included in the fit and figure are all four regions within the plume (the 5-15%, 15-50%, 50-90%, and 90-100% of $\Delta CO$), all colored by the mean $\Delta OA_{initial}$ of each $\Delta CO$ percentile range.

[Figure]

[Figure]

Figure S29. Measured versus predicted (a) $\Delta f_{60}$, (b) $\Delta f_{44}$, and (c) $\overline{D_p}$ between 40-300 nm, using the equation $X = a\, log_{10}(\Delta N_{initial}) + b\,(Physical\ age) + c$ where X=$\Delta f_{60}$, $\Delta f_{44}$, or $\overline{D_p}$ where X=$\Delta f_{60}$, $\Delta f_{44}$, or $\overline{D_p}$. Note that the fit here is the same as that in Eq. 2 except that $\Delta N_{initial}$ replaces $\Delta OA_{initial}$. The values of a, b, and c are provided in Table S5. The Pearson and Spearman coefficients of determination ($R^2_p$ and $R^2_s$, respectively) are provided in each panel, along with the normalized mean bias (NMB) and normalized mean error (NME).  Included in the fit and figure are all four regions within the plume (the 5-15%, 15-50%, 50-90%, and 90-100% of $\Delta CO$), all colored by the mean $\Delta OA_{initial}$ of each $\Delta CO$ percentile range.

---

## Referee Report (RR1)

**Review of "Dilution impacts on smoke aging: Evidence in BBOP data"**

**By Anna L. Hodshire et al.**

**Anonymous Reviewer**

*Summary:*

This manuscript derives empirical relationships for aerosol chemical age, number concentration, and mean mobility diameter which are useful for modeling the climate impact of biomass-burning aerosols. New insight from the BBOP measurement campaign is used to show that the rate of chemical aging of aerosol is affected by the "concentration" of the plume, presumably through decreasing photolytic rates with increased aerosol optical depth, and plume size, presumably because the core of larger plumes and are more protected from mixing with clean ambient air.

Ultimately, this is an interesting study but has a number of key assumptions that need to be further investigated or justified. Although the authors have weakened their language since the original draft due to comments from reviewers, they have not addressed the underlying concerns of the reviewers. This manuscript will require major revisions prior to considering publication.

*General Comments:*

1. In general, the language has improved but is still not very precise and there are grammatical issues.
2. There is a lack of details regarding SP-AMS measurements (as well as description of other instruments). There are only scattered references to the operating conditions/settings used for measuring OA in these plumes. Although the instrumentation is fully described in another manuscript, there is a minimum amount of information required for the SP-AMS measurement: vaporizers equipped and modes used (switching between modes, temperature of thermal vaporizer), calibration, description of CE determination, ToF mode (HR-ToF, C-ToF, V-mode, W-mode), MS sampling timing (Open, closed, PToF, ePToF, pulser period, etc)

   There also needs to be a description of the mass spectra analysis. What software was used. Are you reporting UMR or HR results? Is f60 based on m/z=60 or the specific ion $C_2H_4O_2^+$? How is gas phase subtracted? Assuming constant [$CO_2$] gas phase concentrations of 400 ppm?

   Since the vaporizer modes were switched (presumably intermittently) it is concerning how the authors choose to combine the data from the different modes. These modes

measure inherently different components of the aerosol mass and fractionate the molecules in different ways. While Lee et al (2020) show that molecule fractionation in the different vaporizer modes is similar for the C2H4O2+ ion (used to calculate f60, although that is not stated by the authors), the fractionation is significantly different for the organic fraction of CO2+ (used to calculate f44) (see also Onasch et al 2012, Canagaratna et al 2015, etc).

3. Data are binned by physical age and further into edge-vs-core (such as shown in Figure 1). Each binned datum represents multiple measurements and therefore, in figure 1 and 2, should include error bars representing the variance of those measurements. This could be independent of measurement uncertainty, but would be better if it did include propagated instrument uncertainty.

4. A key assumption of the authors is that $\Delta OA_{initial}$, $\Delta f60$, and $\Delta f44$ can be used to identify dilution of the plume. However, these parameters are known to vary in primary emissions.

   In the manuscript, the authors support the assumption regarding $\Delta OA_{initial}$ with the measured $\Delta f60$ and $\Delta f44$. However, these parameters are all more likely related to variations in POA between fires and within a fire as fire conditions change. The $\Delta f60$ and $\Delta f44$ measurements are the only support the authors provide for their main conclusion.

   In the author's revisions, they have tried to further justify this assumption by making a flawed argument that their interpretation is only invalid if f60 and f44 covary with OA emissions. First this argument is flawed, as $\Delta f60_{initial}$ and $\Delta f44_{initial}$ are more reasonably attributed to differences in POA. Second, it has actually been shown that f60 and f44 of POA can covary with OA emission factors (see Corbin et al 2015; Ortega et al 2015; and Lee et al 2010).

5. Interpretation of Spearman's correlation coefficients and the strength of these coefficients is hindered by the authors choice to combine all data from all flights together for their regressions without normalization despite showing that both aerosol age and emission factors affect the parameter of interest (e.g. $\Delta f_{60}$). This multi-variate dependence is even stated several times by the authors. For example, with 1 exception, all transect sets predictably show that $\Delta f_{60}$ decreases with physical age, but because different transect sets started with different $f_{60}$ values the combined data set does not monotonically decrease with physical age and the regression results in a weak relationship ($R_{Spearman}$ = -0.26).

   This is an example of where the authors should rethink their analytical approach but have instead weakened the language of their results.

   There are several possibilities that the authors could consider. Continuing to use $\Delta f_{60}$ as an example, the authors could:

- Normalizing the data to the initial measured value (e.g. $\Delta f_{60} - \Delta f_{60}initial$) prior to combining the data. This allows you to remove the processes driving variability in the initial $\Delta f_{60}$ (essentially the emission factor) so you can isolate the effects of physical age on $\Delta f_{60}$.
- A multivariate analysis with predictors of $\Delta OA_{initial}$ and physical age. You could do this in any programming language, but using Excel as an example you would use the Data>Data_Analysis>regressions gui. It appears that you use this in your model in section 3.1 in equation 4. Since you did this, why do you even show the single-variable $R_{pearson}$ and $R_{spearman}$ values?
- Lastly, you could analyze each transect set separately to get a $R_{spearman}$ value of $\Delta f_{60}$ versus physical age and then average those values together. If this approach is chosen, then averaging should weighted by the number of transects in the transect set. Also, you should use a jack-knife-like approach, repeating the averaging by systematically excluding 1-2 transects sets to see how dependent the results are on any individual set.

Currently, the analytical approach is inappropriate and therefore should not be used as support for their conclusions, regardless of the strength of language chosen to describe the empirical relationships.

7) SI section: The supplement has improved with more detail but is still lacking. More detail is needed describing the instrument set up, even if it is described fully in another paper.

The heterogeneous chemistry calculations needs a description of the calculation and justification for the methodology used. The only information provided is that it is a "simple calculation" and a list of what the parameters are.

*Specific Comments:*

L31: Here and elsewhere, are you calculating the mass mean mobility diameter or number mean mobility diameter?

L38: "…undergone more decreases in a marker for primary biomass burning organic aerosol." This is an awkward statement

L41-44: "Smoke from biomass burning… influencing… as well as the health of smoke-impacted communities". "Smoke-impacted" is redundant.

L45: Dilution is a process which is a central theme of the manuscript. It should have a proper description of what that process is. I suggest something like, "Dilution is the process where the plume mixes with clean background air, reducing concentration of fire emitted aerosols and gases". The current statement, "Dilution through entrainment…" is not explicit.

L50-52: Lacks explanation of why large plumes dilute slower. Since this is so important to the story, it should have a better description. Currently, it is just stated that they do. Something like "…cores of larger plumes are protected from dilution due the physical distance from background air…" Citations for this are Garofalo et al (2019) and Lee et al (2020).

L86-87: "… evaporation of vapors." Should be "evaporation of more volatile compounds."

L88: "; plumes with higher concentrations will undergo more coagulation…" Are you referring to number concentration, mass concentration, or both?

L101: "… differences in aerosol loading serve as a proxy for differences in dilution rates…" Do you mean rates or amount of dilution prior to first measurement? I have provided other comments in more detail regarding this assumption.

L103: "…given initial plume mass and physical age…" "mass" should be "OA mass concentration".

L115: More description needs to be provided regarding the settings for the SP-AMS.

Was it equipped with a tungsten thermal vaporizer? If so, what temperature was the thermal vaporizer set to? HR-ToF, C-ToF, L-ToF, quadrupole? ToF set to V-mode or W-mode? How was data analyzed? Are you reporting UMR or HR results? Is f60 the levoglucosan fraction (i.e. fC2H4O2 as is discussed by Corbin et al 2015 and Lee et al 2020) or based on the UMR m/z=60 organic fragment after subtracting C5 contributions (also m/z=60, see Cubison et al 2011 and Lee et al 2010). What was the MS timing? Open vs closed timing? PToF or ePToF mode? Pulse period, sampling Hz? What m/z range was scanned?

L116: "PM1 aerosol masses" should be "aerosol mass concentration of PM1 (sub-micron particulates)…"

L119-121: How was collection efficiency determined?

L119-121: It looks like the laser vaporizer was switched between on and off. How frequently was the laser vaporizer switched? Is the data presented in this manuscript with the laser on, off, or both?

These measure inherently different attributes of PM1 that may not be directly comparable or combinable.

Also, because the laser vaporizer fractionates aerosols molecular species differently than the thermal vaporizer (Onasch et al 2012; Corbin et al 2014; Canagaratna et al 2015; Lee et al 2015; Lee et al 2020) single ions such as C2H4O2+ and Org44+ (used to calculate f60 and f44) CANNOT be compared or combined between modes.

L121-L122: "We do not attempt to characterize whether the collection efficiency of the SP-AMS changes as the aerosol ages"

Collection efficiency has been observed to change by a factor of 2 or more as BB POA grows in size and becomes more spherical (See Middlebrook et al 2012, Willis et al 2014, Corbin et al 2015 (ACP), Massoli et al 2015, Collier et al 2018). This change in CE has been observed to bias particles of different morphology/composition differently between different vaporizer modes (laser + oven versus oven-only), specifically affecting the CE of the laser more than the CE of the thermal vaporizer.

L123: "…CE has been recently observed to decrease with aging within a laboratory study…" should be "… decrease with increasing chemical age induced by UV light exposure and OH-equivalent to 10 photochemical days…"

This change in CE is likely irrelevant to this manuscript since the physical age of aerosols described in this study is generally less than 3 hours while the cited study compares CE over the equivalent of 10 days. As mentioned in the previous comment, there are a range of studies that have shown increases in CE as particles grow in size (which presumably also increase with age in the near-field during the particle growth phase) which is more relevant to this study.

L 124: "… no consistent evidence of changing CE in field studies exist yet."

There are lots of studies which show changing CE in the field. For example, see Collier et al (2018), Massoli et al (2015), and Middlebrook et al (2012) for examples of changing CE in field studies. Also see Willis et al (2014) and Corbin et al (2015, ACP) which report the same phenomena in laboratory studies.

L124-125: "We use the f60 and f44 fractional components…"

Here and elsewhere, f60 and f44 are referred to as if they are chemical species instead of parameters describing mass fractions.

For f50, the relevant OA species group is anhydrous sugars with the dominant species being levoglucosan ($C_6H_{10}O_5$) (Lee et al 2010; Cubison et al 2012). This species is indirectly observed by the SP-AMS as the fragment $C_2H_4O_2$ (m/z=60), a fragment of the levoglucosan molecule after the OA is vaporized and the vapors are ionized by a 70 eV electron supply. Similarly, OA observed at m/z 44 is the $CO_2$ fragment of, primarily, OOA after subtracting the gas [CO2] mass.

L128-129: "The f44 fractional component (arising from primarily CO2+…)"

This is another example of imprecise language. Suggest, "f44, the OA fractional component observed by the SP-AMS as the ion fragment CO2+, is a proxy for …" The current wording suggests that f44 is a fraction of something else which isn't specified (i.e. f44/Org).

L129: extra semicolon

L130-132: "…of semivolatile f60-containing species and addition of oxidized f44-containing…"
Another example of imprecise language using f60 and f44. The aerosols contain levoglucosan and anhydrous sugars, not f60, and OOA, not f44.

L135: "changes in O/C and H/C are also influenced by…"

These other processes also affect all of the other parameters discussed in this paper.

L140: "provided CO measurements" should be "measured CO concentration" or "measured CO mixing ratio".

L141: "An SPN1 radiometer provided total shortwave irradiance". The radiometer measured total shortwave irradiance, it did not provide or create the irradiance.

L143: "($\Delta$x)" refers to the smoke contribution and should be placed after "species X from smoke" otherwise $\Delta$x is implied to refer to the background value of x. Also, the smoke doesn't contribute species X, it is comprised of species X. The fire emits/contributes X.

L145-146: This sentence needs to be rewritten. Suggest something like "Variability of the normalized emission ratio ($\Delta$x/$\Delta$CO) along the lagrangian flight path implies production or removal of species X in the plume."

L147: "… average regional background for each species by using the lowest 10% of the CO data for…"

This statement reads as if you subtract a CO concentration from the number size distribution, OA, etc to get a background correction. It should say something like "… background values of X… were determined to correspond with time periods which displayed the lowest 10% of CO concentrations…"

L149: Should read "Mass concentrations of elemental O, H, and C were calculated…"

L152-153: It is not clear what you mean by this sentence.

L154: "inside and outside of the plume".

Does this refer to sampling time periods/locations corresponding lowest 10% of CO or <150 ppbv? Why would you change between these definitions of background?

L157-158: "We only consider data to be in-plume if the absolute CO>=150 ppbv, as comparisons of CO and the number concentration show that in-plume data has CO>150 ppbv and out of plume (background) data has CO < 150 ppbv."

Why did you change the definition of background from time/location corresponding to lowest 10% of CO to <150 ppbv?

What do you mean by "comparisons of CO and the number concentrations"? What independent metric are you defining as "background" here? If number concentration is used to define time/location of background air, why not use that instead of CO <10% or CO < 150 ppbv? The logic here is circular or incomplete.

L 162: "concentrations" should be singular.

L163: Should be "mobility" diameters.

L164: "…as the bulk of observed newly formed particles observed fell below 40 nm" Grammar.

How do you identify "newly formed particles" independent of the particle size? This implies that you observed newly formed particles >40 nm.

L165-166: Grammar in sentence structure.

L187-188: "The centerline…" This sentence needs to be rewritten for clarity.

L190: "…and this physical age is assumed to be constant across the transect, as the crossings took between 50-500 seconds."

While crossing the plume occurred in only a short time, were the transects always perfectly across flow? If not, then wouldn't the aerosol at different sampling times along a transect have different physical ages with larger uncertainty than just 50-500 seconds?

L195: missing comma

L201-202: "thinnest (least CO-dense)… thickest (most CO-dense)…" Use either of the commonly accepted nomenclature of "CO mixing ratio" or "CO concentration".

L207: missing a verb between "plumes" and "from"

L205-207: could the multiple peaks during a transect be explained by spatial variations in the plume structure, such as the core of the plume was higher in some areas than others causing the flight to dip below/above the core and then back into it?

L213: It is hard to tell what the variability in $\Delta BC/\Delta CO$ are since they are plotted on a log scale. They appear to vary by an order of magnitude, i.e. not constant as the authors suggest.

L213: I pointed this out in my general comments. Each dot in figure 1 is a single value that represents multiple measurements in space and time. How well does the value of any single datum in this figure represent the range of data it is derived from? You need to have error bars to show that variability.

L215: "for each transect" should be "each transect set".

L218-219: "…it is apparent that the 5-15 and 9-90 percentiles do show a separation…"

This statement cannot be verified or supported without some idea of error bars, either representing propagated measurement uncertainty, variability of binned data, or both.

L227-231: This is one of the key assumptions of the research, that the initial, background-corrected OA mass concentration can be used as a proxy for the degree of dilution of the plume. The authors provide no support for why $\Delta OA_{initial}$ would represent the degree of dilution even though this is the main storyline of their paper.

First, I do agree that the cores of larger plumes are likely protected from mixing with background air because of the distance between the core and the background air and it is well understood that some plume chemistry and mixing occurs very early after combustion+pyrolysis and prior to measurement. However, this needs to be presented differently. Start with the hypothesis that cores of larger plumes mix slower with background air and then use the observations to prove it by showing something like the rate of change in f60 and f44 as a function of physical age for plumes with different $\Delta OA_{initial}$.

The current presentation is problematic. Think of two hypothetical smoke plumes that are identical in terms of dilution, photolytic reaction rate etc and were measured at the same physical age, but the corresponding fires had different OA emission factors (say, flux of OA from fire B was twice that of fire A). Fire B would have ~2x the measured $\Delta OA_{initial}$ compared to Fire A. This would instead be interpreted by the authors as having half the dilution of the plume from fire B instead of twice the OA emission.

This demonstrates that the author's assumption that $\Delta OA_{initial}$ is a proxy for plume dilution only makes sense if all fires measured emit OA at the same rate and concentration.

L228-229: "(as presumably larger, more intensely burning fires will have larger mass fluxes than smaller…)".

This assumption is false.

Larger and more intense fires do not necessarily correspond to higher emission rates. Emissions of OA depend on a number of factors other than fire intensity (I assume you mean temperature). Hotter, more intense fires (i.e. flaming stage) can burn more efficiently and actually emit less OA than cooler, smoldering fires (Akagi et al 2011; McMeeking et al 2009; May et al 2014). Corbin et al (2015) found that in laboratory burn experiments the vast majority of OA emissions occurred in the "starting phase" before the logs fully caught fire.

L243: missing a comma between items in the list.

L252: delete either "systematically" or "sequentially"

L251-257: See general comments regarding data analysis. The authors choose to combine all data together to determine the effects of aging and dilution on plume characteristics. However, they do not normalize the data in anyway or try another technique to separate these two effects. Normalizing a parameter to the first measured value (say f60) acknowledges that there are differences in f60 between plumes (maybe related to $\Delta OA_{initial}$) and would allow for analysis of temporal trends after emission. One result maybe that the photolytic age of the aerosol mass (as measured by f60 or f44) is slower plumes with higher $\Delta OA_{initial}$, i.e. there is less of a change/unit time of f60 or f44 or $\Delta N/\Delta CO$, etc.

Currently, the analysis is a regression comparing apples and oranges and the results are not meaningful.

L262-266: Include citations to Cubison et al (2011), Garofalo et al (2019), Forrister et al (2015), Lee et al (2020) for constant $\Delta OA/\Delta CO$ as plumes age.

L270-270: Containing text "…estimates of heterogeneous mass losses indicate that after three hours of aging for a range of OH concentrations and reactive uptake coefficients, over 90% of aerosol mass is anticipated to remain…"

This is the basis for which the authors interpret changing f44, f60 as relating to coagulation. However, this statement is only relevant to particle evolution after ~3 hours while nearly all of the observations occur within a physical age of 3 hours.

L281: "with more concentrated plumes". Be more specific by what you mean by more concentrated. Do you mean less diluted, higher mass concentrations of OA, higher CO mixing ratios?

L282-284: "(2) Differences in $\Delta f60$ and $\Delta f44$ for the nearest-to-source measurements indicate that evaporation and/or chemistry likely occurred before the time of these first measurements…"

It is well documented that the f60 and f44 of POA varies between fires (Cubison et al 2011; Jolleys et al 2015; Ortega et al 2013; McClure et al 2020). Since differences of POA emissions

can explain variability in Δf60 and Δf44 for the nearest-to-source measurements, variability of these parameters can NOT be used as evidence of chemistry/evaporation in the smoke plume without knowing the actual f60 and f44 values of fresh POA.

L282-284 "…(assuming that emitted Δf60 and Δf44 do not correlate with $\Delta OA_{initial}$; there is currently no evidence for this alternative hypothesis)."

There is actually a lot of evidence that f60 and f44 can correlate with OA emissions. In laboratory studies, the evolution of emissions as fires progress from starting-to-flaming-to-smoldering has shown that levoglucosan emissions occur primarily at the starting phase by combustion of hemicellulose material which is also when the majority of OA emissions occur (Corbin et al 2015). Ortega et al (2015) and Lee et al (2010) also observed increased values of f60 in lab burns with higher OA emission factors. These laboratory studies support observations of smoke in ambient troposphere (e.g. Lee et al 2010; Aiken et al 2009; Lee et al 2020).

As you note, it is hard to measure the f60 of POA in ambient smoke. However, the lifetime of levoglucosan in the free troposphere is much longer than the age of aerosol in this study (<3 hours) which is probably why there is only a weak trend to lower f60 values with increasing physical age. So, your measurements of Δf60 should be fairly representative of POA and your study (and your first point on lines 280-282 and repeated on line 319) are evidence that f60 is correlated with OA emission factors in wildfires.

L284: "Amounts" should be singular.

L291: Add citation Jolleys et al (2015).

L313-314: delete "tends to be fairly constant or slightly decreasing with physical age and". Saying that it is poorly correlated is enough.

L319-320: Evaporation does not happen from dilution. Evaporation will happen if the air is undersaturated (less than predicted vapor pressure of species X compared to equilibrium predicted by Henry's law). Here and elsewhere, please don't say that dilution causes evaporation, instead that dilution promotes evaporation.

L334: "NME is more variable…" Do you mean larger or higher? The NME is more variable between parameters, but that is meaningless.

L336: too many open brackets.

L337: What do you mean by "biomass burning modeling"? Are you referring to models of BB emissions, aerosol aging, fire spread?

L343-345: Since you present a mulit-variate analysis here, what was the point of the past several pages discussing single variable correlation coefficients? Especially after you show that

"Both physical age and $\Delta OA_{initial}$ appear to influence $\Delta f_{60}$, $\Delta f_{44}$, and $\Delta O/\Delta C$…" (Line 319 and a similar statement on L280-282).

L360-363: This statement needs more explanation and needs citations. Why would you expect plume regions with higher $\Delta OA_{initial}$ to have lower normalized number concentrations?

To a first order, I would expect the opposite. That higher number concentrations would be observed with higher $\Delta OA_{initial}$ because the OA vapor pressure is higher and this promotes new particle nucleation because vapors are more likely to collide with other vapor molecules to nucleate than with existing particles to condense on (Lim et al 2019; and work from Neil Donahue's group).

L371-372: Decreasing normalized number concentrations are ascribed to coagulation. This contradicts the model that changes in f60 and f44 are due to evaporation of solid particulate balanced by condensation of more oxidized-OA described on line 277-279.

L381-382: Awkward and redundant sentence.

L384-393: Discussion of nucleation-mode particles seems out of place here. Maybe move towards beginning of section 3.

Nucleation-mode particles are defined as 20-40nm. This needs a citation.

Earlier, a statement was made "bulk of observed newly formed particles observed fell below 40 nm" which implies that a fraction of newly formed particles were larger 40 nm. Was that a mis-statement? This would contradict your definition that nucleation-mode particles are 20-40 nm?

L388: "one day" should be one transect or transect set.

L405-406: Awkward or redundant sentence.

L408-409: "indicate that evaporation and/or chemistry has likely occurred before the time of initial measurement…" See previous comments questioning validity of this statement.

L437: Format of some citations need to be cleaned up.

Figure 1:
- See comments regarding adding error bars to show data variability of bin. There are a few points that are very different from the rest of the data set (such as in the $\Delta OA/\Delta CO$ and $\Delta N/\Delta CO$ datasets) which makes me think the single value representing the bin is inadequate.
- Please change $\Delta BC/\Delta CO$ to a linear scale

- Your values of f60 are pretty low for fresh BBOA. I am wondering if this is an issue with the SP-AMS settings or how the data was handled.
- Need units for Dp axis, "[nm]".

Figure 2:
- If you insist on combining all of the data together for a single regression, than you should not be drawing lines between points. Instead this should be a scatter plot with markers.
- Legend is inconsistent with figure 1. Either use "edge" or 5%<Δ[CO]<15%.
- Use of "[CO]" is inconsistent with text.
- Need units for Dp axis
- Caption says that panels (d) and (g) have log axis but are plotted on linear axis. Panels (a) and (f) are plotted on log axis (also in corresponding figures in SI).
- Font of "Dp" in caption is different than rest of fonts.
- I think you should also provide a scatter plot of the first measurement of these parameters as a function $\Delta OA_{initial}$.
- Needs error bars

Figure 3:
- Spearman's correlations are not needed here.

Supplemental Information:
"…electrical mobility as in SMPS…" Should be "…similar to the operating principle of the SMPS…"

"…when size distribution suggests that particles smaller than 10 nm contribute negligibly…" Neither the FIMS or CPC 3010 are efficient at counting <10nm Dm particles, so why would the existence of those particles cause differences between the two instruments?

"The SPAMS is thoroughly detailed in Kleinman et al. (2020)…" This still needs to be described here. At least summarizing the operating conditions of the SP-AMS. 1

"An SPN1 radiometer provided total shortwave irradiance…" It probably measured total shortwave irradiance, not "provided" or "created" the irradiance. This instrument needs to be described more. Maybe what exactly the measurements are and what they represent.

"… following parameters assumed for the calculation" missing a verb.

"Heterogeneous chemistry calculations:" There is no citation to justify the calculation. Is this a common methodology used? Has this methodology passed peer review?

Fig S1: Colorbar label is missing an "]"

Fig S7-S11: Why are x-axis on the top and bottom panels different scales?

Fig S13: should be moved to just before figure S19

Fig S19-S22: Caption mis-identifies which panels use a log scale.

Fig S24: Could you also plot the [CO] of each transect similar to S24-S26. I want to see that the absolute concentration is higher in the center of the plume than the edges and that the [CO] of the core decreases in each successive transect of the set to show dilution.

Fig S27: Need the 1:1, 1:2, and 0.5:1 line representing constant lines of oxidation. If it is arbitrary where the intersection of these lines is placed (as you cite from Heald et al 2010), then have an arbitrary intersection near the average of your data. The importance is the trends in H:C vs O:C. Alternatively, remove this figure.

References:

- Aiken, A., Salcedo, D., Cubison, M. J., Huffman, J. A., DeCarlo, P. F., Ulbrich, I. M., Docherty, K. S., Sueper, D., Kimmel, J. R., Worsnop, D. R., Trimborn, A., Northway, M., Stone, E. A., Schauer, J. J., Volkamer, R. M., Fortner, E., de Foy, B., Wang, J., Laskin, A., Shutthanandan, V., Zheng, J., Zhang, R., Gaffney, J., Marley, N. A., Paredes-Miranda, G., Arnott, W. P., Molina, L. T., Sosa, G., and Jimenez, J. L. (2009). Mexico City Aerosol Analysis During MILAGRO Using High Resolution Aerosol Mass Spectrometry at the Urban Supersite (T0)—Part 1: Fine Particle Composition and Organic Source Apportionment. Atmos. Chem. Phys. 9:6633–6653.

- Akagi, S. K., Yokelson, R. J., Wiedinmyer, C., Alvarado, M. J., Reid, J. S., Karl, T., Crounse, J. D., and Wennberg, P. O.: Emission factors for open and domestic biomass burning for use in atmospheric models, Atmos. Chem. Phys., 11, 4039–4072, https://doi.org/10.5194/acp-11-4039-2011, 2011.

- Canagaratna, M. R., Jimenez, J. L., Kroll, J. H., Chen, Q., Kessler, S. H., Massoli, P., et al. (2015). Elemental ratio measurements of organic compounds using aerosol mass spectrometry: Characterization, improved calibration, and implications. Atmospheric Chemistry and Physics, 15(1), 253–272. https://doi.org/10.5194/acp-15-253-2015

- Collier, S., Williams, L. R., Onasch, T. B., Cappa, C. D., Zhang, X., Russell, L. M., et al. (2018). Influence of emissions and aqueous processing on particles containing black carbon in a polluted urban environment: Insights from a soot particle-aerosol mass spectrometer. Journal of Geophysical Research: Atmospheres, 123, 6648–6666. https://doi.org/10.1002/2017JD027851

- Corbin, J. C., Sierau, B., Gysel, M., Laborde, M., Keller, A., Kim, J., et al. (2014). Mass spectrometry of refractory black carbon particles from six sources: Carbon-cluster and oxygenated ions. Atmospheric Chemistry and Physics, 14(5), 2591–2603. https://doi.org/10.5194/acp-14- 2591-2014

- Corbin, J. C., Lohmann, U., Sierau, B., Keller, A., Burtscher, H., and Mensah, A. A.: Black carbon surface oxidation and organic composition of beech-wood soot aerosols, Atmos. Chem. Phys., 15, 11885–11907, https://doi.org/10.5194/acp-15-11885-2015, 2015.

- Cubison, M. J., Ortega, A. M., Hayes, P. L., Farmer, D. K., Day, D., Lechner, M. J., et al. (2011). Effects of aging on organic aerosol from open biomass burning smoke in aircraft and laboratory studies. Atmospheric Chemistry and Physics, 11(23), 12,049–12,064. https://doi.org/ 10.5194/acp-11-12049-2011

- Forrister, H., et al. (2015), Evolution of brown carbon in wildfire plumes, Geophys. Res. Lett., 42, 4623–4630, doi:10.1002/2015GL063897.

- Jolleys, M. D., Coe, H., McFiggans, G., Taylor, J. W., O'Shea, S. J., Le Breton, M., Bauguitte, S. J.-B., Moller, S., Di Carlo, P., Aruffo, E., Palmer, P. I., Lee, J. D., Percival, C. J., and Gallagher, M. W.: Properties and evolution of biomass burning organic aerosol from Canadian boreal forest fires, Atmos. Chem. Phys., 15, 3077–3095, https://doi.org/10.5194/acp-15-3077-2015, 2015

- Lauren A. Garofalo, Matson A. Pothier, Ezra J. T. Levin, Teresa Campos, Sonia M. Kreidenweis, and Delphine K. Farmer (2019): Emission and Evolution of Submicron Organic Aerosol in Smoke from Wildfires in the Western United States, *ACS Earth and Space Chemistry 3* (7), 1237-1247. DOI: 10.1021/acsearthspacechem.9b00125

- Taehyoung Lee, Amy P. Sullivan, Laura Mack, Jose L. Jimenez, Sonia M. Kreidenweis, Timothy B. Onasch, Douglas R. Worsnop, William Malm, Cyle E. Wold, Wei Min Hao & Jeffrey L. Collett Jr. (2010) Chemical Smoke Marker Emissions During Flaming and Smoldering Phases of Laboratory Open Burning of Wildland Fuels, Aerosol Science and Technology, 44:9, i-v, DOI: 10.1080/02786826.2010.499884

- Lee, A. K. Y., Willis, M. D., Healy, R. M., Onasch, T. B., & Abbatt, J. P. D. (2015). Mixing state of carbonaceous aerosol in an urban environment: Single particle characterization using the soot particle aerosol mass spectrometer (SP-AMS). Atmospheric Chemistry and Physics, 15(4), 1823–1841. https://doi.org/10.5194/acp-15-1823-2015

- Lee, J. E., Dubey, M. K., Aiken, A. C., Chylek, P., & Carrico, C. M. (2020). Optical and chemical analysis of absorption enhancement by mixed carbonaceous aerosols in the 2019 Woodbury, AZ, fire plume. Journal of Geophysical Research: Atmospheres, 125, e2020JD032399. https://doi.org/ 10.1029/2020JD032399

- Lim, C. Y., Hagan, D. H., Coggon, M. M., Koss, A. R., Sekimoto, K., de Gouw, J., Warneke, C., Cappa, C. D., and Kroll, J. H.: Secondary organic aerosol formation from the laboratory oxidation of biomass burning emissions, Atmos. Chem. Phys., 19, 12797–12809, https://doi.org/10.5194/acp-19-12797-2019, 2019.

- Massoli, P., et al. (2015), Characterization of black carbon-containing particles from soot particle aerosol mass spectrometer measurements on the R/V Atlantis during CalNex 2010, J. Geophys. Res. Atmos., 120, 2575–2593, doi:10.1002/2014JD022834.

- May, A. A., et al. (2014), Aerosol emissions from prescribed fires in the United States: A synthesis of laboratory and aircraft measurements, J. Geophys. Res. Atmos., 119, 11,826–11,849, doi:10.1002/ 2014JD021848.

- McMeeking, G. R., et al. (2009), Emissions of trace gases and aerosols during the open combustion of biomassin the laboratory, J. Geophys. Res., 114, D19210, doi:10.1029/2009JD011836.

- Ann M. Middlebrook, Roya Bahreini, Jose L. Jimenez & Manjula R. Canagaratna (2012) Evaluation of Composition-Dependent Collection Efficiencies for the Aerodyne Aerosol Mass Spectrometer using Field Data, Aerosol Science and Technology, 46:3, 258-271, DOI: 10.1080/02786826.2011.620041

- Onasch, T. B., Fortner, E. C., Trimborn, A. M., Lambe, A. T., Tiwari, A. J., Marr, L. C., et al. (2015). Investigations of SP-AMS carbon ion distributions as a function of refractory black carbon particle type. Aerosol Science and Technology, 49(6), 409–422. https://doi.org/ 10.1080/02786826.2015.1039959

- Ortega, A. M., Day, D. A., Cubison, M. J., Brune, W. H., Bon, D., De Gouw, J. A., & Jimenez, J. L. (2013). Secondary organic aerosol for- mation and primary organic aerosol oxidation from biomass-burning smoke in a flow reactor during FLAME-3. Atmospheric Chemistry and Physics, 13(22), 11,551–11,571. https://doi.org/10.5194/acp-13-11551-2013

- Willis, M. D., Lee, A. K. Y., Onasch, T. B., Fortner, E. C., Williams, L. R., Lambe, A. T., Worsnop, D. R., and Abbatt, J. P. D.: Collection efficiency of the soot-particle aerosol mass spectrometer (SP-AMS) for internally mixed particulate black carbon, Atmos. Meas. Tech., 7, 4507–4516, https://doi.org/10.5194/amt-7-4507-2014, 2014.

---

## Referee Report (RR2)

[referee-annotated manuscript omitted]

---

## Author Response (AR2)

We thank the reviewer and editor for their comments. As well, we thank the reviewer for their intimate knowledge of the SP-AMS technique and their insightful questions.

The lead author, who did all of the new data analysis and plot-making for this manuscript, has a new position, so we addressed all of the comments posted here as best as we can given our constraints, but time for new analysis is limited. We have addressed all minor comments, and we have addressed the more substantive comments to the best of our current ability.

To aid the review process, we are placing reviewer and editor comments in black text, our responses in blue text, any changes to the text in red, and, in some instances, reproduce text from the previously submitted manuscript (*italic magenta*). We have numbered the reviewer (R) and editor (E) comments to assist the conversation. We note that the line numbers were unfortunately cut off in our submission, leading to difficulty for both the reviewer and the editor in identifying correct line numbers to cite. Fortunately, the reviewer often quoted sections they were referencing, allowing us to find and make corrections. There are unfortunately a small number of technical comments we were not able to address due to line number confusion and we have noted those throughout. The track-changes main text and supplement are included at the end of this document, with the main text starting on page 47 and the supplement starting on page 80.

**Editor comments**

E1) R1.25: Referee #1 had pointed out that changes in the Krevelen diagram may be also caused by mixing of different air masses. While you added the suggested references to Chen et al. (2015) to the method section, this information should be also added on p. 10.

After further consideration, we have decided that the Van Krevelen diagram (Figure S27) and discussion do not add any substantial information to the paper. We have removed the figure and associated text.

E2) R1.19: Referee #1 referred to your previous paper (Hodshire et al., 2019) where 'a variety of reasons' are discussed that may affect deltaOA/deltaCO evolution with ageing. I suggest that you include a bit more discussion on these reasons, i.e. similar to the information as included in the SI of your previous paper.

We have modified the text as follows:

"In general, both the cores and edges do not show any positive or negative trend in $\Delta OA/\Delta CO$ with physical aging, with $R_{\Delta OA,initial}$ and $R_{age}$ showing very weak correlations of 0.02 and +0.03 (with $R_{\Delta OA,initial}$ and $R_{age}$ ranging between -0.25 to +0.17 and 0 to 0.07, respectively, when individual flights are left out sequentially; Table S2). The absolute variability is dominated by differences between plumes. Many previous field campaigns similarly show little change in $\Delta OA/\Delta CO$ with aging (Hodshire et al., 2019a and references therein). This may be due to a balance between evaporation and condensation over the period of time that the plume is observed (Hodshire et al., 2019a). This hypothesis is supported by the observed $\Delta f_{60}$ and $\Delta f_{44}$: $\Delta f_{60}$ and $\Delta f_{44}$ show clear signs of changes with aging, consistent with previous studies (Cubison et al., 2011; Garofalo et al., 2019; May et al., 2015)."

E3) R1.39: Referee #1 asked for a detailed discussion to justify the assumption of the initial conditions. Your response is rather vague. I understand that you may not be able to do a quantitative estimate but some more discussion on how this assumption may be qualitatively affect your results should be added.

We copy comment R1.39 for reference (in black italics):
*L308: Again, how can the authors rule out differences in the initial conditions that are independent of physical or chemical aging? This seems to be an underlying assumption throughout this entire study, but I do not find that the authors have really justified this assumption. Given how central it is to everything, I strongly suggest that an explicit discussion must be included wherein the authors review the evidence for and against their assumption.*

We also copy our original response for reference (also in italics):

*We have added more text and qualifiers to section 3 addressing this issue, following comments R1.24 and R2.47. We add the following text to this discussion:*

*"We were unable to quantify the impact on potential interfire variability in the emission values of the metrics studied here (such as variable $f_{60}$ and $f_{44}$). We anticipate that being able to capture this additional source of variability may lead to stronger fits and correlation."*

*And*
*"We also suggest further refinement of our fit equations, as further variables (such as photolysis rates) and better quantification of interfire variability (such as variable emission rates) are anticipated to improve these fits."*

We have now added more discussion on this issue, particularly in response to the new comment #4 from reviewer 2. Please see that response for an extended discussion of this issue, and we copy relevant portions of that response here.

"This is an interesting question, and we have provided a more detailed line of reasoning below and modified the text to clarify this point.

In the two paragraphs above, the reviewer posits that the covariance of $\Delta OA_{initial}$ with $\Delta f_{60}$ and $\Delta f_{44}$ is primarily due to correlations of the OA emission factor with $\Delta f_{60}$ and $\Delta f_{44}$ at the time of emission (due to variabilities in burn conditions) and not due to evaporation and chemistry between the time of emission and the time of the first measurement. For this to be correct, variability in the emission factor of OA would need to be a significant contributor to the variability in $\Delta OA_{initial}$ (if the relative variability in the OA emission factor is much smaller than the relative variability in $\Delta OA_{initial}$, other factors contributing to variability in $\Delta OA_{initial}$ will wash out this emissions-based covariance between $\Delta OA_{initial}$ with $\Delta f_{60}$ and $\Delta f_{44}$).

In Figure 2 of our manuscript, $\Delta OA_{initial}$ varies by nearly 2 orders of magnitude (factor of 100), with multiple transects/percentiles on the high and low ends (i.e., not just a single outlier driving the upper or lower bound). In the Andreae, (2019) biomass burning emissions review paper, the OC emission factors have standard deviations that are about ½ of the mean values, so a factor of 3 variability in emission factors in the $-1\sigma$ to $+1\sigma$ range. Hence, variability in emission factors should not likely explain a large fraction of the variability in $\Delta OA_{initial}$. We note that (1) the emission fluxes from fires are the product of the emission factor and the fuel consumption rate (kg fuel per area per time) with fuel consumption rates varying greatly between fires (e.g., the difference between a surface fire and a crown fire), and (2) even for a fixed emission flux, variability in dilution rates from differences in fire size or atmospheric stability can lead to orders-of-magnitude differences in $\Delta OA_{initial}$ , even after just 15 minutes (see Figures 6c and 7c in Bian et al., 2017).

The variability in $\Delta OA_{initial}$ is thus likely driven much more by variability in dilution and fuel consumption rates, not emission factors. It is possible that OA emission factors are correlated with fuel consumption rates because flaming wildfires (with high modified combustion efficiency, MCE) may be correlated with fast fuel consumption. However, OA emission factors are lower during flaming, high-MCE conditions (McMeeking et al., 2009), so it would be an anticorrelation rather than a correlation (the lower OA emissions factors during flaming would counter the higher fuel-consumption rates during flaming, so the OA emission fluxes would be damped by this relationship and create less variability in $\Delta OA_{initial}$).

In Figure 2, much of the $\Delta OA_{initial}$ variability comes from including both the core and edge of the same plume. Would we expect the difference in $\Delta OA_{initial}$ to come from differences in the OA emission factor? This would require (1) no additional dilution at the edge relative to the core, which is inconsistent with how diffusion works and with the CO and $CO_2$ concentrations between the edge and core, and (2) the edge of the plume to be emissions from flaming conditions (low OA emission factors) and the core of the plume to come from smoldering conditions (high OA emission factors), which, while possible, does not seem like it would be the dominant (or even a common) scenario. The improbability of the two conditions above (especially #1) is further evidence that OA emission factors are not a large contributor to the variability in $\Delta OA_{initial}$.

Preliminary work (June et al., in *prep*) from the FIREX-AQ campaign is showing $\Delta OA_{initial}$ to be uncorrelated with MCE, lending further evidence that the variability in burn conditions and emissions factors do not drive much of the variability in $\Delta OA_{initial}$. FIREX-AQ was another biomass burning field campaign in which pseudo-Lagrangian research flights tracked wildfire plumes, similar to the BBOP field campaign.

[Figure]

(Nicole June, et al, *in prep*)

Hence, there are multiple lines of reasoning that lead to the conclusion that OA emission factors are not a large contributor to the variability in $\Delta OA_{initial}$ (making relationships between OA emission factors with $\Delta f_{60}$ and $\Delta f_{44}$ unlikely to significantly contribute to the observed relationships between $\Delta OA_{initial}$ with $\Delta f_{60}$ and $\Delta f_{44}$). On the other hand, theoretical studies (Bian et al., 2017; Hodshire et al., 2019a), lab analyses (Hodshire et al., 2019b), and an independent field analysis (Palm et al., 2020), have shown that we expect evaporation and chemistry before the first transect and that the extend of these processes depends on the plume concentrations.

This reasoning for ruling out the role of variability of emission factors driving relationships between $\Delta OA_{initial}$ to and $\Delta f_{60}$ and $\Delta f_{44}$ should have been explicit in previous versions of the manuscript, particularly after the confusion in the first round of comments. We now have added the following text to the manuscript:

"Prior studies have shown that $f_{60}$ and $f_{44}$ at the time of emissions correlate with OA emissions factors through variability in burn conditions (Hennigan et al. 2011; Cubison et al. 2011; McClure et al. 2020), and this relationship might also contribute to our observed correlation between $\Delta f_{60}$ and $\Delta f_{44}$ with $\Delta OA_{initial}$. For this emissions relationship to be an important factor, the variability in the OA emission factor needs to be a significant contributor to the variability in $\Delta OA_{initial}$. If the relative variability in the OA emission factor is much smaller than the relative variability in $\Delta OA_{initial}$, other factors contributing to variability in $\Delta OA_{initial}$ will negate an emissions-based covariance between $\Delta OA_{initial}$ with $\Delta f_{60}$ and $\Delta f_{44}$. While our observed $\Delta OA_{initial}$ in Figure 2 spans nearly a factor of 100, Andreae (2019) shows that the OA emission factors have a $-1\sigma$ to $+1\sigma$ range of around a factor 3. Hence, variability in fuel consumption rates and dilution prior to the first transect likely dominate the variability in $\Delta OA_{initial}$, and the relationships of $\Delta f_{60}$ and $\Delta f_{44}$ with $\Delta OA_{initial}$ are unlikely to be influenced much by variability in burn conditions. We conclude that evaporation and/or chemistry prior to the first measurement appears to drive the initial relationship between $\Delta f_{60}$ and $\Delta f_{44}$ with $\Delta OA_{initial}$, consistent with (1) the theoretical work of Hodshire et al. (2019a), (2) an analysis of what chemistry would be missed in laboratory experiments if the initial 10-60 minutes of chemistry was not considered, following field experiments (Hodshire et al., 2019b), and (3) the recent field analysis (Palm et al., 2020)."

McMeeking et al., Emissions of trace gases and aerosols during the open combustion of biomass in the laboratory, 114, D19210, doi:10.1029/2009JD011836, 2009.

Bian, Q., Jathar, S. H., Kodros, J. K., Barsanti, K. C., Hatch, L. E., May, A. A., Kreidenweis, S. M., and Pierce, J. R.: Secondary organic aerosol formation in biomass-burning plumes: Theoretical analysis of lab studies and ambient plumes, Atmos. Chem. Phys., 17, 5459-5475, doi:10.5194/acp-2016-949, 2017.

Andreae, M., Atmos. Chem. Phys., 19, 8523–8546, https://doi.org/10.5194/acp-19-8523-2019, 2019.

A. L. Hodshire, Q. Bian, E. Ramnarine, C. R. Lonsdale, M. J. Alvarado, S. M. Kreidenweis, S. H. Jathar, J. R. Pierce: More than emissions and chemistry: Fire size, dilution, and background aerosol also greatly influence near-field biomass burning aerosol aging, J. Geophys. Res., 124, https://doi.org/10.1029/2018JD029674, 2019.

Anna L. Hodshire, Ali Akherati, Matthew J. Alvarado, Benjamin Brown-Steiner, Shantanu H. Jathar, Jose L. Jimenez, Sonia M. Kreidenweis, Chantelle R. Lonsdale, Timothy B. Onasch, Amber M. Ortega, Jeffrey R. Pierce: Aging Effects on Biomass Burning Aerosol Mass and Composition: A Critical Review of Field and Laboratory Studies, Env. Sci. Tech., https://doi.org/10.1021/acs.est.9b02588, 2019b.

Palm et al., Quantification of organic aerosol and brown carbon evolution in fresh wildfire plumes, https://doi.org/10.1073/pnas.2012218117, 2020."

E4) There are several instances, where language is inaccurate or not appropriate. Some examples are included in the following, but please check the complete manuscript for similar expressions. (Please note that the line numbers were cut off in the uploaded manuscript file and only the last two digits are legible. In the following, I refer to these line numbers as they appear in the pdf.)

We agree that the manuscript will benefit from careful editing. We have addressed the editor and reviewer comments on this issue and have gone through the manuscript again to attempt to catch any poor phrasing.

E5) p. 7, l. 19: 'for many of the individual metrics' – be more specific here.

We have changed this sentence to read
"The remaining variables plotted also show some noise and few clear trends, but it is apparent that the 5-15 and 90-100 percentiles do show a separation for some of the individual metrics, in particular $\Delta f_{44}$ and $\Delta O/\Delta C$."

E6) Ep. 8, l. 43: 'each variable ages downwind' should be rephrased

We have changed this sentence to read
"The Spearman tests show correlation coefficients for each flight set (Table S1) with the initial $\Delta OA$ of a flight set ($\Delta OA_{initial}$) against $\Delta OA/\Delta CO$, $\Delta f60$, $\Delta f44$, $\Delta H/\Delta C$, and $\Delta O/\Delta C$ as the smoke aerosol age downwind."

E7) p. 11, l. 54: 'our fits were made during daytime conditions' Wouldn't it be more accurate to say 'our fits were made FOR daytime conditions' ?

Changed.

E8) p. 8, l. 68: 'deltaf60 may be evaporating ' should be rephrased

We have rephrased this to:
"...consistent with the hypotheses that compounds containing species that can fragment to m/z 60 may be evaporating…"

**Reviewer #2 second round of comments**

Review of "Dilution impacts on smoke aging: Evidence in BBOP data"

By Anna L. Hodshire et al. Anonymous Reviewer Summary:

This manuscript derives empirical relationships for aerosol chemical age, number concentration, and mean mobility diameter which are useful for modeling the climate impact of biomass-burning aerosols. New insight from the BBOP measurement campaign is used to show that the rate of chemical aging of aerosol is affected by the "concentration" of the plume, presumably through decreasing photolytic rates with increased aerosol optical depth, and plume size, presumably because the core of larger plumes and are more protected from mixing with clean ambient air.

Ultimately, this is an interesting study but has a number of key assumptions that need to be further investigated or justified. Although the authors have weakened their language since the original draft due to comments from reviewers, they have not addressed the underlying concerns of the reviewers. This manuscript will require major revisions prior to considering publication.

General Comments:

 R1) In general, the language has improved but is still not very precise and there are grammatical issues.

We agree that the manuscript has benefited from careful editing. We have addressed the editor and reviewer comments on this issue and have gone through the manuscript again to attempt to catch any poor phrasing and grammatical issues.

 R2) There is a lack of details regarding SP-AMS measurements (as well as description of other instruments). There are only scattered references to the operating conditions/settings used for measuring OA in these plumes. Although the instrumentation is fully described in another manuscript, there is a minimum amount of information required for the SP-AMS measurement:

vaporizers equipped and modes used (switching between modes, temperature of thermal vaporizer), calibration, description of CE determination, ToF mode (HR-ToF, C-ToF, V-mode, W-mode), MS sampling timing (Open, closed, PToF, ePToF, pulser period, etc)

There also needs to be a description of the mass spectra analysis. What software was used. Are you reporting UMR or HR results? Is f60 based on m/z=60 or the specific ion C2H4O2+? How is gas phase subtracted? Assuming constant [CO2] gas phase concentrations of 400 ppm?

Since the vaporizer modes were switched (presumably intermittently) it is concerning how the authors choose to combine the data from the different modes. These modes measure inherently different components of the aerosol mass and fractionate the molecules in different ways. While Lee et al (2020) show that molecule fractionation in the different vaporizer modes is similar for the C2H4O2+ ion (used to calculate f60, although that is not stated by the authors), the fractionation is significantly different for the organic fraction of CO2+ (used to calculate f44) (see also Onasch et al 2012, Canagaratna et al 2015, etc).

We have greatly expanded the supplementary details of the SP-AMS, responding to all points here, including 2 new figures (Figs S29 and S30). We have also added a sentence in this main text pointing to this supplementary text. The revised supplementary text is as follows:

"The Soot Particle – Aerosol Mass Spectrometer (SP-AMS) operating on the DOE G1 aircraft during BBOP has been described in detail by Collier et al. (2016), Sedlacek et al. (2018), and Kleinman et al. (2020). The SP-AMS sampled PM1 through a constant pressure inlet operating at a pressure of ~620 Torr (Bahreini et al., 2008). The SP-AMS was equipped with dual vaporizers: (1) standard resistively heated tungsten vaporizer; and (2) 1064 nm intracavity laser vaporizer (Onasch et al., 2012). The standard tungsten vaporizer was operated at a nominal value of 600°C for the full data set. The SP-AMS operating with the laser vaporizer OFF is effectively the same as a standard HR-AMS, measuring non-refractory particulate matter (NR-PM). The SP-AMS operating in dual vaporizer mode, with both the standard tungsten vaporizer and the laser vaporizer ON measures the NR-PM and is additionally sensitive to refractory black carbon (rBC).

Flight data was collected at a rapid rate using "Fast-MS" in V-mode (i.e., mass spectral resolution ~2000) with 1 second sample time, with negligible particle time-of-flight (PTOF) data (DeCarlo et al., 2006; Lack et al., 2009). The pulsed, orthogonal extraction time-of-flight mass spectrometer (TOF-MS) was operated with a 60 μs pulser period and collected mass spectra from m/z 11 to m/z 955. "Fast-MS" data was collected in open (i.e., sample) mode for 52 seconds and in closed (i.e., background) mode for 8 seconds every minute. The laser vaporizer was operated by either automatically alternated laser ON and OFF each minute or manually sampling with the laser ON or OFF for long periods of time, such as full plume transects. The majority of the data (>76%) was collected in dual vaporizer mode (i.e., laser on).

The SP-AMS was calibrated for NR-PM with ammonium nitrate and for rBC with Regal black 8 independent times during BBOP. The average ionization efficiency (IE) with respect to ammonium nitrate was measured to be 8.1e-8 and the relative ionization efficiency (RIE) of rBC was measured to be 0.28, although the rBC from the SP-AMS was not used in this study.

Collier et al. (2016) determined the SP-AMS laser OFF collection efficiency (CE) to be 0.5 through comparisons with an independent HR-AMS located at the Mount Bachelor Observatory during over-flights. SP-AMS measured NR-PM values collected with the laser ON and OFF were compared for 16 different biomass burning plumes (Sedlacek et al., 2018; Kleinman et al., 2020). In each case, the plume was sampled with the laser ON and with the laser OFF, independently, and the measured plume NR-PM was normalized to CO to account for potential changes in the plume dilution between transects. The average ratio for NR-PM laser ON to laser OFF was 1.52. From these results, the average CE of NR-PM measured with the laser ON to be 0.76 with a standard deviation of 0.07 (Sedlacek et al., 2018; Kleinman et al., 2020). There is substantial evidence in the published literature for the CE of the tungsten vaporizer (Lim et al., 2019) and the laser vaporizer (Willis et al., 2014) to change as a function of chemical composition and rBC coating thickness. Unfortunately for various reasons, instrument comparisons of measurements of PM1 mass loading concentrations were very limited during BBOP, such that there does not exist a useful estimate of a changing CE for either SP-AMS vaporizer with changing plume conditions.

The SP-AMS data was analyzed using ToF-AMS Analysis Toolkit 1.61B and ToF-AMS HR Analysis 1.21B in Igor Pro. Gas phase carbon dioxide ($CO_2$) was directly measured on the G1 aircraft and was used to subtract gas phase contributions to $CO_2^+$ ion signal in the SP-AMS. SP-AMS standard NR-PM chemical species (i.e., Org, $SO_4$, $NO_3$, $NH_4$, Chl) were calculated using high resolution (HR) fits. $f_{44}$ and $f_{60}$ are unit mass resolution (UMR) ratios, whereas O:C ratios were derived using HR fits. Although it was not directly characterized for uncertainties during the BBOP campaign, we estimate uncertainties as follows. The AMS uncertainty is estimated following the methods in (Bahreini et al. 2009) (first equation of their supplemental information), leading to 37% uncertainty for organics. The laser vaporizer adds additional uncertainty up to 20%. Thus summing the uncertainties in quadrature leads to a 42% uncertainty in organics.

We further analyzed the UMRs and the potential for laser ON specific ion signals to interfere with laser OFF NR-PM ion signals with the SP-AMS data. The chemical composition of the measured wildfire plumes during BBOP were > 90% NR-PM organic material (Collier et al., 2016; Kleinman et al., 2020). rBC mass fractions were typically below 2% (Kleinman et al., 2020), though the number fractions were higher (Sedlacek et al., 2018). Despite these low concentrations, the SP-AMS laser ON (relative to laser OFF) was observed to generate $C_n^+$ ion signals with an identifiable fragmentation pattern for rBC material and the laser ON to OFF

NR-PM signal was observed to increase by ~50% on average. Similar results have been published for ambient urban aerosol (e.g., Lee et al. 2015). Recent laboratory work to investigate these issues has eliminated laser alignment issues and indirect heating as potential causes for these observations (Avery et al., 2020). Thus, these observations are likely due to a combination of different collection efficiencies (CEs) and relative ionization efficiencies (RIEs) for the two vaporizers when used in dual vaporizer mode (i.e., laser ON).

The HR ion signals at m/z 44 are dominated by $CO_2^+$ and $C_2H_4O^+$ ions (Fig. S29). The ratio of $C_2H_4O^+/CO_2^+$ increases with plume mass loading (i.e., concentration) and decreases with distance from the fire (Fig. S29), inline with the observations reported here for decreases in oxidation levels as a function of dilution. The HR ion signals at m/z 60 are dominated by $C_2H_4O_2^+$ and $C_5^+$ (Fig. S30). HR fitting of $C_5^+$ indicated that it averaged ~6% of the $C_2H_4O_2^+$ ion signal, independent of the laser vaporizer state (i.e., ON or OFF). For large $C_2H_4O_2^+$ ion signals in relatively undiluted biomass burning plumes, this ratio is likely controlled by the errors in fitting a small peak in the wings of a larger peak (Corbin et al., 2015). At lower ion signal levels, the $C_5^+/C_2H_4O_2^+$ becomes significantly noisier, but the average does not change significantly. Laser ON may slightly increase the average ratio at lower $C_2H_4O_2^+$ ion signals, which could overestimate $f_{60}$ for relatively dilute plumes. If this were true, the observed decrease in $f_{60}$ with plume dilution (i.e., due to fire size and atmospheric age) would be slightly smaller than reported here.

Past research on SP-AMS ion signals from the laser vaporizer and the standard tungsten vaporizer have identified several complicating factors when operating the SP-AMS in dual vaporizer mode. First, organic material coating rBC particles and detected using the laser vaporizer have noted different fragmentation patterns (Onasch et al., 2012) and chemical compositions (Canagaratna et al., 2015) compared with the same organic material detected using the standard tungsten vaporizer. Further, there are reports of SP-AMS laser vaporizer detecting refractory $CO_2^+$ ions from rBC particles (Corbin et al., 2014). Currently, we have not assessed the potential for refractory $CO_2^+$ ion signals during BBOP as both the rBC and Org signals are highly correlated in biomass burning plumes, making minor changes to these ratios difficult to ascertain. To address the question of whether the laser vaporizer generated different ion signals from similar organic compounds, we analyzed the laser ON and OFF plume transect pairs that were used for determining laser ON CE values relative to laser OFF.

As shown in Fig. S31, the HR O:C, UMR $f_{44}$, and UMR $f_{60}$ ratios are highly correlated between laser ON and OFF conditions, though differ by apparent factors. Laser ON HR O:C ratios are approximately 4% lower than laser OFF. In large part, this is due to the UMR $f_{44}$ ratios, which are dominated by $CO_2^+$ ions, being 17% lower for laser ON. UMR $f_{60}$ ratios are 18% higher in laser ON than OFF. These observations are in line with the published results from Canagaratna et al., (2015), which observed that laser vaporizer only HR O:C ratios were ~17% lower than tungsten vaporizer only HR O:C ratios for the same organic material and the HR H:C ratios were ~16% higher. In the case of BBOP, the laser vaporizer signals represented approximately 1/3 of the total organic signal with dual vaporizers. The BBOP measured 4% lower HR O:C ratios are similar in magnitude to 5.6% (i.e., 0.33*17%) expected if the Canagaratna et al. (2015) results applied to BBOP measurements.

The BBOP SP-AMS data used in this manuscript is used to measure trends in OA. O:C, $f_{44}$, and $f_{60}$ with plume dilution, either at different plume ages and/or different concentration percentiles across a biomass plume (i.e., edge vs. center). A question is whether the mixing of laser ON and OFF data here somehow biases the results due to the different absolute values between the two different states. A quick extension of the above plume pair analysis (Fig. S31) includes several "background" measurements made between the plumes (i.e., below 150 ppbv CO) and compared for laser ON vs. OFF to investigate if this ratio changes substantially between plume (i.e., high level) and background (i.e., low level) levels. The laser ON:OFF ratios of measured HR O:C averaged 0.95±0.049 in background and 0.96±0.029 in plume, UMR $f_{44}$ averaged 0.89±0.085 in background and 0.85±0.068 in plume, and UMR $f_{60}$ averaged 1.17±0.23 in background and 1.15±0.13 in plume. These results suggest that the observed laser ON/OFF ratios do not change from low to high signal levels, such that the trends observed for laser OFF should hold for laser ON, and vice versa. Further, the laser ON vs. OFF data points are randomly distributed throughout the measurements rather than systematically distributed to near- vs. far-field measurements or core vs. edge measurements. Hence, there should be no systematic bias due to the use of the combined laser ON and OFF data, although this combination of laser-on and -off data may contribute to noise in the observed trends."

[Figure]

Figure S29. (a) High resolution fits at m/z 44 for a biomass burning plume during 0730b research flight with laser ON. (b) Correlation of HR $CO_2^+$ ion and HR total ion signal at m/z 44, colored by distance downwind (km) from fire.

[Figure]

Figure S30. High resolution fits at m/z 60 for a biomass burning plume during 0730b research flight with laser ON.

[Figure]

Figure S31. Laser ON versus laser OFF SP-AMS HR O:C, UMR f44, and UMR f60 ratios.

R3) Data are binned by physical age and further into edge-vs-core (such as shown in Figure 1). Each binned datum represents multiple measurements and therefore, in figure 1 and 2, should include error bars representing the variance of those measurements. This could be independent of measurement uncertainty, but would be better if it did include propagated instrument uncertainty.

We have not added these error bars due to the time constraints of the lead author as described above, as adding errors bars requires multiple analysis for each instrument and flight used in this work. We are not making any statements about systematic variance with age based on Figure 1 and the trend statistics in Figure 2 and elsewhere would be done using the mean-value datapoints (that are currently plotted) regardless.

However, we have explicitly addressed the lack of error bars in the text:

"Note that in Figure (and Figures S14-S18), the points represent the mean values for each transect and do not include error bars for uncertainty in the mean or measurement uncertainty as characterization of systematic variance (within plume percentiles) with age is beyond the scope of this study."

"As with Figure 1, the points in Figure 2 represent the mean values for each transect and percentile, and we do not include error bars as we do not attempt to characterize systematic variance (within plume percentiles) with age in this study."

R4) A key assumption of the authors is that DOAinitial, Df60, and Df44 can be used to identify dilution of the plume. However, these parameters are known to vary in primary emissions.

This is similar to question E3) from the editor. We respond in full here and copy relevant sections to question E3). To clarify, while we use $\Delta OA_{initial}$ as a proxy for dilution, we do not use $\Delta f_{60}$ and $\Delta f_{44}$ to identify dilution in the plume. Rather, we investigate if $\Delta f_{60}$ and $\Delta f_{44}$ (and other variables) systematically vary with $\Delta OA_{initial}$. To help make this more clear, we have added the following at the beginning of section 3.1 (underline new material):
"We color each line by the mean $\Delta OA$ within a $\Delta CO$ percentile bin from the transect closest to the fire, $\Delta OA_{initial}$, in order to examine whether each variable ($\Delta OA/\Delta CO$, $\Delta f_{60}$, $\Delta f_{44}$ $\Delta H/\Delta C$, and $\Delta O/\Delta C$) vary with $\Delta OA_{initial}$."

In the manuscript, the authors support the assumption regarding DOAinitial with the measured Df60 and Df44. However, these parameters are all more likely related to variations in POA between fires and within a fire as fire conditions change. The Df60 and Df44 measurements are the only support the authors provide for their main conclusion.

In the author's revisions, they have tried to further justify this assumption by making a flawed argument that their interpretation is only invalid if f60 and f44 covary with OA emissions. First this argument is flawed, as Df60initial and Df44initial are more reasonably attributed to differences in POA. Second, it has actually been shown that f60 and f44 of POA can covary with OA emission factors (see Corbin et al 2015; Ortega et al 2015; and Lee et al 2010).

This is an interesting question, and we have provided a more detailed line of reasoning below and modified the text to clarify this point.

In the two paragraphs above, the reviewer posits that the covariance of $\Delta OA_{initial}$ with $\Delta f_{60}$ and $\Delta f_{44}$ is primarily due to correlations of the OA emission factor with $\Delta f_{60}$ and $\Delta f_{44}$ at the time of emission (due to variabilities in burn conditions) and not due to evaporation and chemistry between the time of emission and the time of the first measurement. For this to be correct, variability in the emission factor of OA would need to be a significant contributor to the variability in $\Delta OA_{initial}$ (if the relative variability in the OA emission factor is much smaller than the relative variability in $\Delta OA_{initial}$, other factors contributing to variability in $\Delta OA_{initial}$ will wash out this emissions-based covariance between $\Delta OA_{initial}$ with $\Delta f_{60}$ and $\Delta f_{44}$).

In Figure 2 of our manuscript, $\Delta OA_{initial}$ varies by nearly 2 orders of magnitude (factor of 100), with multiple transects/percentiles on the high and low ends (i.e., not just a single outlier driving the upper or lower bound). In the Andreae, (2019) biomass burning emissions review paper, the OC emission factors have standard deviations that are about ½ of the mean values, so a factor of 3 variability in emission factors in the $-1\sigma$ to $+1\sigma$ range. Hence, variability in emission factors should not likely explain a large fraction of the variability in $\Delta OA_{initial}$. We note that (1) the emission fluxes from fires are the product of the emission factor and the fuel consumption rate (kg fuel per area per time) with fuel consumption rates varying greatly between fires (e.g., the difference between a surface fire and a crown fire), and (2) even for a fixed emission flux, variability in dilution rates from differences in fire size or atmospheric stability can lead to orders-of-magnitude differences in $\Delta OA_{initial}$ , even after just 15 minutes (see Figures 6c and 7c in Bian et al., 2017).

The variability in $\Delta OA_{initial}$ is thus likely driven much more by variability in dilution and fuel consumption rates, not emission factors. It is possible that OA emission factors are correlated with fuel consumption rates because flaming wildfires (with high modified combustion efficiency, MCE) may be correlated with fast fuel consumption. However, OA emission factors are lower during flaming, high-MCE conditions (McMeeking et al., 2009), so it would be an anticorrelation rather than a correlation (the lower OA emissions factors during flaming would counter the higher fuel-consumption rates during flaming, so the OA emission fluxes would be damped by this relationship and create less variability in $\Delta OA_{initial}$).

In Figure 2, much of the $\Delta OA_{initial}$ variability comes from including both the core and edge of the same plume. Would we expect the difference in $\Delta OA_{initial}$ to come from differences in the OA

emission factor? This would require (1) no additional dilution at the edge relative to the core, which is inconsistent with how diffusion works and with the CO and $CO_2$ concentrations between the edge and core, and (2) the edge of the plume to be emissions from flaming conditions (low OA emission factors) and the core of the plume to come from smoldering conditions (high OA emission factors), which, while possible, does not seem like it would be the dominant (or even a common) scenario. The improbability of the two conditions above (especially #1) is further evidence that OA emission factors are not a large contributor to the variability in $\Delta OA_{initial}$.

Preliminary work (June et al., in *prep*) from the FIREX-AQ campaign is showing $\Delta OA_{initial}$ to be uncorrelated with MCE, lending further evidence that the variability in burn conditions and emissions factors do not drive much of the variability in $\Delta OA_{initial}$.

[Figure]

(Nicole June, et al, *in prep*)

Hence, there are multiple lines of reasoning that lead to the conclusion that OA emission factors are not a large contributor to the variability in $\Delta OA_{initial}$ (making relationships between OA emission factors with $\Delta f_{60}$ and $\Delta f_{44}$ unlikely to significantly contribute to the observed relationships between $\Delta OA_{initial}$ with $\Delta f_{60}$ and $\Delta f_{44}$). On the other hand, theoretical studies (Bian et al., 2017; Hodshire et al., 2019a), lab analyses (Hodshire et al., 2019b), and an independent field analysis (Palm et al., 2020), have shown that we expect evaporation and chemistry before the first transect and that the extent of these processes depends on the plume concentrations.

This reasoning for ruling out the role of variability of emission factors driving relationships between $\Delta OA_{initial}$ to and $\Delta f_{60}$ and $\Delta f_{44}$ should have been explicit in previous versions of the manuscript, particularly after the confusion in the first round of comments. We now have added the following text to the manuscript:

"Prior studies have shown that $f_{60}$ and $f_{44}$ at the time of emissions correlate with OA emissions factors through variability in burn conditions (Hennigan et al. 2011; Cubison et al. 2011; McClure et al. 2020), and this relationship might also contribute to our observed correlation between $\Delta f_{60}$ and $\Delta f_{44}$ with $\Delta OA_{initial}$. For this emissions relationship to be an important factor, the variability in the OA emission factor needs to be a significant contributor to the variability in $\Delta OA_{initial}$. If the relative variability in the OA emission factor is much smaller than the relative variability in $\Delta OA_{initial}$, other factors contributing to variability in $\Delta OA_{initial}$ will negate an emissions-based covariance between $\Delta OA_{initial}$ with $\Delta f_{60}$ and $\Delta f_{44}$. While our observed $\Delta OA_{initial}$ in Figure 2 spans nearly a factor of 100, Andreae (2019) shows that the OA emission factors have a $-1\sigma$ to $+1\sigma$ range of around a factor 3. Hence, variability in fuel consumption rates and dilution prior to the first transect likely dominate the variability in $\Delta OA_{initial}$, and the relationships of $\Delta f_{60}$ and $\Delta f_{44}$ with $\Delta OA_{initial}$ are unlikely to be influenced much by variability in burn conditions. We conclude that evaporation and/or chemistry prior to the first measurement appears to drive the initial relationship between $\Delta f_{60}$ and $\Delta f_{44}$ with $\Delta OA_{initial}$, consistent with (1) the theoretical work of Hodshire et al. (2019a), (2) an analysis of what chemistry would be missed in laboratory experiments if the initial 10-60 minutes of chemistry was not considered, following field experiments (Hodshire et al., 2019b), and (3) the recent field analysis (Palm et al., 2020)."

McMeeking et al., Emissions of trace gases and aerosols during the open combustion of biomass in the laboratory, 114, D19210, doi:10.1029/2009JD011836, 2009.

Bian, Q., Jathar, S. H., Kodros, J. K., Barsanti, K. C., Hatch, L. E., May, A. A., Kreidenweis, S. M., and Pierce, J. R.: Secondary organic aerosol formation in biomass-burning plumes: Theoretical analysis of lab studies and ambient plumes, Atmos. Chem. Phys., 17, 5459-5475, doi:10.5194/acp-2016-949, 2017.

Andreae, M., Atmos. Chem. Phys., 19, 8523–8546, https://doi.org/10.5194/acp-19-8523-2019, 2019.

A. L. Hodshire, Q. Bian, E. Ramnarine, C. R. Lonsdale, M. J. Alvarado, S. M. Kreidenweis, S. H. Jathar, J. R. Pierce: More than emissions and chemistry: Fire size, dilution, and background aerosol also greatly influence near-field biomass burning aerosol aging, J. Geophys. Res., 124, https://doi.org/10.1029/2018JD029674, 2019.

Anna L. Hodshire, Ali Akherati, Matthew J. Alvarado, Benjamin Brown-Steiner, Shantanu H. Jathar, Jose L. Jimenez, Sonia M. Kreidenweis, Chantelle R. Lonsdale, Timothy B. Onasch, Amber M. Ortega, Jeffrey R. Pierce: Aging Effects on Biomass Burning Aerosol Mass and Composition: A Critical Review of Field and Laboratory Studies, Env. Sci. Tech., https://doi.org/10.1021/acs.est.9b02588, 2019b.

Palm et al., Quantification of organic aerosol and brown carbon evolution in fresh wildfire plumes, https://doi.org/10.1073/pnas.2012218117, 2020.

R5) Interpretation of Spearman's correlation coefficients and the strength of these coefficients is hindered by the authors choice to combine all data from all flights together for their regressions without normalization despite showing that both aerosol age and emission factors affect the parameter of interest (e.g. Df60). This multi-variate dependence is even stated several times by the authors. For example, with 1 exception, all transect sets predictably show that Df60 decreases with physical age, but because different transect sets started with different f60 values the combined data set does not monotonically decrease with physical age and the regression results in a weak relationship (RSpearman = -0.26).

This is an example of where the authors should rethink their analytical approach but have instead weakened the language of their results.

We thank the reviewer for their suggestions here. It prompted careful consideration, but we have determined to keep the current approach. Our aim with the Spearman's correlation coefficients as used in the manuscript is to answer the question, "If all you know is the time and if all you know is the $\Delta OA_{initial}$, how well could you predict the variability between plumes?", and we now state this more clearly in the manuscript. We also are drawing more attention to the multivariate fits in Figure 3 earlier in the manuscript (details on the changes to the manuscript below). Further, we want the correlation between $\Delta OA_{initial}$ and the initial values of the variables in Figure 2 to be included in the overall $R_{\Delta OA,initial}$ as these relations are highly unlikely to be driven by covariation of emissions (see the response to R4).

There are several possibilities that the authors could consider. Continuing to use Df60 as an example, the authors could:

● Normalizing the data to the initial measured value (e.g. Df60- Df60initial) prior to combining the data. This allows you to remove the processes driving variability in the initial Df60 (essentially the emission factor) so you can isolate the effects of physical age on Df60.

There is information in the initial values that we do not want to remove. And because a relationship between the initial values of some variables with $\Delta OA_{initial}$ exists even though OA emission factors is not a major driver of $\Delta OA_{initial}$ variability, the initial values of e.g. $\Delta f_{60}$ and $\Delta f_{44}$ is very likely not due to the emission factor (see the response to R4)

- A multivariate analysis with predictors of DOAinitial and physical age. You could do this in any programming language, but using Excel as an example you would use the Data>Data_Analysis>regressions gui. It appears that you use this in your model in section 3.1 in equation 4. Since you did this, why do you even show the single-variable Rpearson and Rspearman values?

These separate analyses address different questions (see above), and we have added discussion on this to the manuscript.

- Lastly, you could analyze each transect set separately to get a Rspearman value of Df60 versus physical age and then average those values together. If this approach is chosen, then averaging should weighted by the number of transects in the transect set. Also, you should use a jack-knife-like approach, repeating the averaging by systematically excluding 1-2 transects sets to see how dependent the results are on any individual set.

This approach is not consistent with the question we sought to answer with these coefficients, and rather, the multivariate fit in Figure 3 with coefficients in Table S3 serves to separate the role of the two predictors.

We note that we did do a leave-one-out analysis (Table S2) after the first round of reviewer comments, following suggestions made in reviewer comments and responses R1.21 and R2.59.

We believe that our approach is appropriate for the question we sought to answer. We have now added the following text:

"We calculate these correlation coefficients separately for Figure 2 to determine how well the variability can be predicted from the $\Delta OA_{initial}$ or age alone (and whether the data are correlated vs. anticorrelated with these predictors). To complement these independent correlation coefficients, we also perform multivariate linear regressions (Eqns. 4 and 5 and Figure 3, discussed later) to explicitly decouple the influence of the two predictors."

R6)  SI section: The supplement has improved with more detail but is still lacking. More detail is needed describing the instrument set up, even if it is described fully in another paper.

We have greatly expanded the SI. Please see our response to R2.

R7) The heterogeneous chemistry calculations needs a description of the calculation and justification for the methodology used. The only information provided is that it is a "simple calculation" and a list of what the parameters are.

This description was in Section S2 in the supplement. The main text states *"However, estimates of heterogeneous mass losses indicate that after three hours of aging (the range of time the BBOP measurements were taken in) for a range of OH concentrations and reactive uptake coefficients, less than 10% of aerosol mass is lost to heterogeneous reactions (Fig. S23; see SI text S2 for more details on the calculation)."* We have modified Section S2 in the supplement in response to this comment:

"We test the impact of heterogeneous chemistry on aerosol mass loss within the smoke plume. We performed a simple calculation of reactive uptake of OH molecules with particle-phase organics that resulted in loss of organic products. These calculations include assumed values of particle diameter, OH concentration, OH diffusion coefficient, and OH reactive uptake coefficient. The following parameters are assumed for the calculations:

- OH diffusivity = 3.5e-5 $[m^2 \ s^{-1}]$
- Particle diameter varied from 1 - 1000 [nm]
- Constant OH concentration varied from 1e5 to 5e7 [molecules $cm^{-3}$]
- Reactive uptake coefficients varied from 0.1 to 1 [unitless]
- Molecular weight of organics = 200 [g $mol^{-1}$]
- Density of organics = 1.4 [g $cm^{-3}$]
- Total run time = 3 [hours]

The collision rate of OH with the particle surface was calculated using the condensation equations in Seinfeld and Pandis (2006). As a calculation of the upper bound limit of evaporation due to heterogeneous chemistry, we assume each collision results in removing an organic molecule on the surface of the particle (assumed to be 200 amu), fragmenting and removing the molecule from the particle. The fragmentation products are not assumed to participate in further reaction. Figure S23a shows the resulting final:initial mass ratios after four hours of aging, indicating that for the aerosol sizes containing most of the mass in this study (>100 nm) and under expected OH concentrations ($<10^7 \ cm^{-3}$), >90% of the aerosol mass remains after 3 hours in all but the cases with a reactive uptake coefficient of 1 and an OH concentration of $10^7 \ cm^{-3}$. Note however that (1) the reactive uptake coefficient is likely lower than 1 (Slade and Knopf, 2013), (2) not every reaction will lead to complete evaporation of all products, and (3) OH concentrations are often lower than $10^7 \ cm^{-3}$ (Juncosa Calahorrano et al., 2020)."

Juncosa Calahorrano, J. F., Lindaas, J., O'Dell, K., Palm, B. B., Peng, Q., Flocke, F., Pollack, I. B., Garofalo, L. A., Farmer, D. K., Pierce, J. R., Collett, J. L., Weinheimer, A., Campos, T., Hornbrook, R. S., Hall, S. R., Ullmann, K., Pothier, M. A., Apel, E. C., Permar, W., Hu, L., Hills, A. J., Montzka, D., Tyndall, G., Thornton, J. A. and Fischer, E. V.: Daytime Oxidized Reactive Nitrogen Partitioning in Western U.S. Wildfire Smoke Plumes, J. Geophys. Res. Atmos., 1–47, doi:10.1029/2020jd033484, 2020.

Specific Comments:

R8) L31: Here and elsewhere, are you calculating the mass mean mobility diameter or number mean mobility diameter?

Mean number diameter-we've added 'number' here and 'number' and 'mobility' elsewhere in the paper when appropriate.

R9) L38: "...undergone more decreases in a marker for primary biomass burning organic aerosol." This is an awkward statement

We have changed this to
"We further find that on the plume edges, the organic aerosol is more oxygenated while a marker for primary biomass burning aerosol emissions has decreased in relative abundance."

R10) L41-44: "Smoke from biomass burning... influencing... as well as the health of smoke-impacted communities". "Smoke-impacted" is redundant.

We have removed the second 'smoke'.

R11) L45: Dilution is a process which is a central theme of the manuscript. It should have a proper description of what that process is. I suggest something like, "Dilution is the process where the plume mixes with clean background air, reducing concentration of fire emitted aerosols and gases". The current statement, "Dilution through entrainment..." is not explicit.

We have updated this statement to be more explicit:
"Dilution of a smoke plume occurs as the plume travels downwind, mixing with regional 'background' air, reducing the concentrations of the smoke aerosols and vapors and potentially driving changes in the physical and chemical properties of the emissions (Adachi et al., 2019; Akagi et al., 2012; Bian et al., 2017; Cubison et al., 2011; Hecobian et al., 2011; Hodshire et al., 2019a, 2019b; Jolleys et al., 2012, 2015; Konovalov et al., 2019; May et al., 2015; Noyes et al., 2020; Sakamoto et al., 2015)."

R12) L50-52: Lacks explanation of why large plumes dilute slower. Since this is so important to the story, it should have a better description. Currently, it is just stated that they do. Something like "...cores of larger plumes are protected from dilution due the physical distance from background air..." Citations for this are Garofalo et al (2019) and Lee et al (2020).

We have altered this to

"Large, thick plumes dilute more slowly than small, thin plumes for similar atmospheric conditions, as the cores of larger plumes are at a greater physical distance to the background air, shielding them from dilution for longer (Akagi et al., 2012; Bian et al., 2017; Cubison et al., 2011; Hecobian et al., 2011; Hodshire et al., 2019a, 2019b; Jolleys et al., 2012, 2015; Konovalov et al., 2019; May et al., 2015; Sakamoto et al., 2015, Lee et al., 2020, Garofalo et al., 2019)."

R13) L86-87: "... evaporation of vapors." Should be "evaporation of more volatile compounds."

We have updated the text to this suggestion.

R14) L88: "; plumes with higher concentrations will undergo more coagulation..." Are you referring to number concentration, mass concentration, or both?

Aerosol number; we have added this to the text.

R15) L101: "... differences in aerosol loading serve as a proxy for differences in dilution rates..." Do you mean rates or amount of dilution prior to first measurement? I have provided other comments in more detail regarding this assumption.

'Amount' is more appropriate here. We have updated this statement to be more inclusive of other points:

"The differences in aerosol loading serve as a proxy for differences in initial fire and plume sizes, mass fluxes, and subsequent amount of dilution. The extent to which dilution has occurred prior to the first observation is not a measurable quantity, and fire sizes and mass fluxes were not estimated as a part of the BBOP campaign."

R16) L103: "...given initial plume mass and physical age..." "mass" should be "OA mass concentration".

Added.

R17) L115: More description needs to be provided regarding the settings for the SP-AMS.

Was it equipped with a tungsten thermal vaporizer? If so, what temperature was the thermal vaporizer set to? HR-ToF, C-ToF, L-ToF, quadrupole? ToF set to V-mode or W-mode? How was data analyzed? Are you reporting UMR or HR results? Is f60 the levoglucosan fraction (i.e. fC2H4O2 as is discussed by Corbin et al 2015 and Lee et al 2020) or based on the UMR m/z=60 organic fragment after subtracting C5 contributions (also m/z=60, see Cubison et al 2011 and

Lee et al 2010). What was the MS timing? Open vs closed timing? PToF or ePToF mode? Pulse period, sampling Hz? What m/z range was scanned?

Please see our response to R2.

R18) L116: "PM1 aerosol masses" should be "aerosol mass concentration of PM1 (sub-micron particulates)..."

Done

R19) L119-121: How was collection efficiency determined?

Please see our response to R2.

R20) L119-121: It looks like the laser vaporizer was switched between on and off. How frequently was the laser vaporizer switched? Is the data presented in this manuscript with the laser on, off, or both?

These measure inherently different attributes of PM1 that may not be directly comparable or combinable.

Also, because the laser vaporizer fractionates aerosols molecular species differently than the thermal vaporizer (Onasch et al 2012; Corbin et al 2014; Canagaratna et al 2015; Lee et al 2015; Lee et al 2020) single ions such as C2H4O2+ and Org44+ (used to calculate f60 and f44) CANNOT be compared or combined between modes.

Please see our response to R2.

R21) L121-L122: "We do not attempt to characterize whether the collection efficiency of the SP-AMS changes as the aerosol ages"

Collection efficiency has been observed to change by a factor of 2 or more as BB POA grows in size and becomes more spherical (See Middlebrook et al 2012, Willis et al 2014, Corbin et al 2015 (ACP), Massoli et al 2015, Collier et al 2018). This change in CE has been observed to bias particles of different morphology/composition differently between different vaporizer modes (laser + oven versus oven-only), specifically affecting the CE of the laser more than the CE of the thermal vaporizer.

We have modified this and the following text: "There is evidence from other studies that the CE of the tungsten vaporizer (laser off mode) (Lim et al., 2019) and the laser vaporizer (laser on mode) (Willis et al., 2014) changes as a function of chemical composition, rBC coating thickness, size, and sphericity in laboratory studies (Middlebrook et al., 2012; Willis et al., 2014; Corbin et al., 2015; Massoli et al., 2015; Collier et al., 2018) and an aircraft study (Kleinmane et al., 2007). Results pertinent to changes in CE due to aging in smoke plumes are scarce (see discussion in Kleinman et al., 2020). We assume these CEs for the laser on and off modes are constant in space and time, which is a limitation of this study."

Kleinman, L. I., et al. (2007), Aircraft observations of aerosol composition and ageing in New England and Mid‐Atlantic States during the summer 2002 New England Air Quality Study field campaign, J. Geophys. Res., 112, D09310, doi:10.1029/2006JD007786.

R22) L123: "...CE has been recently observed to decrease with aging within a laboratory study..." should be "... decrease with increasing chemical age induced by UV light exposure and OH- equivalent to 10 photochemical days..."

This change in CE is likely irrelevant to this manuscript since the physical age of aerosols described in this study is generally less than 3 hours while the cited study compares CE over the equivalent of 10 days. As mentioned in the previous comment, there are a range of studies that have shown increases in CE as particles grow in size (which presumably also increase with age in the near-field during the particle growth phase) which is more relevant to this study.

We have removed this sentence as part of the revised text given in R21.

R23) L 124: "... no consistent evidence of changing CE in field studies exist yet."

There are lots of studies which show changing CE in the field. For example, see Collier et al (2018), Massoli et al (2015), and Middlebrook et al (2012) for examples of changing CE in field studies. Also see Willis et al (2014) and Corbin et al (2015, ACP) which report the same phenomena in laboratory studies.

We have removed this sentence as part of the revised text given in R21.

R24) L124-125: "We use the f60 and f44 fractional components..."

Here and elsewhere, f60 and f44 are referred to as if they are chemical species instead of parameters describing mass fractions.

For f50, the relevant OA species group is anhydrous sugars with the dominant species being levoglucosan ($C_6H_{10}O_5$) (Lee et al 2010; Cubison et al 2012). This species is indirectly observed by the SP-AMS as the fragment $C_2H_4O_2$ (m/z=60), a fragment of the levoglucosan molecule after the OA is vaporized and the vapors are ionized by a 70 eV electron supply. Similarly, OA observed at m/z 44 is the $CO_2$ fragment of, primarily, OOA after subtracting the gas [$CO_2$] mass.

We have updated this imprecise language throughout regarding how we refer to f44 and f60.

 R25) L128-129: "The f44 fractional component (arising from primarily CO2+...)"

This is another example of imprecise language. Suggest, "f44, the OA fractional component observed by the SP-AMS as the ion fragment CO2+, is a proxy for ..." The current wording suggests that f44 is a fraction of something else which isn't specified (i.e. f44/Org).

We've updated this as suggested
"$f_{44}$, the OA fractional component observed by the SP-AMS as the ion fragment $CO_2$+ as well as some acid groups, is a proxy for SOA arising from oxidative aging (Alfarra et al., 2004; Cappa and Jimenez, 2010; Jimenez et al., 2009; Volkamer et al., 2006)."

 R26) L129: extra semicolon

Fixed

 R27) L130-132: "...of semivolatile f60-containing species and addition of oxidized f44-containing..." Another example of imprecise language using f60 and f44. The aerosols contain levoglucosan and anhydrous sugars, not f60, and OOA, not f44.

We've updated this to
"...likely due to both evaporation and/or oxidation of semivolatile species that contribute to m/z 60 in the SP-AMS and addition of oxidized species that contribute to m/z 44 in the SP-AMS…"

 R28) L135: "changes in O/C and H/C are also influenced by..."
These other processes also affect all of the other parameters discussed in this paper.

We have updated this to
"Changes in O/C and H/C (as well as changes in total OA mass, number, $f_{44}$, and $f_{60}$) are also influenced by mixing of different air masses and co-oxidation of different VOC precursors (Chen et al. 2015)."

R29) L140: "provided CO measurements" should be "measured CO concentration" or "measured CO mixing ratio".

Updated to 'measured CO concentrations'

R30) L141: "An SPN1 radiometer provided total shortwave irradiance". The radiometer measured total shortwave irradiance, it did not provide or create the irradiance.

Updated 'provided' to 'measured'

R31) L143: "(Dx)" refers to the smoke contribution and should be placed after "species X from smoke" otherwise Dx is implied to refer to the background value of x. Also, the smoke doesn't contribute species X, it is comprised of species X. The fire emits/contributes X.

Updated to
"To determine the contribution to the concentrations of species X from smoke emissions ($\Delta X$)..."

R32) L145-146: This sentence needs to be rewritten. Suggest something like "Variability of the normalized emission ratio (Dx/DCO) along the lagrangian flight path implies production or removal of species X in the plume."

Changed to
"Increases or decreases of $\Delta X/\Delta CO$ along the Lagrangian flight path indicate whether the total amount of X in the plume has increased or decreased (implying production or removal) since time of emission."

R33) L147: "... average regional background for each species by using the lowest 10% of the CO data for..."

This statement reads as if you subtract a CO concentration from the number size distribution, OA, etc to get a background correction. It should say something like "... background values of X... were determined to correspond with time periods which displayed the lowest 10% of CO concentrations..."

We agree that this is confusing and have updated to
"The background concentration of X is determined as a regional average of the observed out-of-plume concentrations of X. To avoid using smoke-impacted measurements we apply a threshold of only using measurements of X that occur in regions that correspond to the lowest

10% of CO data. We determine the lowest 10% of CO concentrations from each flight during time periods with a similar altitude, latitude, and longitude as the smoke plume. We perform sensitivity calculations on our assumptions of background regions and discuss them in Sect. 3."

R34) L149: Should read "Mass concentrations of elemental O, H, and C were calculated..."

Fixed

R34.b) L152-153: It is not clear what you mean by this sentence.

We are unable to address this comment, as L152-153 in the original submission are equation 1 (where X = O or H):

$$\frac{\Delta X}{\Delta C} = \frac{(X_{in\ plume} - X_{out\ of\ plume})}{(C_{in\ plume} - C_{out\ of\ plume})} \qquad \text{Eq. 1}$$

R35) L154: "inside and outside of the plume".

Does this refer to sampling time periods/locations corresponding lowest 10% of CO or <150 ppbv? Why would you change between these definitions of background?

We have modified our explanation of how we quantify the background in R33. We use the two definitions in order to have a buffer between the plumes and background to ensure to the best of our ability that we are reporting in plume and out of plume values. We note that in our analyses where we split into different ΔCO bins we do not use the lowest 5% of the in-plume region, again to provide a buffer between in-plume and out-of-plume. We have added the following text:

"We note that we use different definitions of in-plume and background (i.e. the lowest 10% of CO measurements) in order to provide a buffer between the plume and background to ensure to the best of our abilities that we are capturing non-smoke-impacted air for the background and smoke-impacted air for in-plume cases. The regions of the lowest 10% of CO measurements always fall under 150 ppbv (Figs. S7-S11). Similarly, we exclude the lowest 5% of CO data in the in-plume measurements in our analyses to provide a further buffer between smoke-impacted and background air."

R36) L157-158: "We only consider data to be in-plume if the absolute CO>=150 ppbv, as comparisons of CO and the number concentration show that in-plume data has CO>150 ppbv and out of plume (background) data has CO < 150 ppbv."

Why did you change the definition of background from time/location corresponding to lowest 10% of CO to <150 ppbv?

See response to R35.

What do you mean by "comparisons of CO and the number concentrations"? What independent metric are you defining as "background" here? If number concentration is used to define time/location of background air, why not use that instead of CO <10% or CO < 150 ppbv? The logic here is circular or incomplete.

This statement was misleading, and we have removed it.

R37) L 162: "concentrations" should be singular.

We are unable to address this comment as line 162 does not include the word concentrations. We have looked for incorrect usage elsewhere and have not found any.

R38) L163: Should be "mobility" diameters.

Fixed

R39) L164: "...as the bulk of observed newly formed particles observed fell below 40 nm" Grammar.

Fixed

R40) How do you identify "newly formed particles" independent of the particle size? This implies that you observed newly formed particles >40 nm.

This statement is confusing and we have updated to
"Frequently, the background-corrected, normalized number concentration in the FIMS size range between 20-40 nm increases by 1-2 orders of magnitude relative to typical plume conditions, indicating possible nucleation events, primarily at the edges or in between smoke plumes (Figs. S7-S11)."

R41) L165-166: Grammar in sentence structure.

Changed to

"Smoke plumes contain particles with diameters larger than 262 nm (Janhäll et al., 2009): thus, we cannot provide total number concentrations, but we can infer how the evolution of $\Delta N/\Delta CO$ within our observed size range evolves."

R42) L187-188: "The centerline..." This sentence needs to be rewritten for clarity.

Rewritten to

"The centerline is subjectively chosen to approximately capture the most-concentrated portion of each plume pass (as estimated using total aerosol number concentrations)."

R43) L190: "...and this physical age is assumed to be constant across the transect, as the crossings took between 50-500 seconds."

While crossing the plume occurred in only a short time, were the transects always perfectly across flow? If not, then wouldn't the aerosol at different sampling times along a transect have different physical ages with larger uncertainty than just 50-500 seconds?

Good point, we have modified the sentence as follows:

"...and this physical age is assumed to be constant across the transect, as the crossings took between 50-500 seconds; however, transects that were not perfectly tangential to the mean wind would have sampled different plume ages on the opposite sides of the plume."

R44) L195: missing comma
Fixed

R45) L201-202: "thinnest (least CO-dense)... thickest (most CO-dense)..." Use either of the commonly accepted nomenclature of "CO mixing ratio" or "CO concentration".
Changed to "lowest CO mixing ratio" and "highest CO mixing ratio"

R46) L207: missing a verb between "plumes" and "from"
The original sentence is (section in question bolded, underlined)
*"We note that some plumes show more than one maxima in CO concentrations within a given plume crossing, which implies that there may be more than one fire or fire front, and that these **plumes from** separate fires or fronts are not mixing perfectly."*
We do not find a missing verb.

R47) L205-207: could the multiple peaks during a transect be explained by spatial variations in the plume structure, such as the core of the plume was higher in some areas than others causing the flight to dip below/above the core and then back into it?

This is a reasonable point. We have added the following:

"Multiple maxima could also imply vertical variations in the location of the core of the plumes that the flights did not sample."

R48) L213: It is hard to tell what the variability in DBC/DCO are since they are plotted on a log scale. They appear to vary by an order of magnitude, i.e. not constant as the authors suggest.

Yes. We have modified the text to be as follows:

"Figure 1 shows that for this specific plume, $\Delta OA/\Delta CO$ and $\Delta BC/\Delta CO$ systematically vary little with age for both the 5-15 and 90-100 percentile of $\Delta CO$ (p-values>0.5), yet both show non-systematic variability between transects."

R49) L213: I pointed this out in my general comments. Each dot in figure 1 is a single value that represents multiple measurements in space and time. How well does the value of any single datum in this figure represent the range of data it is derived from? You need to have error bars to show that variability.

See the response to R3.

R50) L215: "for each transect" should be "each transect set".
Fixed

R51) L218-219: "...it is apparent that the 5-15 and 9-90 percentiles do show a separation..."

This statement cannot be verified or supported without some idea of error bars, either representing propagated measurement uncertainty, variability of binned data, or both.

We have modified the text as follows:

"The remaining variables plotted also show some noise and few clear trends, but it is apparent that the transect-mean values 5-15 and 90-100 percentiles do show a separation for some of the individual metrics, in particular $\Delta f_{44}$ and $\Delta O/\Delta C$."

R52) L227-231: This is one of the key assumptions of the research, that the initial, background-corrected OA mass concentration can be used as a proxy for the degree of dilution of the plume. The authors provide no support for why DOAinitial would represent the degree of dilution even though this is the main storyline of their paper.

First, I do agree that the cores of larger plumes are likely protected from mixing with background air because of the distance between the core and the background air and it is well understood that some plume chemistry and mixing occurs very early after combustion+pyrolysis and prior to measurement. However, this needs to be presented differently. Start with the hypothesis that cores of larger plumes mix slower with background air and then use the observations to prove it by showing something like the rate of change in f60 and f44 as a function of physical age for plumes with different DOAinitial.

The current presentation is problematic. Think of two hypothetical smoke plumes that are identical in terms of dilution, photolytic reaction rate etc and were measured at the same physical age, but the corresponding fires had different OA emission factors (say, flux of OA from fire B was twice that of fire A). Fire B would have ~2x the measured DOAinitial compared to Fire A. This would instead be interpreted by the authors as having half the dilution of the plume from fire B instead of twice the OA emission.

This demonstrates that the author's assumption that DOAinitial is a proxy for plume dilution only makes sense if all fires measured emit OA at the same rate and concentration.

Please see the response to R4 for an extended discussion of this topic.

Specific to this comment, the example of the 2x difference in emission factors is fitting as 2x is around the range that OA emission factors vary by (Andreae, 2019), whereas our observed range of initial OA concentrations was nearly a factor of 100. OA emissions factors can only be a minor contributor to variability. Additional reasoning is given in response to R4 (starting on Page 13 of this response).

R53) L228-229: "(as presumably larger, more intensely burning fires will have larger mass fluxes than smaller...)".

This assumption is false.
Larger and more intense fires do not necessarily correspond to higher emission rates. Emissions of OA depend on a number of factors other than fire intensity (I assume you mean temperature). Hotter, more intense fires (i.e. flaming stage) can burn more efficiently and actually emit less OA than cooler, smoldering fires (Akagi et al 2011; McMeeking et al 2009; May et al 2014). Corbin et al (2015) found that in laboratory burn experiments the vast majority of OA emissions occurred in the "starting phase" before the logs fully caught fire.

The reviewer appears to be discussing emission factors, not emission rates (emission factor ⬚ fuel consumption rate per area ⬚ fire area). By intensity, we meant heat release per fire (i.e., fire radiative power [FRP] = fire radiative flux * fire area). FRP has been used to estimate smoke emissions from satellites (using assumed emissions factors), e.g. Ichoku and Ellison (2014). Further, it is unequivocal that having a larger-area fire with all else equal is going to emit more material than a smaller fire.

We have updated this to
"(as larger fires and fires with faster rates of fuel consumption per area will have larger mass fluxes than smaller fires or fires with less fuel consumption per area, all else equal)"

Ichoku, C. and Ellison, L.: Global top-down smoke-aerosol emissions estimation using satellite fire radiative power measurements, Atmos. Chem. Phys., 14, 6643–6667, https://doi.org/10.5194/acp-14-6643-2014, 2014.

 R54) L243: missing a comma between items in the list. L252: delete either "systematically" or "sequentially"
Both fixed

 R55) L251-257: See general comments regarding data analysis. The authors choose to combine all data together to determine the effects of aging and dilution on plume characteristics. However, they do not normalize the data in anyway or try another technique to separate these two effects. Normalizing a parameter to the first measured value (say f60) acknowledges that there are differences in f60 between plumes (maybe related to DOAinitial) and would allow for analysis of temporal trends after emission. One result maybe that the photolytic age of the aerosol mass (as measured by f60 or f44) is slower plumes with higher DOAinitial , i.e. there is less of a change/unit time of f60 or f44 or DN/DCO, etc.

Currently, the analysis is a regression comparing apples and oranges and the results are not meaningful.

Please see our response to R5.

 R56) L262-266: Include citations to Cubison et al (2011), Garofalo et al (2019), Forrister et al (2015), Lee et al (2020) for constant DOA/DCO as plumes age.

We are citing Cubison and Garofalo here. We have added the other two citations.

R57) L270-270: Containing text "...estimates of heterogeneous mass losses indicate that after three hours of aging for a range of OH concentrations and reactive uptake coefficients, over 90% of aerosol mass is anticipated to remain..."

This is the basis for which the authors interpret changing f44, f60 as relating to coagulation. However, this statement is only relevant to particle evolution after ~3 hours while nearly all of the observations occur within a physical age of 3 hours.

We do not follow the reviewer's thought process here. We do not interpret changes $f_{44}$ or $f_{60}$ 'as relating to coagulation' in this paper. To first order, coagulation does nothing to change $f_{44}$ or $f_{60}$ (there may be some minor indirect effects due to reducing particle surface area, hence changing condensation/evaporation fluxes), and to the sensitivity we have with the measurements, it only changes particle number.

The point of this statement is to say that little mass/composition changes should occur from het. chem over the 3 hours timespan that the measurements were taken. To make that more clear, we have updated this to
...estimates of heterogeneous mass losses indicate that after three hours of aging (the range of time the BBOP measurements were taken in) for a range of OH concentrations and reactive uptake coefficients...

R58) L281: "with more concentrated plumes". Be more specific by what you mean by more concentrated. Do you mean less diluted, higher mass concentrations of OA, higher CO mixing ratios?

Updated to "with plumes with higher $\Delta OA_{initial}$"

R59) L282-284: "(2) Differences in $\Delta f60$ and $\Delta f44$ for the nearest-to-source measurements indicate that evaporation and/or chemistry likely occurred before the time of these first measurements..."

It is well documented that the f60 and f44 of POA varies between fires (Cubison et al 2011; Jolleys et al 2015; Ortega et al 2013; McClure et al 2020). Since differences of POA emissions can explain variability in $\Delta f60$ and $\Delta f44$ for the nearest-to-source measurements, variability of these parameters can NOT be used as evidence of chemistry/evaporation in the smoke plume without knowing the actual f60 and f44 values of fresh POA.

 As well, the sentence in quotes has been rewritten:

"(2) The differences in $\Delta f_{60}$ and $\Delta f_{44}$ are apparent even for the nearest-to-source measurements that are ~15 minutes after the time of emission."

R60) L282-284 "...(assuming that emitted $\Delta f60$ and $\Delta f44$ do not correlate with $\Delta OA_{initial}$; there is currently no evidence for this alternative hypothesis)."

There is actually a lot of evidence that f60 and f44 can correlate with OA emissions. In laboratory studies, the evolution of emissions as fires progress from starting-to-flaming-to-smoldering has shown that levoglucosan emissions occur primarily at the starting phase by combustion of hemicellulose material which is also when the majority of OA emissions occur (Corbin et al 2015). Ortega et al (2015) and Lee et al (2010) also observed increased values of f60 in lab burns with higher OA emission factors. These laboratory studies support observations of smoke in ambient troposphere (e.g. Lee et al 2010; Aiken et al 2009; Lee et al 2020).

As you note, it is hard to measure the f60 of POA in ambient smoke. However, the lifetime of levoglucosan in the free troposphere is much longer than the age of aerosol in this study (<3 hours) which is probably why there is only a weak trend to lower f60 values with increasing physical age. So, your measurements of Df60 should be fairly representative of POA and your study (and your first point on lines 280-282 and repeated on line 319) are evidence that f60 is correlated with OA emission factors in wildfires.

 As well, the sentence in quotes has been rewritten and extended:
"Prior studies have shown that $f_{60}$ and $f_{44}$ at the time of emissions correlate with OA emissions factors through variability in burn conditions (Hennigan et al. 2011; Cubison et al. 2011; McClure et al. 2020), and this relationship might also contribute to our observed correlation between $\Delta f_{60}$ and $\Delta f_{44}$ with $\Delta OA_{initial}$. For this emissions relationship to be an important factor, the variability in the OA emission factor needs to be a significant contributor to the variability in $\Delta OA_{initial}$. If the relative variability in the OA emission factor is much smaller than the relative variability in $\Delta OA_{initial}$, other factors contributing to variability in $\Delta OA_{initial}$ will negate an emissions-based covariance between $\Delta OA_{initial}$ with $\Delta f_{60}$ and $\Delta f_{44}$. While our observed $\Delta OA_{initial}$ in Figure 2 spans nearly a factor of 100, Andreae (2019) shows that the OA emission factors have a $-1\sigma$ to $+1\sigma$ range of around a factor 3. Hence, variability in fuel consumption rates and dilution prior to the first transect likely dominate the variability in $\Delta OA_{initial}$, and the relationships of $\Delta f_{60}$ and $\Delta f_{44}$ with $\Delta OA_{initial}$ are unlikely to be influenced much by variability in burn conditions. We conclude that evaporation and/or chemistry prior to the first measurement appears to drive the initial relationship between $\Delta f_{60}$ and $\Delta f_{44}$ with $\Delta OA_{initial}$, consistent with (1) the theoretical work of Hodshire et al. (2019a), (2) an analysis of what chemistry would be missed in laboratory experiments if the initial 10-60 minutes of chemistry was not considered, following field experiments (Hodshire et al., 2019b), and (3) the recent field analysis (Palm et al., 2020)."

R61) L284: "Amounts" should be singular.

Fixed

R62) L291: Add citation Jolleys et al (2015).

Unable to determine the location asked for this reference placement.

R63) L313-314: delete "tends to be fairly constant or slightly decreasing with physical age and". Saying that it is poorly correlated is enough.

Removed

R64) L319-320: Evaporation does not happen from dilution. Evaporation will happen if the air is undersaturated (less than predicted vapor pressure of species X compared to equilibrium predicted by Henry's law). Here and elsewhere, please don't say that dilution causes evaporation, instead that dilution promotes evaporation.

Fixed

R65) L334: "NME is more variable..." Do you mean larger or higher? The NME is more variable between parameters, but that is meaningless.

Changed to larger

R66) L336: too many open brackets.
Fixed

R67) L337: What do you mean by "biomass burning modeling"? Are you referring to models of BB emissions, aerosol aging, fire spread?

Changed to "modeling of aging biomass burning aerosol"

R68) L343-345: Since you present a mulit-variate analysis here, what was the point of the past several pages discussing single variable correlation coefficients? Especially after you show that "Both physical age and DOAinitial appear to influence Df60, Df44, and DO/DC..." (Line 319 and a similar statement on L280-282).

Please see our response to R5. We have added text to discuss the synergy between the analysis on Figure 2 with that of Figure 3.

R69) L360-363: This statement needs more explanation and needs citations. Why would you expect plume regions with higher DOAinitial to have lower normalized number concentrations? As stated in the text, the driving process for this should be coagulation. ("*Although we would anticipate that plume regions with higher initial ΔOA would have lower normalized number concentrations due to coagulation"*) Coagulation is proportional to the square of the particle concentration for particles of the same size, and Sakamoto et al., (2016) showed that mass concentrations can be used to estimate the rates that the median diameter increases due to coagulation in biomass burning plumes.
Sakamoto, K. M., Laing, J. R., Stevens, R. G., Jaffe, D. A. and Pierce, J. R.: The evolution of biomass-burning aerosol size distributions due to coagulation: Dependence on fire and meteorological details and parameterization, Atmos. Chem. Phys., 16(12), 7709–7724, doi:10.5194/acp-16-7709-2016, 2016.
We have added that citation to this statement

To a first order, I would expect the opposite. That higher number concentrations would be observed with higher DOAinitial because the OA vapor pressure is higher and this promotes new particle nucleation because vapors are more likely to collide with other vapor molecules to nucleate than with existing particles to condense on (Lim et al 2019; and work from Neil Donahue's group).

High particle mass concentrations generally suppress nucleation because high mass concentrations correspond to a high condensation sink (Westervelt et al., 2014). A high condensation sink cuts down new particle formation through lowering condensable vapor supersaturations (reducing nucleation and growth rates) and increasing the coagulational losses of any clusters that form. Further, high concentrations of particles mean fast coagulation of the primary particles, reducing number concentrations.

We don't follow the rationale about the OA vapor pressure being higher when OA concentrations are high. Under high aerosol concentrations, the low-volatility OA material forming in the gas phase (that could be involved in nucleation) will have a lower ambient vapor pressure than under low aerosol concentrations because the condensation sink is faster. Higher concentrations of pre-existing particles do not promote new particle formation (there are documented cases where high concentrations of particles do not appear to suppress new particle formation, hypothesized to be to particle-phase mass transfer limitations on condensation, but we are not aware of any cases where high concentration of particles promotes new particle formation).

Westervelt, D.M., Pierce, J.R., Adams, P.J.: Analysis of feedbacks between nucleation rate, survival probability, and cloud condensation nuclei formation, Atmos. Chem. Phys., 14, 5577-5597, doi:10.5194/acp-14-5577-2014, 2014.

R70) L371-372: Decreasing normalized number concentrations are ascribed to coagulation. This contradicts the model that changes in f60 and f44 are due to evaporation of solid particulate balanced by condensation of more oxidized-OA described on line 277-279.

As stated in our response to R57) coagulation is not related to changes in f44 and f60. Coagulation reduces particle number and increases the average particle size. There is no contradiction between these statements: the first focuses on coagulation and the second on evaporation-chemistry-condensation.

R71) L381-382: Awkward and redundant sentence.

Unsure what sentence is in question here.

R72) L384-393: Discussion of nucleation-mode particles seems out of place here. Maybe move towards beginning of section 3.

After careful consideration, we have kept this subsection here. This subsection discusses particle number concentrations, and nucleation-mode particles are a part of that discussion.

Nucleation-mode particles are defined as 20-40nm. This needs a citation.
We are not defining the nucleation-mode particles to be 20-40 nm. We are inferring that nucleation is occurring due to the 'banana' behavior in the data that is observed in this size range. (The FIMS does not measure below 20 nm). Updated to
"Particles appear in the 20-40 nm size range in the FIMS measurements independently of plume OA concentrations (Figs S7-S11), implying that nucleation events may be occuring for some of the transects."

Earlier, a statement was made "bulk of observed newly formed particles observed fell below 40 nm" which implies that a fraction of newly formed particles were larger 40 nm. Was that a mis-statement? This would contradict your definition that nucleation-mode particles are 20-40 nm?

This phrase has been removed.

 R73) L388: "one day" should be one transect or transect set. L405-406: Awkward or redundant sentence.

Fixed to 'one set of transects'.
Updated to
Differences in initial values of $\Delta f_{60}$, $\Delta f_{44}$, and $\Delta O/\Delta C$ are evidence that evaporation and/or chemistry has likely occurred before the time of initial measurement and that plumes or plume regions with lower initial aerosol loading can undergo these changes more rapidly than thicker plumes.

 R74) L408-409: "indicate that evaporation and/or chemistry has likely occurred before the time of initial measurement..." See previous comments questioning validity of this statement.

We changed "indicate" to "are evidence"; see our earlier responses.

 R75) L437: Format of some citations need to be cleaned up.

We have checked and corrected the citations.

R76) Figure 1:

- See comments regarding adding error bars to show data variability of bin. There are a few points that are very different from the rest of the data set (such as in the DOA/DCO and DN/DCO datasets) which makes me think the single value representing the bin is inadequate.

- Please change DBC/DCO to a linear scale
Changed.

- Your values of f60 are pretty low for fresh BBOA. I am wondering if this is an issue with the SP-AMS settings or how the data was handled.

The f60 values in Figure 1 and Figure 2 are within the range of the closest transect of prior biomass burning field studies (Garofalo et al., 2019; Cubison et al., 2011) and lab studies (Hennigan et al., 2011).

Hennigan, C. J., Miracolo, M. A., Engelhart, G. J., May, A. A., Presto, A. A., Lee, T., Sullivan, A. P., McMeeking, G. R., Coe, H., Wold, C. E., Hao, W.-M., Gilman, J. B., Kuster, W. C., de Gouw, J., Schichtel, B. A., Collett Jr., J. L., Kreidenweis, S. M., and Robinson, A. L.: Chemical and physical transformations of organic aerosol from the photo-oxidation of open biomass burning emissions in an environmental chamber, Atmos. Chem. Phys., 11, 7669–7686, https://doi.org/10.5194/acp-11-7669-2011, 2011.

Cubison, M. J., Ortega, A. M., Hayes, P. L., Farmer, D. K., Day, D., Lechner, M. J., Brune, W. H., Apel, E., Diskin, G. S., Fisher, J. A., Fuelberg, H. E., Hecobian, A., Knapp, D. J., Mikoviny, T., Riemer, D., Sachse, G. W., Sessions, W., Weber, R. J., Weinheimer, A. J., Wisthaler, A., and Jimenez, J. L.: Effects of aging on organic aerosol from open biomass burning smoke in aircraft and laboratory studies, Atmos. Chem. Phys., 11, 12049–12064, https://doi.org/10.5194/acp-11-12049-2011, 2011.

Garofalo, L. A., Pothier, M. A., Levin, E. J. T., Campos, T., Kreidenweis, S. M., & Farmer, D. K. (2019). Emission and evolution of submicron organic aerosol in smoke from wildfires in the western United States. Acs Earth And Space Chemistry, 3, 1237-1247. doi:10.1021/acsearthspacechem.9b00125

- Need units for Dp axis, "[nm]".
Added.

R77) Figure 2:

- If you insist on combining all of the data together for a single regression, than you

  should not be drawing lines between points. Instead this should be a scatter plot with

  markers.

After careful consideration, we have determined to keep the lines. Without these lines the reader would not be able to follow individual sets of transects because some points from different sets would be colored similarly.

- Legend is inconsistent with figure 1. Either use "edge" or 5%<D[CO]<15%.

We have updated the figure to dCO to be consistent with the text and figure legend.

- Use of "[CO]" is inconsistent with text.

We have updated the figure to dCO to be consistent with the text and figure legend.

- Need units for Dp axis

Updated.

- Caption says that panels (d) and (g) have log axis but are plotted on linear axis. Panels

 (a) and (f) are plotted on log axis (also in corresponding figures in SI).

Fixed

- Font of "Dp" in caption is different than rest of fonts.

This was to get the overbar in Word. We anticipate that during the switch to LaTeX for typesetting, fonts will be homogenized.

- I think you should also provide a scatter plot of the first measurement of these

 parameters as a function DOAinitial.

We have made this figure as Figure S24 in the SI and added the following text to the main text:
"We include in the supporting information scatter plots of each parameter of Fig. 1 as a function of $\Delta OA_{initial}$ (Fig. S24), and observe no trends other than the cores of the plumes generally having a higher $\Delta OA_{initial}$ than the edges of the plumes, as expected."

[Figure]

Figure S24. Scatter plot of each parameter of Figure 1 against $\Delta OA_{initial}$.

- Needs error bars

Please see our response to R3.

R78) Figure 3:

• Spearman's correlations are not needed here.

We include both correlation coefficients and see no reason to remove one.

Supplemental Information:

R79) "...electrical mobility as in SMPS..." Should be "...similar to the operating principle of the SMPS..."

Fixed

R80) "...when size distribution suggests that particles smaller than 10 nm contribute negligibly..." Neither the FIMS or CPC 3010 are efficient at counting <10nm Dm particles, so why would the existence of those particles cause differences between the two instruments?

This was a typo--it should have been 20 nm and we have fixed it.

R81) "The SPAMS is thoroughly detailed in Kleinman et al. (2020)..." This still needs to be described here. At least summarizing the operating conditions of the SP-AMS.

We have greatly expanded this supplementary discussion on the SP-AMS (see our response to R2).

R82) "An SPN1 radiometer provided total shortwave irradiance..." It probably measured total shortwave irradiance, not "provided" or "created" the irradiance. This instrument needs to be described more. Maybe what exactly the measurements are and what they represent.

Changed to 'measured'. The papers cited provide instrument details, which is a standard practice in scientific papers.

R83) "... following parameters assumed for the calculation" missing a verb.

Fixed

R84) "Heterogeneous chemistry calculations:" There is no citation to justify the calculation. Is this a common methodology used? Has this methodology passed peer review?

Explained in R7. The collision rate calculation was from Seinfeld and Pandis (2006), and we added upper-bound assumptions about mass loss per reaction.

Seinfeld, J. H. and Pandis, S. N.: Atmospheric Chemistry and Physics, 2nd edn., John Wiley and Sons, New York, 2006.

R85) Fig S1: Colorbar label is missing an "]"

Fixed

R86) Fig S7-S11: Why are x-axis on the top and bottom panels different scales?

The bottom panel is a continuation of the top panel (in time, the x axis). Added this information explicitly to the caption.

Fig S13: should be moved to just before figure S19

These figures are in the order that they are referenced in the main text.

R87) Fig S19-S22: Caption mis-identifies which panels use a log scale.

Corrected

R88) Fig S24: Could you also plot the [CO] of each transect similar to S24-S26. I want to see that the absolute concentration is higher in the center of the plume than the edges and that the [CO] of the core decreases in each successive transect of the set to show dilution.

We have made this figure, Figure S27, and have added the following text to the main text: "To more clearly see this, Fig. S28 provides the same style of figure as Figs. S26-S27 for in-plume CO concentrations. Generally CO peaks around the centerline and is highest in the most fresh transect, but shows variability across transects."

[Figure]

Figure S27. Total in-plume CO (ppbv) irradiance for each flight along each transect included in this study. The titles indicate the flight. The black color indicates the earliest transect, with increasingly lighter colors indicating increasingly downwind transects. The centerline was estimated from the number size distribution and the estimated center of the fire (Figures S1-S6).

 R89) Fig S27: Need the 1:1, 1:2, and 0.5:1 line representing constant lines of oxidation. If it is arbitrary where the intersection of these lines is placed (as you cite from Heald et al 2010), then have an arbitrary intersection near the average of your data. The importance is the trends in H:C vs O:C. Alternatively, remove this figure.
We have removed this discussion and figure.

The Soot Particle – Aerosol Mass Spectrometer (SP-AMS) operating on the DOE G1 aircraft during BBOP has been described in detail by Collier et al. (2016), Sedlacek et al. (2018), and Kleinman et al. (2020). The SP-AMS sampled $PM_1$ through a constant pressure inlet operating at a pressure of ~620 Torr (Bahreini et al., 2008). The SP-AMS was equipped with dual vaporizers: (1) standard resistively heated tungsten vaporizer; and (2) 1064 nm intracavity laser vaporizer (Onasch et al., 2012). The standard tungsten vaporizer was operated at a nominal value of 600°C for the full data set. The SP-AMS operating with the laser vaporizer OFF is effectively the same as a standard HR-AMS, measuring non-refractory particulate matter (NR-PM). The SP-AMS operating in dual vaporizer mode, with both the standard tungsten vaporizer and the laser vaporizer ON measures the NR-PM and is additionally sensitive to refractory black carbon (rBC).

Flight data was collected at a rapid rate using "Fast-MS" in V-mode (i.e., mass spectral resolution ~2000) with 1 second sample time, with negligible particle time-of-flight (PTOF) data (DeCarlo et al., 2006; Lack et al., 2009). The pulsed, orthogonal extraction time-of-flight mass spectrometer (TOF-MS) was operated with a 60 μs pulser period and collected mass spectra from m/z 11 to m/z 955. "Fast-MS" data was collected in open (i.e., sample) mode for 52 seconds and in closed (i.e., background) mode for 8 seconds every minute. The laser vaporizer was operated by either automatically alternated laser ON and OFF each minute or manually sampling with the laser ON or OFF for long periods of time, such as full plume transects. The majority of the data (>76%) was collected in dual vaporizer mode (i.e., laser on).

The SP-AMS was calibrated for NR-PM with ammonium nitrate and for rBC with Regal black 8 independent times during BBOP. The average ionization efficiency (IE) with respect to ammonium nitrate was measured to be 8.1e-8 and the relative ionization efficiency (RIE) of rBC was measured to be 0.28, although the rBC from the SP-AMS was not used in this study.

Collier et al. (2016) determined the SP-AMS laser OFF collection efficiency (CE) to be 0.5 through comparisons with an independent HR-AMS located at the Mount Bachelor Observatory during over-flights. SP-AMS measured NR-PM values collected with the laser ON and OFF were compared for 16 different biomass burning plumes (Sedlacek et al., 2018; Kleinman et al., 2020). In each case, the plume was sampled with the laser ON and with the laser OFF, independently, and the measured plume NR-PM was normalized to CO to account for potential changes in the plume dilution between transects. The average ratio for NR-PM laser

ON to laser OFF was 1.52. From these results, the average CE of NR-PM measured with the laser ON to be 0.76 with a standard deviation of 0.07 (Sedlacek et al., 2018; Kleinman et al., 2020). There is substantial evidence in the published literature for the CE of the tungsten vaporizer (Lim et al., 2019) and the laser vaporizer (Willis et al., 2014) to change as a function of chemical composition and rBC coating thickness. Unfortunately for various reasons, instrument comparisons of measurements of $PM_1$ mass loading concentrations were very limited during BBOP, such that there does not exist a useful estimate of a changing CE for either SP-AMS vaporizer with changing plume conditions.

The SP-AMS data was analyzed using ToF-AMS Analysis Toolkit 1.61B and ToF-AMS HR Analysis 1.21B in Igor Pro. Gas phase carbon dioxide ($CO_2$) was directly measured on the G1 aircraft and was used to subtract gas phase contributions to $CO_2^+$ ion signal in the SP-AMS. SP-AMS standard NR-PM chemical species (i.e., Org, $SO_4$, $NO_3$, $NH_4$, Chl) were calculated using high resolution (HR) fits. $f_{44}$ and $f_{60}$ are unit mass resolution (UMR) ratios, whereas O:C ratios were derived using HR fits. Although it was not directly characterized for uncertainties during the BBOP campaign, we estimate uncertainties as follows. The AMS uncertainty is estimated following the methods in (Bahreini et al. 2009) (first equation of their supplemental information), leading to 37% uncertainty for organics. The laser vaporizer adds additional uncertainty up to 20%. Thus summing the uncertainties in quadrature leads to a 42% uncertainty in organics. (The Soot Photometer (SP2) had an uncertainty of 20%.)

We further analyzed the UMRs and the potential for laser ON specific ion signals to interfere with laser OFF NR-PM ion signals with the SP-AMS data. The chemical composition of the measured wildfire plumes during BBOP were > 90% NR-PM organic material (Collier et al., 2016; Kleinman et al., 2020). rBC mass fractions were typically below 2% (Kleinman et al., 2020), though the number fractions were higher (Sedlacek et al., 2018). Despite these low concentrations, the SP-AMS laser ON (relative to laser OFF) was observed to generate $C_n^+$ ion signals with an identifiable fragmentation pattern for rBC material and the laser ON to OFF NR-PM signal was observed to increase by ~50% on average. Similar results have been published for ambient urban aerosol (e.g., Lee et al. 2015). Recent laboratory work to investigate these issues has eliminated laser alignment issues and indirect heating as potential causes for these observations (Avery et al., 2020). Thus, these observations are likely due to a combination of different collection efficiencies (CEs) and relative ionization efficiencies (RIEs) for the two vaporizers when used in dual vaporizer mode (i.e., laser ON).

The HR ion signals at m/z 44 are dominated by $CO_2^+$ and $C_2H_4O^+$ ions (Fig. S31). The ratio of $C_2H_4O^+/CO_2^+$ increases with plume mass loading (i.e., concentration) and decreases with distance from the fire (Fig. S31), inline with the observations reported here for decreases in oxidation levels as a function of dilution. The HR ion signals at m/z 60 are dominated by $C_2H_4O_2^+$ and $C_5^+$ (Fig. S32). HR fitting of $C_5^+$ indicated that it averaged ~6% of the $C_2H_4O_2^+$ ion signal, independent of the laser vaporizer state (i.e., ON or OFF). For large $C_2H_4O_2^+$ ion signals in relatively undiluted biomass burning plumes, this ratio is likely controlled by the errors in fitting a small peak in the wings of a larger peak (Corbin et al., 2014). At lower ion signal levels, the $C_5^+$/ $C_2H_4O_2^+$ becomes significantly noisier, but the average does not change significantly. Laser ON may slightly increase the average ratio at lower $C_2H_4O_2^+$ ion signals, which could overestimate $f_{60}$ for relatively dilute plumes. If this were true, the observed decrease in $f_{60}$ with plume dilution (i.e., due to fire size and atmospheric age) would be slightly smaller than reported here.

       Past research on SP-AMS ion signals from the laser vaporizer and the standard tungsten vaporizer have identified several complicating factors when operating the SP-AMS in dual vaporizer mode. First, organic material coating rBC particles and detected using the laser vaporizer have noted different fragmentation patterns (Onasch et al., 2012) and chemical compositions (Canagaratna et al., 2015) compared with the same organic material detected using the standard tungsten vaporizer. Further, there are reports of SP-AMS laser vaporizer detecting refractory $CO_2^+$ ions from rBC particles (Corbin et al., 2014). Currently, we have not assessed the potential for refractory $CO_2^+$ ion signals during BBOP as both the rBC and Org signals are highly correlated in biomass burning plumes, making minor changes to these ratios difficult to ascertain. To address the question of whether the laser vaporizer generated different ion signals from similar organic compounds, we analyzed the laser ON and OFF plume transect pairs that were used for determining laser ON CE values relative to laser OFF.

       As shown in Fig. S33, the HR O:C, UMR $f_{44}$, and UMR $f_{60}$ ratios are highly correlated between laser ON and OFF conditions, though differ by apparent factors. Laser ON HR O:C ratios are approximately 4% lower than laser OFF. In large part, this is due to the UMR $f_{44}$ ratios, which are dominated by $CO_2^+$ ions, being 17% lower for laser ON. UMR $f_{60}$ ratios are 18% higher in laser ON than OFF. These observations are in line with the published results from Canagaratna et al., (2015), which observed that laser vaporizer only HR O:C ratios were ~17% lower than tungsten vaporizer only HR O:C ratios for the same organic material and the HR H:C ratios were ~16% higher. In the case of BBOP, the laser vaporizer signals represented approximately 1/3 of the total organic signal with dual vaporizers. The BBOP measured 4% lower HR O:C ratios are similar in magnitude to 5.6% (i.e., 0.33*17%) expected if the Canagaratna et al. (2015) results applied to BBOP measurements.

       The BBOP SP-AMS data used in this manuscript is used to measure trends in OA. O:C, $f_{44}$, and $f_{60}$ with plume dilution, either at different plume ages and/or different concentration percentiles across a biomass plume (i.e., edge vs. center). A question is whether the mixing of laser ON and OFF data here somehow biases the results due to the different absolute values between the two different states. A quick extension of the above plume pair analysis (Fig. S33) includes several "background" measurements made between the plumes (i.e., below 150 ppbv CO) and compared for laser ON vs. OFF to investigate if this ratio changes substantially between plume (i.e., high level) and background (i.e., low level) levels. The laser ON:OFF ratios of measured HR O:C averaged 0.95±0.049 in background and 0.96±0.029 in plume, UMR $f_{44}$ averaged 0.89±0.085 in background and 0.85±0.068 in plume, and UMR $f_{60}$ averaged 1.17±0.23

in background and 1.15±0.13 in plume. These results suggest that the observed laser ON/OFF ratios do not change from low to high signal levels, such that the trends observed for laser OFF should hold for laser ON, and vice versa. Further, the laser ON vs. OFF data points are randomly distributed throughout the measurements rather than systematically distributed to near- vs. far-field measurements or core vs. edge measurements. Hence, there should be no systematic bias due to the use of the combined laser ON and OFF data, although this combination of laser-on and -off data may contribute to noise in the observed trends.

**Text S2. Heterogeneous chemistry calculations**

We test the impact of heterogeneous chemistry on aerosol mass loss within the smoke plume. We performed a simple calculation of reactive uptake of OH molecules with particle-phase organics that resulted in loss of organic products. These calculations include assumed values of particle diameter, OH concentration, OH diffusion coefficient, and OH reactive uptake coefficient.  The following parameters are assumed for the calculations:

- OH diffusivity = 3.5e-5 [$m^2$ $s^{-1}$]
- Particle diameter varied from 1 - 1000 [nm]
- Constant OH concentration varied from 1e5 to 5e7 [molecules $cm^{-3}$]
- Reactive uptake coefficients varied from 0.1 to 1 [unitless]
- Molecular weight of organics = 200 [g $mol^{-1}$]
- Density of organics = 1.4 [g $cm^{-3}$]
- Total run time = 3 [hours]

The collision rate of OH with the particle surface was calculated using the condensation equations in Seinfeld and Pandis (2006). As a  calculation of the upper bound limit of evaporation due to heterogeneous chemistry, we assume each collision results in removing an organic molecule on the surface of the particle (assumed to be 200 amu), fragmenting and removing the molecule from the particle. The fragmentation products are not assumed to participate in further reaction. Figure S23a shows the resulting final:initial mass ratios after four hours of aging, indicating that for the aerosol sizes containing most of the mass  in this study (>10$0$ nm) and under expected OH concentrations (<10$^7$ $cm^{-3}$), >90% of the aerosol mass remains after 3 hours in all but the cases with a reactive uptake coefficient of 1 and an OH concentration of 10$^7$ $cm^{-3}$. Note however that (1) the reactive uptake coefficient is likely lower than 1  (Slade and Knopf, 2013), (2) not every reaction will lead to complete evaporation of all products, and (3) OH concentrations are often lower than 10$^7$ $cm^{-3}$ (Juncosa Calahorrano et al., 2020).

Avery, A.M., Williams, L.R., Fortner, E.C., Robinson, W.A., and Onasch, T.B.: Particle detection using the dual-vaporizer configuration of the soot particle Aerosol Mass Spectrometer (SP-AMS). Aerosol Sci. Technol., doi:10.1080/02786826.2020.1844132, 2020.

Bahreini, R.; Dunlea, E. J.; Matthew, B. M.; Simons, C.; Docherty, K. S.; DeCarlo, P. F.; Jimenez, J. L.; Brock, C. A.; Middlebrook, A. M.: Design and Operation of a Pressure-Controlled Inlet for Airborne Sampling with an Aerodynamic Aerosol Lens. Aerosol Sci. Technol. 42 (6), 465−471, 2008

Badosa, J., Wood, J., Blanc, P., Long, C. N., Vuilleumier, L., Demengel, D. and Haeffelin, M.: Solar irradiances measured using SPN1 radiometers: uncertainties and clues for development, Atmospheric Measurement Techniques, 7, 4267–4283, 2014.

Bahreini, R., Ervens, B., Middlebrook, a. M., Warneke, C., de Gouw, J. a., DeCarlo, P.F., Jimenez, J.L., Brock, C. a., Neuman, J. a., Ryerson, T.B., Stark, H., Atlas, E., Brioude, J., Fried, A., Holloway, J.S., Peischl, J., Richter, D., Walega, J., Weibring, P., Wollny, a. G., and Fehsenfeld, F.C.: Organic aerosol formation in urban and industrial plumes near Houston and Dallas, Texas. J. Geophys. Res., 114:D00F16, 2009.

Canagaratna, M.R., Massoli, P., Browne, E.C., Franklin, J.P., Wilson, K.R., Onasch, T.B., Kirchstetter, T.W., Fortner, E.C., Kolb, C.E., Jayne, J.T., Kroll, J.H., and Worsnop, D.R.: Chemical Compositions of Black Carbon Particle Cores and Coatings via Soot Particle Aerosol Mass Spectrometry with Photoionization and Electron Ionization. J. Phys. Chem. A, 119(19):4589–4599, 2015.

Collier, S., Zhou, S., Onasch, T.B., Jaffe, D.A., Kleinman, L., Sedlacek, A.J., Briggs, N.L., Hee, J., Fortner, E., Shilling, J.E., Worsnop, D., Yokelson, R.J., Parworth, C., Ge, X., Xu, J., Butterfield, Z., Chand, D., Dubey, M.K., Pekour, M.S., Springston, S., and Zhang, Q.: Regional Influence of Aerosol Emissions from Wildfires Driven by Combustion Efficiency: Insights from the BBOP Campaign. Environ. Sci. Technol., 50(16):acs.est.6b01617, 2016.

Corbin, J.C., Sierau, B., Gysel, M., Laborde, M., Keller, A., Kim, J., Petzold, A., Onasch, T.B., Lohmann, U., and Mensah, A. A.: Mass spectrometry of refractory black carbon particles from six sources: carbon-cluster and oxygenated ions. Atmos. Chem. Phys., 14(5):2591–2603, 2014.

Corbin, J., Othman, A., D. Allan, J., R. Worsnop, D., D. Haskins, J., Sierau, B., Lohmann, U., and A. Mensah, A. (2015). Peak-fitting and integration imprecision in the Aerodyne aerosol mass spectrometer: effects of mass accuracy on location-constrained fits. Atmos. Meas. Tech., 8(11):4615–4636.

DeCarlo, P.F., Kimmel, J.R., Trimborn, A., Northway, M.J., Jayne, J.T., Aiken, A.C., Gonin, M., Fuhrer, K., Horvath, T., Docherty, K.S., Worsnop, D.R., and Jimenez, J.L.: Field-Deployable, High-Resolution, Time-of-Flight Aerosol Mass Spectrometer. Anal. Chem., 78(24):8281–8289, 2006.

Juncosa Calahorrano, J. F., Lindaas, J., O'Dell, K., Palm, B. B., Peng, Q., Flocke, F., Pollack, I. B., Garofalo, L. A., Farmer, D. K., Pierce, J. R., Collett, J. L., Weinheimer, A., Campos, T., Hornbrook, R. S., Hall, S. R., Ullmann, K., Pothier, M. A., Apel, E. C., Permar, W., Hu, L., Hills, A. J., Montzka, D., Tyndall, G., Thornton, J. A. and Fischer, E. V.: Daytime Oxidized Reactive Nitrogen Partitioning in Western U.S. Wildfire Smoke Plumes, J. Geophys. Res. Atmos., 1–47, doi:10.1029/2020jd033484, 2020.

Kleinman, L. I., Sedlacek, A. J., III, Adachi, K., Buseck, P. R., Collier, S., Dubey, M. K., Hodshire, A. L., Lewis, E., Onasch, T. B., Pierce, J. R., Shilling, J., Springston, S. R., Wang, J., Zhang, Q., Zhou, S. and Yokelson, R. J.: Rapid Evolution of Aerosol Particles and their Optical Properties Downwind of Wildfires in the Western U.S, Aerosols/Field Measurements/Troposphere/Physics (physical properties and processes), doi:10.5194/acp-2020-239, 2020.

Lack, D. A.; Corbett, J. J.; Onasch, T.; Lerner, B.; Massoli, P.; Quinn, P. K.; Bates, T. S.; Covert, D. S.; Coffman, D.; Sierau, B.; Herndon, S.; Allan, J.; Baynard, T.; Lovejoy, E.; Ravishankara, A. R.; Williams, E.: Particulate emissions from commercial shipping: Chemical, physical, and optical properties. J. Geophys. Res. 114 (D7), D00F04, 2009.

Lee, A.K.Y., Willis, M.D., Healy, R.M., Onasch, T.B.B., and Abbatt, J.P.D.: Mixing state of carbonaceous aerosol in an urban environment: single particle characterization using the soot particle aerosol mass spectrometer (SP-AMS). Atmos. Chem. Phys., 15(4):1823–1841, 2015.

Lim, C.Y., Hagan, D.H., Coggon, M.M., Koss, A.R., Sekimoto, K., de Gouw, J., Warneke, C., Cappa, C.D., and Kroll, J.H.: Secondary organic aerosol formation from the laboratory oxidation of biomass burning emissions. Atmos. Chem. Phys., 19(19):12797–12809, 2019.

Long, C. N., Bucholtz, A., Jonsson, H., Schmid, B., Vogelmann, A. and Wood, J.: A Method of Correcting for Tilt from Horizontal in Downwelling Shortwave Irradiance Measurements on Moving Platforms, The Open Atmospheric Science Journal, 4(1), 78–87, doi:10.2174/1874282301004010078, 2010.

Onasch, T. B., Trimborn, A., Fortner, E. C., Jayne, J. T., Kok, G. L., Williams, L. R., Davidovits, P. and Worsnop, D. R.: Soot particle aerosol mass spectrometer: Development, validation, and initial application, Aerosol Sci. Technol., 46(7), 804–817, doi:10.1080/02786826.2012.663948, 2012.

Sedlacek III, A.J., Buseck, P.R., Adachi, K., Onasch, T.B., Springston, S.R., and Kleinman, L.: Formation and evolution of tar balls from northwestern US wildfires. Atmos. Chem. Phys., 18(15):11289–11301, 2018.

Seinfeld, J. H. and Pandis, S. N.: Atmospheric Chemistry and Physics, 2nd edn., John Wiley and Sons, New York, 2006.

Slade, J. H. and Knopf, D. A.: Heterogeneous OH oxidation of biomass burning organic aerosol surrogate compounds: assessment of volatilisation products and the role of OH concentration on the reactive uptake kinetics, Phys. Chem. Chem. Phys., 15(16), 5898–5915, 2013.

Wang, J., Pikridas, M., Spielman, S. R. and Pinterich, T.: A fast integrated mobility spectrometer for rapid measurement of sub-micrometer aerosol size distribution, Part I: Design and model evaluation, J. Aerosol Sci., 108, 44–55, 2017.

Willis, M.D., Lee, A.K.Y., Onasch, T.B., Fortner, E.C., Williams, L.R., Lambe, A.T., Worsnop, D.R., and Abbatt, J.P.D.: Collection efficiency of the soot-particle aerosol mass spectrometer (SP-AMS) for internally mixed particulate black carbon. Atmos. Meas. Tech., 7(12):4507–4516, 2014.

Table S1. Flight description table.

| Flight name, date | Number of sets of pseudo-Lagrangian transects | Fire name | Fuel[1] |  |
|---|---|---|---|---|
| '726a', 07-26-2013 | 2 | Mile Marker 28 | grasslands, shrub brush, | |

| | | | timber, and timber litter | |
|---|---|---|---|---|
| '730a', 07-30-2013 | 1 | Colockum Tarps | grass, trees | |
| '730b', 07-30-2013 | 2 | Colockum Tarps | grass, trees | |
| '809a', 08-09-2013 | 1 | Colockum Tarps | grass, trees |  |
| '821b', 08-21-2013 | 1 | Government Flats | |  |

[1]When known
[2]Instruments relevant to this study

Table S2. Calculated $R_{\Delta OA,initial}$ and $R_{age}$ values for $\Delta OA/\Delta CO$, $\Delta f_{60}$, $\Delta f_{44}$ $\Delta H/\Delta C$, $\Delta O/\Delta C$, $\Delta N/\Delta CO$, and $D_p$ when one flight is left out of the statistical analysis. We include the original R values as the first row for comparison. Red values indicate that the correlation has improved compared to all flights in the statistical analysis (closer to ±1). Blue values indicate that the correlation has worsened (closer to 0) compared to all flights in the statistical analysis. Black values denote no change in the correlation compared to all flights in the statistical analysis. Note that for flights '726a' and '730b' both sets of Lagrangian transects have been left out.

| ΔOA/ΔCO | | |
|---|---|---|
| Flight left out, date | Resulting $R_{\Delta OA,\ initial}$ | Resulting $R_{age}$ |
| None | +0.02 | +0.03 |
| '726a', 07-26-2013 | +0.12 | 0.0 |
| '730a', 07-30-2013 | +0.02 | +0.07 |
| '730b', 07-30-2013 | +0.17 | 0.0 |
| '809a', 08-09-2013 | -0.25 | +0.02 |
| '821b', 08-21-2013 | +0.05 | +0.03 |
| $\Delta f_{60}$ | | |
| Flight left out, date | Resulting $R_{\Delta OA,\ initial}$ | Resulting $R_{age}$ |
| None | +0.43 | -0.26 |
| '726a', 07-26-2013 | +0.58 | -0.38 |
| '730a', 07-30-2013 | +0.39 | -0.37 |
| '730b', 07-30-2013 | +0.52 | -0.19 |
| '809a', 08-09-2013 | +0.3 | -0.21 |
| '821b', 08-21-2013 | +0.4 | -0.26 |
| $\Delta f_{44}$ | | |
| Flight left out, date | Resulting $R_{\Delta OA,\ initial}$ | Resulting $R_{age}$ |
| None | -0.55 | +0.5 |
| '726a', 07-26-2013 | -0.63 | +0.4 |
| '730a', 07-30-2013 | -0.62 | +0.54 |
| '730b', 07-30-2013 | -0.45 | +0.46 |
| '809a', 08-09-2013 | -0.54 | +0.54 |
| '821b', 08-21-2013 | -0.42 | +0.57 |
| ΔH/ΔCO | | |

| Flight left out, date | Resulting R$_{\Delta OA, initial}$ | Resulting R$_{age}$ |
|---|---|---|
| None | -0.04 | -0.06 |
| '726a', 07-26-2013 | -0.04 | -0.12 |
| '730a', 07-30-2013 | -0.13 | -0.2 |
| '730b', 07-30-2013 | 0.0 | -0.16 |
| '809a', 08-09-2013 | 0.02 | -0.01 |
| '821b', 08-21-2013 | -0.01 | -0.05 |
| **ΔO/ΔCO** | | |
| Flight left out, date | Resulting R$_{\Delta OA, initial}$ | Resulting R$_{age}$ |
| None | -0.45 | +0.56 |
| '726a', 07-26-2013 | -0.54 | +0.46 |
| '730a', 07-30-2013 | -0.52 | +0.55 |
| '730b', 07-30-2013 | -0.21 | +0.54 |
| '809a', 08-09-2013 | -0.5 | +0.61 |
| '821b', 08-21-2013 | -0.32 | +0.63 |
| **ΔN/ΔCO** | | |
| Flight left out, date | Resulting R$_{\Delta OA, initial}$ | Resulting R$_{age}$ |
| None | -0.03 | -0.27 |
| '726a', 07-26-2013 | -0.03 | -0.13 |
| '730a', 07-30-2013 | -0.03 | -0.3 |
| '730b', 07-30-2013 | -0.21 | -0.43 |
| '809a', 08-09-2013 | -0.07 | -0.2 |
| '821b', 08-21-2013 | 0.0 | -0.37 |
| $\overline{D_p}$ | | |
| Flight left out, date | Resulting R$_{\Delta OA, initial}$ | Resulting R$_{age}$ |

| | | |
|---|---|---|
| None | -0.15 | +0.53 |
| '726a', 07-26-2013 | -0.18 | +0.43 |
| '730a', 07-30-2013 | -0.17 | +0.57 |
| '730b', 07-30-2013 | +0.19 | +0.63 |
| '809a', 08-09-2013 | -0.28 | +0.52 |
| '821b', 08-21-2013 | -0.18 | +0.52 |

Table S3. Fit coefficients *a, b,* and *c* for the fits shown in Fig. 3 , equation 4. The units of *a* are (metric), but note that the units of $\Delta OA_{initial}$ must be µg m$^{-3}$; the units of *b* are (metric)/hr, and the units of *c* are (metric), where (metric) = the units of $\Delta f_{60,} \Delta f_{44}$, $\Delta O/\Delta C$, or $\overline{D_p}$ , respectively.

| Metric | a | b | c |
|---|---|---|---|
| $\Delta f_{60}$ | 2.8e-03 | -6.4e-04 | 4.7e-03 |
| $\Delta f_{44}$ | -1.1e-02 | 5.8e-03 | 4.4e-02 |

| | | | |
|---|---|---|---|
| ΔO/ΔC | -3.6e-02 | 2.6e-02 | 0.24 |
| $\overline{D_p}$ | -1.5 | 10 | 150 |

Table S4. Fit coefficients *a, b,* and *c* for the fits shown in Fig. S28 , equation 5. The units of *a* are (metric); the units of *b* are (metric)/hr, and the units of *c* are (metric), where (metric) = the units of $\Delta f_{60,}$ $\Delta f_{44}$, ΔO/ΔC, or $\overline{D_p}$ , respectively.

| Metric | *a* | *b* | *c* |
|---|---|---|---|
| $\Delta f_{60}$ | 0.14 | -6.6e-02 | -5.3 |
| $\Delta f_{44}$ | -0.14 | 0.11 | -2.9 |
| ΔO/ΔC | -7.3e-02 | 6.1e-02 | -1.3 |

| | | | |
|---|---|---|---|
| $\overline{D_p}$ | -6.3e-03 | 4.0e-02 | 5.1 |

Table S5. Fit coefficients *a, b,* and *c* for the fits shown in Fig. S29 , equation 4 (but with $\Delta N_{initial}$ in place of $\Delta OA_{initial}$). The units of *a* are (metric); the units of *b* are (metric)/hr, and the units of *c* are (metric), where (metric) = the units of $\Delta f_{60,}$ $\Delta f_{44}$, $\Delta O/\Delta C$, or $\overline{D_p}$ , respectively.

| Metric | *a* | *b* | *c* |
|---|---|---|---|
| $\Delta f_{60}$ | 2.0e-03 | -5.4e-04 | -1.5e-03 |
| $\Delta f_{44}$ | -1.1e-02 | 5.3e-03 | 8.4e-02 |
| $\Delta O/\Delta C$ | -4.1e-02 | 2.4e-02 | 0.4 |
| $\overline{D_p}$ | -3.5 | 10 | 160 |

[Figure]

[Figure]

Figure S1. The flight path for flight '730b', colored by the FIMS total number concentration. The red dots are MODIS fire/thermal anomalies. The black star indicates the approximate center of the fire and the black dashed line indicates the approximate centerline of the plume, estimated by the number concentration.

[Figure]

Figure S2. The flight path for '726a'. Top two panels: the legs used in this study are colored by each ΔCO percentile bin used in the main text analyses. The green traces indicate the locations of the lowest 10% of CO, used to compute averaged backgrounds for this flight. Bottom two panels: the flight track colored by time since take-off in minutes. The numbers indicate the leg numbers as identified in the BBOP database. There were two complete flight paths for this day. The red dots are MODIS fire/thermal anomalies. The black star indicates the approximate center of the fire and the black dashed line indicates the approximate centerline of the plume, estimated by the number concentration.

[Figure]

Figure S3. The flight path for '730a'. Top panel: the legs used in this study are colored by each ΔCO percentile bin used in the main text analyses. The green traces indicate the locations of the lowest 10% of CO, used to compute averaged backgrounds for this flight. Bottom panel: the flight track colored by time since take-off in minutes. The numbers indicate the leg numbers as identified in the BBOP database. The red dots are MODIS fire/thermal anomalies. The black star indicates the approximate center of the fire and the black dashed line indicates the approximate centerline of the plume, estimated by the number concentration.

[Figure]

Figure S4. The flight path for '730b'. Top two panels: the legs used in this study are colored by each ΔCO percentile bin used in the main text analyses. The green traces indicate the locations of the lowest 10% of CO, used to compute averaged backgrounds for this flight. Bottom two panels: the flight track colored by time since take-off in minutes. The numbers indicate the leg numbers as identified in the BBOP database. There were two complete flight paths for this flight. The red dots are MODIS fire/thermal anomalies. The black star indicates the approximate center of the fire and the black dashed line indicates the approximate centerline of the plume, estimated by the number concentration.

[Figure]

Figure S5. The flight path for '809a'. Top panel: the legs used in this study are colored by each ΔCO percentile bin used in the main text analyses. The green traces indicate the locations of the lowest 10% of CO, used to compute averaged backgrounds for this flight. Bottom panel: the flight track colored by time since take-off in minutes. The numbers indicate the leg numbers as identified in the BBOP database. The Worldview image for this day had clouds over the fire location at the time of the satellite passover. Thus we estimate a fire center using Worldview and MODIS images for this region on the previous day (8-08-2013) (light green star) and the following day (8-10-2013) (salmon-colored star). The black star indicates our estimated center of the fire on 8-09-2013 and the black dashed line indicates the approximate centerline of the plume, estimated by the number concentration.

[Figure]

Figure S6. The flight path for '821b'. Top panel: the legs used in this study are colored by each ΔCO percentile bin used in the main text analyses. The green traces indicate the locations of the lowest 10% of CO, used to compute averaged backgrounds for this flight. Bottom panel: the flight track colored by time since take-off in minutes. The numbers indicate the leg numbers as identified in the BBOP database. The red dots are MODIS fire/thermal anomalies. The black star indicates the approximate center of the fire and the black dashed line indicates the approximate centerline of the plume, estimated by the number concentration.

[Figure]

Figure S7. Number size distribution data, dN/dlogD$_p$, from the FIMS; CO (white solid line); and total short wave (SW) irradiance (black dots) data for the '726a' flight. The bottom panel is a continuation in time from the top panel. The dotted dashed line indicates CO=150 ppb, our cutoff for in-plume/out-of-plume. The second set of Lagrangian transects for this flight start at the plume at approximately 86 minutes into the flight.

[Figure]

Figure S8. Number size distribution data, dN/dlogD$_p$, from the FIMS; CO (white solid line); and total short wave (SW) irradiance (black dots) data for the '730a' flight. The bottom panel is a continuation in time from the top panel. The dotted dashed line indicates CO=150 ppb, our cutoff for in-plume/out-of-plume.

[Figure]

Figure S9. Number size distribution data, dN/dlogD$_p$, from the FIMS; CO (white solid line); and total short wave (SW) irradiance (black dots) data for the '730b' flight. The bottom panel is a continuation in time from the top panel. The dotted dashed line indicates CO=150 ppb, our cutoff for in-plume/out-of-plume. For this figure, the top panel contains all of the first Lagrangian set of flight transects, and the bottom panel contains all of the second Lagrangian set of flight transects.

[Figure]

Figure S10. Number size distribution data, dN/dlogD$_p$, from the FIMS; CO (white solid line); and total short wave (SW) irradiance (black dots) data for the '809a' flight. The bottom panel is a continuation in time from the top panel. The dotted dashed line indicates CO=150 ppb, our cutoff for in-plume/out-of-plume.

[Figure]

Figure S11. Number size distribution data, dN/dlogD$_p$, from the FIMS; CO (white solid line); and total short wave (SW) irradiance (black dots) data for the '821b' flight. The bottom panel is a continuation in time from the top panel. The dotted dashed line indicates CO=150 ppb, our cutoff for in-plume/out-of-plume.

[Figure]

Figure S12. FIMS data for '809a' for the two legs that ~overlap (Figure S5) for the 51, 106, and 219 nm size bins. The solid line is from the plane flying north to south (right to left in this figure) and the dashed line is from the plane flying south to north (left to right in this figure). In the absence of FIMS measurement artifacts, we expect these two lines to roughly match each other. Each y axis is in units of number in bin.

[Figure]

Figure S13. Same as Figure 2 but using only the first 50% of data for each leg of the FIMS and CO data for panels f-g.

[Figure]

Figure S14. Aerosol properties for the first set (left-hand column) and second set (right-hand column) of pseudo-Lagrangian transects from flight '726a' (a-b) ΔOA/ΔCO (right y-axis) and ΔBC/ΔCO (left y-axis), (c-d) $\Delta f_{60}$ (right y-axis) and $\overline{\Delta f_{44}}$ (left y-axis), (e-f) ΔH/ΔC (right y-axis) and ΔO/ΔC (left y-axis),  (g-h) ΔN/ΔCO, and (i-j) $\overline{D_p}$ against physical age. For each transect, the data is divided into edge (the lowest 5-15% of ΔCO data; red points), core (90-100% of ΔCO data; blue points), and intermediate regions (15-50% and 50-90% of ΔCO data; light green and dark green points). ΔBC/ΔCO is shown in log scale and the x-axis for the right-hand column has been shifted backwards to improve clarity. Note that the left-hand and right-hand columns do not always have the same y-axis limits.

[Figure]

Figure S15. Aerosol properties for the set of pseudo-Lagrangian transects from flight '730a' (a) ΔOA/ΔCO (right y-axis) and ΔBC/ΔCO (left y-axis), (b) $\Delta f_{60}$ (right y-axis) and $\Delta f_{44}$ (left y-axis), (c) ΔH/ΔC (right y-axis) and ΔO/ΔC (left y-axis), (d) ΔN/ΔCO, and (e) $\overline{D_p}$ against physical age. For each transect, the data is divided into edge (the lowest 5-15% of ΔCO data; red points), core (90-100% of ΔCO data; blue points), and intermediate regions (15-50% and 50-90% of ΔCO data; light green and dark green points). ΔBC/ΔCO is shown in log scale to improve clarity.

[Figure]

Figure S16. Aerosol properties for the first set (left-hand column) and second set (right-hand column) of pseudo-Lagrangian transects from flight '730b' (a-b) ΔOA/ΔCO (right y-axis) and ΔBC/ΔCO (left y-axis), (c-d) $\Delta f_{60}$ (right y-axis) and $\Delta f_{44}$ (left y-axis), (e-f) ΔH/ΔC (right y-axis) and ΔO/ΔC (left y-axis), (g-h) ΔN/ΔCO, and (i-j) $\overline{D_p}$ against physical age. For each transect, the data is divided into edge (the lowest 5-15% of ΔCO data; red points), core (90-100% of ΔCO data; blue points), and intermediate regions (15-50% and 50-90% of ΔCO data; light green and dark green points). ΔBC/ΔCO is shown in log scale to improve clarity. Note that the left-hand and right-hand columns do not always have the same y-axis limits.

[Figure]

Figure S17. Aerosol properties for the set of pseudo-Lagrangian transects from flight '809a' (a) ΔOA/ΔCO (right y-axis) and ΔBC/ΔCO (left y-axis), (b) Δ$f_{60}$ (right y-axis) and Δ$f_{44}$ (left y-axis), (c) ΔH/ΔC (right y-axis) and ΔO/ΔC (left y-axis), (d) ΔN/ΔCO, and (e) $\overline{D_p}$ against physical age. For each transect, the data is divided into edge (the lowest 5-15% of ΔCO data; red points), core (90-100% of ΔCO data; blue points), and intermediate regions (15-50% and 50-90% of ΔCO data; light green and dark green points). ΔBC/ΔCO is shown in log scale and the x-axis for the right-hand column has been shifted backwards to improve clarity.

[Figure]

Figure S18. Aerosol properties for the set of pseudo-Lagrangian transects from flight '821b' (a) ΔOA/ΔCO (right y-axis) and ΔBC/ΔCO (left y-axis), (b) $\Delta f_{60}$ (right y-axis) and $\Delta f_{44}$ (left y-axis), (c) ΔH/ΔC (right y-axis) and ΔO/ΔC (left y-axis), (d) ΔN/ΔCO, and (e) $\overline{D_p}$ against physical age. For each transect, the data is divided into edge (the lowest 5-15% of ΔCO data; red points), core (90-100% of ΔCO data; blue points), and intermediate regions (15-50% and 50-90% of ΔCO data; light green and dark green points). ΔBC/ΔCO is shown in log scale and the x-axis for the right-hand column has been shifted backwards to improve clarity.

[Figure]

Figure S19. Various normalized parameters as a function of age for the 7 sets of pseudo-Lagrangian transects. Separate lines are shown for the edges (lowest 5-15% of ΔCO; dashed lines) cores (highest 90-100% of ΔCO; solid lines), and intermediate regions (15-50% and 50-90%; dotted and dashed-dot lines). (a) ΔOA/ΔCO, (b) $\Delta f_{60}$, (c) $\Delta f_{44}$, (d) ΔH/ΔC, (e) ΔO/ΔC, (f) $\Delta N_{40\text{-}262\ nm}$/ΔCO, and (g) $\overline{D_p}$ between 40-262 nm against physical age for all flights, colored by $\Delta OA_{initial}$. Some flights have missing data. Also provided is the Spearman correlation coefficient, R, between each variable and $\Delta OA_{initial}$ and physical age for each variable. Note that panels (a) and (f) have a log y-axis.

[Figure]

Figure S20. Various normalized parameters as a function of age for the 7 sets of pseudo-Lagrangian transects. Separate lines are shown for the edges (lowest 5-15% of ΔCO; dashed lines) and cores (highest 90-100% of ΔCO; solid lines). (a) $\Delta OA/\Delta CO$, (b) $\Delta f_{60}$, (c) $\Delta f_{44}$, (d) $\Delta H/\Delta C$, (e) $\Delta O/\Delta C$, (f) $\Delta N_{40\text{-}262\ nm}/\Delta CO$, and (g) $\overline{D_p}$ between 40-262 nm against physical age for all flights, colored by $\Delta OA_{initial}$. Some flights have missing data. Also provided is the Spearman correlation coefficient, R, between each variable and $\Delta OA_{initial}$ and physical age for each variable. Note that panels (a) and (f) have a log y-axis. This figure is identical to Figure 2 but uses an in-plume CO cutoff of 200 ppb.

[Figure]

Figure S21. Various normalized parameters as a function of age for the 7 sets of pseudo-Lagrangian transects. Separate lines are shown for the edges (lowest 5-25% of ΔCO; dashed lines) and cores (highest 75-100% of ΔCO; solid lines). (a) ΔOA/ΔCO, (b) $\Delta f_{60}$, (c) $\Delta f_{44}$, (d) ΔH/ΔC, (e) ΔO/ΔC, (f) $\Delta N_{40\text{-}262\ nm}$/ΔCO, and (g) $\overline{D_p}$ between 40-262 nm against physical age for all flights, colored by $\Delta OA_{initial}$. Some flights have missing data. Also provided is the Spearman correlation coefficient, R, between each variable and $\Delta OA_{initial}$ and physical age for each variable. Note that panels (a) and (f) have a log y-axis.  This figure is identical to Figure 2 but uses different ΔCO percentile widths.

[Figure]

Figure S22. Various normalized parameters as a function of age for the 7 sets of pseudo-Lagrangian transects. Separate lines are shown for the edges (lowest 5-15% of ΔCO; dashed lines) and cores (highest 90-100% of ΔCO; solid lines). (a) ΔOA/ΔCO, (b) $\Delta f_{60}$, (c) $\Delta f_{44}$, (d) ΔH/ΔC, (e) ΔO/ΔC, (f) $\Delta N_{40\text{-}262\ nm}$/ΔCO, and (g) $\overline{D_p}$ between 40-262 nm against physical age for all flights, colored by $\Delta OA_{initial}$. Some flights have missing data. Also provided is the Spearman correlation coefficient, R, between each variable and $\Delta OA_{initial}$ and physical age for each variable. Note that panels (a) and (f) have a log y-axis.  This figure is identical to Figure 2 except that it uses the location of the lowest 25% of CO data to determine the background concentrations of each species.

[Figure]

Figure S23. Calculated (final aerosol mass):(initial aerosol mass) ratios for mass loss through heterogeneous chemistry over a range of aerosol diameters and OH concentrations over 3 hours. As an upper-bound case, (a) it is assumed that for each OH collision, 200 amu of mass is lost. As a middle-bound, (b) it is assumed that 50% of OH collisions result in a 200 amu mass loss. As a more-realistic loss rate, (c) assumes that 10% of all OH collisions result in an 200 amu mass loss. See SI text S2 for more details.

[Figure]

Figure S24. Scatter plot of each parameter of Figure 1 against $\Delta OA_{initial}$.

[Figure]

Figure S25$4$. Raw $f_{60}$ data for each flight along each transect included in this study. The titles indicate the flight. The black color indicates the earliest transect, with increasingly lighter colors indicating increasingly downwind transects. The centerline was estimated from the number size distribution and the estimated center of the fire (Figures S1-S6).

[Figure]

Figure S26 5. Raw $f_{44}$ data for each flight along each transect included in this study. The titles indicate the flight. The black color indicates the earliest transect, with increasingly lighter colors indicating increasingly downwind transects. The centerline was estimated from the number size distribution and the estimated center of the fire (Figures S1-S6).

[Figure]

Figure S27. Total in-plume CO (ppbv) irradiance for each flight along each transect included in this study. The titles indicate the flight. The black color indicates the earliest transect, with increasingly lighter colors indicating increasingly downwind transects. The centerline was estimated from the number size distribution and the estimated center of the fire (Figures S1-S6).

[Figure]

Figure S28. Total in-plume shortwave (SW) irradiance for each flight along each transect included in this study. The titles indicate the flight. The black color indicates the earliest transect, with increasingly lighter colors indicating increasingly downwind transects. The centerline was estimated from the number size distribution and the estimated center of the fire (Figures S1-S6).

[Figure]

Figure S27. The Van Krevelen diagram of ΔH/ΔC versus ΔO/ΔC for all points in the 7 sets of pseudo-Lagrangian transects, colored by ΔOA$_{initial}$.

[Figure]

Figure S29. Measured versus predicted (a) $\Delta f_{60}$, (b) $\Delta f_{44}$, and (c) $\overline{D_p}$ between 40-262 nm, using the equation $ln(X) = a\ ln(\Delta OA_{initial}) + b\ ln(Physical\ age) + c$ (Eq. 5) where $X = \Delta f_{60}$, $\Delta f_{44}$, or $\overline{D_p}$. The values of a, b, and c are provided in Table S4. The Pearson and Spearman coefficients of determination ($R^2_p$ and $R^2_s$, respectively) are provided in each panel, along with the normalized mean bias (NMB) and normalized mean error (NME). Included in the fit and figure are all four regions within the plume (the 5-15%, 15-50%, 50-90%, and 90-100% of $\Delta CO$), all colored by the mean $\Delta OA_{initial}$ of each $\Delta CO$ percentile range.

[Figure]

Figure S30. Measured versus predicted (a) $\Delta f_{60}$, (b) $\Delta f_{44}$, and (c) $\overline{D_p}$ between 40-300 nm, using the equation $X = a\, log_{10}(\Delta N_{initial}) + b\, (Physical\ age) + c$ where $X=\Delta f_{60}$, $\Delta f_{44}$, or $\overline{D_p}$ where $X=\Delta f_{60}$, $\Delta f_{44}$, or $\overline{D_p}$. Note that the fit here is the same as that in Eq. 2 except that $\Delta N_{initial}$ replaces $\Delta OA_{initial}$. The values of a, b, and c are provided in Table S5. The Pearson and Spearman coefficients of determination ($R^2_p$ and $R^2_s$, respectively) are provided in each panel, along with the normalized mean bias (NMB) and normalized mean error (NME). Included in the fit and figure are all four regions within the plume (the 5-15%, 15-50%, 50-90%, and 90-100% of $\Delta CO$), all colored by the mean $\Delta OA_{initial}$ of each $\Delta CO$ percentile range.

[Figure]

Figure S31. (a) High resolution fits at m/z 44 for a biomass burning plume during 0730b research flight with laser ON. (b) Correlation of HR $CO_2^+$ ion and HR total ion signal at m/z 44, colored by distance downwind (km) from fire.

[Figure]

Figure S32. High resolution fits at m/z 60 for a biomass burning plume during 0730b research flight with laser ON.

[Figure]

Figure S33. Laser ON versus laser OFF SP-AMS HR O:C, UMR f44, and UMR f60 ratios.

[Figure]

Figure S31. High resolution fits at m/z 60 for a biomass burning plume during 0730b research flight with laser ON.

---

## Author Response (AR3)

We thank the reviewer for their response in the form of a marked-up pdf manuscript on our third submission. We provide a track changes document and respond here to any specific comments from the review that require a discussion (rather than updating the text). When we refer to line numbers, we will refer to the numbers in the marked-up pdf that the reviewer provided. The line numbers were unfortunately cut off past the one-hundredth place, so for clarify we will refer to page number and the last 2 digits of the line number. We quote reviewer remarks in black text. We will quote original, unaltered text in *pink italics* and new, edited text in red.

The new track-changes manuscript is included at the end of these responses (starting on page 6)

Reviewer response:
"This manuscript provides significant insight into aerosol aging in smoke plumes. However, I suggest revisions because the work needs further data analysis and revision to adequately support their conclusions."

We have enjoyed thinking through and responding to the many scientific issues debates the anonymous reviewers and editor have brought up over multiple rounds of review. This round of review restates issues that the reviewer had with the previous round of review without building upon or responding to our responses, limiting our ability to further change our analyses.

Page 2 line 55: We do not change our wording ('*Large, thick*' and '*small, thin*') as we think they are good descriptions of the range of smoke plumes observed.

Pge 3, lines 95-96: We agree that some of our initial sentences in the introduction are very similar and have deleted this first sentence to cut down on redundancies.

Page 4, Line 06-07: The reviewer states that they are skeptical of *"The differences in aerosol loading serve as a proxy for differences in initial fire and plume sizes, mass fluxes, and subsequent amount of dilution."* We hope that the combination of our analyses below and careful language here (that aerosol loading differences serve as a proxy) will bring up similar thoughts and debate within the scientific community. Additionally, we refer to our responses to the reviewer on this topic from earlier rounds of review.

Page 4 line 30: have added temperature

Page 4 line 33: have updated to "Results pertinent to changes in CE due to aging (including physical aging as well as chemical changes including oxidation, coating thickness, and sphericity) in smoke plumes are scarce (see discussion in Kleinman et al., 2020)."

Page 4 line 35: We are using UMR and have added this to the text.

Page 5 line 38: changed

Page 5 line 40: added high-resolution

Page 5 line 58/59: our current text states both the option the reviewer suggested but is also inclusive of e.g. the possibility of downwind sources.

Page 5 line 59: the rest of the paragraph discusses background so we don't think moving this statement will flow well.

Page 5 line 65-66: Original text is "*Mass concentrations of O, H, and C are calculated using the O/C and H/C and OA data from the SP-AMS (assuming all of the OA mass is from O, C, and H)*". The reviewer states that "elemental analysis gives mass concentrations of O, H, C used to calculate O/C, etc". We agree, however the data was provided as O/C and H/C so we had to back-calculate O and C and H. The reviewer also notes that OA mass could have a contribution from N and S, and we now acknowledge this.

Page 6 line 73: The reviewer asks on equation 2 "Why not just use mz60_in minus mz60_out".

$$\Delta f \;=\; \frac{(f_{in}*OA_{in}) - (f_{out}*OA_{out})}{\Delta OA} \qquad\qquad \text{Eq. 2}$$

We have access to f and OA, and so found the mass concentrations of the different mz values as the product of f*OA.

Page 6 line 76: The reviewer has multiple comments here,
1) Won't "lowest 10%" be more closely related to the amount of time spent in the plume vs out of the smoke plume?

Could you spend <10% our of the smoke plume and therefore your bkgd would be influenced by smoke?
This is true that less than 10% of the time could have been spent out of smoke. Visual inspection of the data collected during the flights (e.g number concentrations Figs S7-S11) indicate that this is almost certainly not the case however. As well, the lowest 10% of CO data fell under the 150 threshold.

2) Check why Figure 2 shows data for DeltaCO 5-15%
Our percentile bins are for Delta(CO) for within the data deemed "in plume" whereas our thresholds are for absolute CO. This is stated in line 74 of page 6, and we include absolute on line 76 to help keep variables straight in the reader's mind.

Page 6 line 96: There was an unnecessary space in the link. Fixed now.

Page 6 line 97: updated

Page 6 line 99-00: Text states "*The FIMS also showed additional smearing in flushing smoky air with cleaner air when exiting the plume with maximum observed flushing timescales around 30 seconds*"
The reviewer wrote
Not sure what "flushing" means? Does this refer to the residence time of air in the FIMS? Do Aerosols have a longer residence time than air?
We have changed to "The FIMS also showed additional smearing in when transitioning from high concentrations to lower concentrations with maximum observed lags of around 30 seconds behind CO decreases."
And
Shouldn't this be the same for high particle concentration air replacing clean air? (referring to "observed flushing timescales around 30 s")
Yes, we were puzzled by this as well, as the residence time in an instrument should cause the same lag when concentrations are increasing or decreasing. However, this is not what we observed (concentrations rose and peaked in step with CO, but lagged only when concentrations were decreasing) implying some non-linear process in the instrument or tubing.

Page 7 line 18: There was no previous note for BC (it is highlighted on page 5 but not commented on).

Page 7 line 25: The reviewer has highlighted the last portion of "*Note that in Figure 1 (and Figures S14-S18), the points represent the mean values for each transect/percentile and do not include error bars for uncertainty in the mean or measurement uncertainty as characterization of systematic variance (within plume percentiles) with age is beyond the scope of this study.*"
And comments:
without characterization of statistical uncertainty, there is no way of verifying trends in data with physical age or b/w core/intermediat/edge sections of the individual plumes!

We have addressed this concern in detail in our previous responses to reviewer comments. The reviewer has not discussed anything in our response specifically, so we defer to our prior responses.

Page 7 lines 28-30: We think the location of this statement is reasonable given the structure of the paragraph. We have clarified the 4 bins by changing the text to "edges, intermediate outer and intermediate inner regions, and the core"

Page 8 line 41: We cannot systematically characterize the variability between the transects, which could yield further insight if we could.

Page 8 lines 57-60: Our text here is

*We note that ΔOAinitial does not actually represent the true initial emitted OA from each fire, but instead serves as a proxy for the general fire size, intensity, and emission rate (as larger fires and fires with faster rates of fuel consumption per area will have larger mass fluxes than smaller fires or fires with less fuel consumption per area, all else equal).*

The reviewer states

I think this may be a big assumption. Emission factors (e.g. mass_CO/mass_fuel) can vary greatly (Akagi 2011, ACP). (*CO was not a great example)

In our second round of reviews, we discussed emissions factors in great detail, included in R4 and R53. We refer to these detailed responses, but in short, we showed that the variability due to fire size and dilution spans a much larger range than emission factors to the point where emission-factor variability will be in the noise of OA concentration variability due to fuel-burn rates, fire size, and dilution rates. The reviewer has not provided new arguments for us to discuss, and we hope that the community will take up this debate and work to prove or disprove our hypotheses that are supported by this one data set.

Page 9 line 73: The reviewer highlights our first mention of Spearman rank-order correlation tests and states:

This series of analysis does not make any sense to perform when the data are compiled in this way (all single transects together).

It is not expected to show any power and doesn't.

My suggestion is to remove the spearman test. Figure 2 is mildly informative and can stay.

Defn: Spearman Rank-Order Correlation Test.
1) convert x/y values to rank_x/rank_y
2) pearson's correlation rank_y vs rank_x

We have responded to the reviewer's concerns about the Spearman tests in the 2nd round of reviews, comment R5. The reviewer does not appear to be responding to our prior response but is instead restating their original comments. We understand that the reviewer does not agree with this approach but are unsure what further discussion to present to continue this dialogue.

Page 9 line 91-92: We discussed in round 2 of reviewers why we did the multivariate analysis and why we did not normalize for $\Delta OA_{initial}$, including responses to R5, R4, R52. The reviewer is not responding to arguments in our prior responses, and is restating their prior comment. We refer to our prior responses.

Page 10 lines 18-21: the heterogeneous chemistry calculation is a quick estimate that is detailed thoroughly in the SI. It is not meant to be a major part of our analysis and we do not think that is needs to be detailed in the methods.

Page 10 line 28: the reviewer doesn't provide a citation(s) for this comment, and we are unsure how to alter the text based on it.

Page 10 lines 34-35: The reviewer highlights *"The relative variability in the OA emission factor is much smaller than the relative variability in $\Delta OA_{initial}$, and other factors contributing to variability in $\Delta OA_{initial}$ will negate an emissions-based covariance between $\Delta OA_{initial}$ with $\Delta f_{60}$ and $\Delta f_{44}$."*
And comments
Logical flaw:

Variability in DeltaOA_initial could be b/c of fuel emission factors and atmospheric processes, e.g. dilution, condensation, evaporation.

If OA fuel emission factors is relatively constant, then the variability in DeltaOA_initial (by exclusion) has to do with atm. processes. These processes likely do influence f60 and f44, which is stated in Garofalo et al, Lee et al, and in this manuscript.
In R4 of the previous round, we provided 2.5 pages that discussed multiple lines of reasoning as to why emission factors cannot be a significant driver of >2 orders-of-magnitude variability in $\Delta OA_{initial}$. The reviewer is not addressing or acknowledging any of our points/analysis in this response but rather restates their earlier comment. We discuss earlier in the results, "While our observed $\Delta OA_{initial}$ in Figure 2 spans nearly a factor of 100, Andreae (2019) shows that the OA emission factors have a -1σ to +1σ range of around a factor 3." From this work, OA fuel emission factors are relatively constant compared to the variability in $\Delta OA_{initial}$.

Page 13 line 15-16: We note that this statement is for lower normalized number, not total number.

[revised manuscript text omitted]